# Impacts of multi-layer overlap on contrail radiative forcing

Inés Sanz-Morère[1], Sebastian D. Eastham[1], Florian Allroggen[1], Raymond L. Speth[1], Steven R. H. Barrett[1]

[1]Laboratory for Aviation and the Environment, Massachusetts Institute of Technology, Cambridge, MA 02139, United States of America

*Corresponding author*: Sebastian Eastham (seastham@mit.edu)

**Abstract.** Condensation trails ("contrails") which form behind aircraft are estimated to cause on the order of 50% of the total climate forcing of aviation, matching the total impact of all accumulated aviation-attributable $CO_2$. The climate impacts of these contrails are highly uncertain, in part due to the effect of overlap between contrails and other cloud layers. Although literature estimates suggest that overlap could change even the sign of contrail radiative forcing, the impacts of cloud-contrail overlaps are not well understood, and the effect of contrail-contrail overlap has never been quantified. In this study we develop and apply a new model of contrail radiative forcing which explicitly accounts for overlap between cloud layers. Assuming maximum possible overlap to provide an upper bound on impacts, cloud-contrail overlap is found to reduce the shortwave cooling effect attributable to aviation by 66%, while reducing the longwave warming effect by only 37%. Therefore, on average in 2015, cloud-contrail overlap increased the net radiative forcing from contrails. We also quantify the sensitivity of contrail radiative forcing to cloud cover with respect to geographic location. Clouds significantly increase warming at high latitudes and over sea, transforming cooling contrails into warming ones in the North-Atlantic corridor. Based on the same data, our results indicate that disregarding overlap between a given pair of contrail layers can result in longwave and shortwave radiative forcing being overestimated by up to 16% and 25% respectively, with the highest bias observed at high optical depths ($> 0.4$) and high solar zenith angles ($> 75°$). When applied to estimated global contrail coverage data for 2015, contrail-contrail overlap reduces

both the longwave and shortwave forcing by ~2% relative to calculations which ignore overlap. The effect

is greater for longwave radiation, resulting in a 3% net reduction in the estimated RF when overlap is correctly accounted for. This suggests that contrail-contrail overlap radiative effects can likely be neglected in estimates of the current-day environmental impacts of aviation. However, the effect of contrail-contrail overlap may increase in the future as the airline industry grows into new regions.

# 1 Introduction

Condensation trails ("contrails") are ice clouds which form in aircraft engine exhaust plumes. Contrails cause "cooling" effects, by scattering incoming shortwave solar radiation ($RF_{SW}$), as well as "warming" effects, by absorbing and re-emitting outgoing terrestrial radiation ($RF_{LW}$). Previous studies have found the latter effect to be dominant, particularly at night when the cooling effects associated with reductions in incoming shortwave radiation do not exist (Liou, 1986; Meerkötter et al., 1999). The difference between these two effects is the net contrail radiative forcing (RF) (Penner et al., 1999; IPCC 2013).

**Table 1.** Existing estimates of the longwave (LW), shortwave (SW), and net radiative forcing (RF) from contrails. ([1]: Estimated fuel burn for 2000 and 2002 taken from Olsen et al., 2013; [2]: From Ponater et al., 2002, which reports on the same data; [3]: contrail modeling corrected with observations (Lee et al., 2020); *: definition of "visible" varies between studies, and is clarified in the main text)

| Source | Target element | Target year | Fuel burn [Tg] | Global mean optical depth ($\overline{\tau}$) | $RF_{LW}$ [mW/m²] | $RF_{SW}$ [mW/m²] | Net RF [mW/m²] | Contrail modeling |
|---|---|---|---|---|---|---|---|---|
| Marquart et al., 2003 | Linear/visible* contrails (or lifetime < 5h) | 1992 | 112.0 | 0.15[2] | +4.9 | -1.4 | +3.5 | Fractional coverage in ECHAM4 |
| Frömming et al., 2011 | | 2000 | 152.0[1] | 0.08 | +7.9 | -2.0 | +5.9 | |
| Burkhardt and Kärcher, 2011 | | 2002 | 151.6 | / | +5.5 | -1.2 | +4.3 | CCMod in ECHAM4 |
| Spangenberg et al., 2013 | | 2006 | 151.6 | / | +9.6 | -3.9 | +5.7 | Coverage from Aqua MODIS |
| Burkhardt & Kärcher, 2011 | Contrail cirrus | 2002 | 154.0[1] | 0.05 | +47 | -9.6 | +38 | CCMod in ECHAM4 |
| Chen & Gettelman, 2013 | | 2006 | 151.6 | / | +41 | -26 | +15 | Fractional volume in CAM5 |
| | | | | | / | / | +57[3] | |
| Schumann et al., 2013 | | 2006 | 151.6 | ~0.2 | +126 | -77 | +49 | Lagrangian contrail model (CoCiP) |
| Schumann et al., 2015 | | 2006 | 151.6 | 0.34 | +143 | -80 | +63 | |
| Bock and Burkhardt, 2016 | | 2006 | 151.6 | / | / | / | +56 | CCMod in ECHAM5 |

The net radiative forcing impacts of contrails have been quantified using both global climate models (GCMs) (e.g. Chen and Gettelman, 2013; Ponater et al., 2002) and dedicated modeling approaches such as the Contrail Cirrus Prediction Tool (CoCiP) (Schumann, 2012) and the Contrail Evolution and Radiation Model (CERM) (Caiazzo et al., 2017). These approaches have resulted in estimates of total contrail radiative forcing ranging from +15.2 mW/m$^2$ (Chen and Gettelman, 2013) to +63.0 mW/m$^2$ (Schumann et al., 2015) for 2006, as shown in Table 1. Normalizing by the total aviation fuel burn in each given year, this gives a range of +0.1 to +0.4 mW/m$^2$/Tg. As such, the net radiative forcing impacts of contrails are comparable in magnitude to the radiative forcing impacts of aviation-attributable $CO_2$ emissions, which Lee et al. (2020) estimated at +0.11 mW/m$^2$/Tg for 2005.

The scaling of contrail radiative forcing impacts with future traffic growth will depend on multiple factors, especially (i) potential changes in contrail properties with changes in engine efficiency and the use of biofuels (Schumann, 2000; Caiazzo et al., 2017; Burkhardt et al., 2018; Kärcher, 2018); (ii) changes in background conditions due to climate change (Chen and Gettelman, 2016; Bock and Burkhardt, 2019); (iii) the emergence of new markets with different prevailing atmospheric conditions (Boeing, 2020); and (iv) increased likelihood of contrail-contrail overlap as existing markets and flight paths become more saturated. Major uncertainties in contrail radiative forcing estimation are related to the available data on ice supersaturation in the atmosphere, and the growth and lifetime of contrails (Schumann and Heymsfield, 2017, Kärcher, 2018; Lee et al., 2020).

The objective of this work is to provide a consistent, quantitative analysis of the effect of overlap between natural and artificial cloud layers (both cloud-contrail and contrail-contrail) on contrail radiative forcing. This includes both parametric analysis of individual columns and an assessment of how global contrail RF is affected. The impact of natural clouds on contrail radiative forcing has been repeatedly identified as an important contributor to overall contrail impacts, but significant uncertainty remains regarding the magnitude of the effect (Markowicz and Witek, 2011a; Schumann et al., 2012; Spangenberg et al., 2013; Schumann and Heymsfield, 2017). Contrail-contrail overlap has been modeled in the past as a component in contrail RF estimates, but no work has yet been published which quantifies its contribution to overall

forcing. Furthermore, the response of overlapping impacts to variations on local conditions, including cloud properties, atmospheric conditions, and surface properties, has not been parametrically quantified. This work aims to provide insight into how each of these factors affects the impact of multiple-layer overlap on contrail radiative forcing.

We start by reviewing existing literature on cloud layer overlap modeling in the context of contrails, including past studies modeling cloud-contrail and contrail-contrail overlaps (Section 2). We then present the radiative forcing model (Section 3.1) and its input data (Section 3.2), followed by the experimental design used to compute the effects and sensitivities of cloud layer overlap (Section 3.3).

We present three analyses. Firstly (Section 4.1), we perform a parametric study to quantify the effect of multiple layer overlap on the RF attributable to a single contrail. This includes the effect of variations in parameters such as optical depth and ambient temperature. Alongside this parametric evaluation, we also evaluate our model results against the widely-used Fu-Liou radiative transfer model. Secondly (Section
4.2), we expand this parametric analysis to quantify how the effect of overlap varies with location and season, using estimated global atmospheric data for 2015. Thirdly (Section 4.3), we estimate the specific contribution of multiple layer overlaps to the simulated 2015 global contrail radiative forcing, isolating both cloud-contrail and contrail-contrail impacts.

These analyses are followed by a discussion of limitations to our approach, and potential avenues for future research (Section 5). This includes limitations associated with the base RF model and with the representation of cloud overlap.

## 2 Review of past approaches for modeling cloud layers overlaps in contrail-related studies

Past studies have shown that overlapping with other cloud layers is likely to reduce both the shortwave
(cooling) and longwave (warming) RF associated with contrails. However, there is little agreement on how cloud-contrail overlap might change the net RF, due to uncertainty over whether they would more strongly mitigate the shortwave or longwave component. Meanwhile contrail-contrail specific impact on

global contrail RF has never been quantified. In this section we discuss previous literature addressing the treatment of multiple layer overlap in the context of contrail radiative forcing calculations.

## 2.1 Previous examples of cloud-contrail overlap modeling

**Table 2.** Previous evaluations of the effect that overlap with natural clouds has on contrail RF. MRO: Maximum-Random Overlap; defined by Geleyn and Hollingsworth (1978) as assuming that clouds in adjacent layers are maximally overlapping, while clouds separated by one or more clear layer are randomly overlapping. ECMWF: European Centre for Medium-Range Weather Forecasts. RT: Radiative Transfer. +/-/=: increase/ decrease/ remains the same – impact on net RF from clouds, +/- means that both effects have been found depending on cloud properties.

| Source | Cloud-contrail overlap model | Net effect of overlap on contrail RF | Comments |
|---|---|---|---|
| Minnis et al., 1999 | Contrail coverage randomly overlaps with other clouds | +/- | Effect varies with type of cloud |
| Meerkötter et al., 1999 | Experiments testing various RT models | + | Effect of low-level cloud |
| Myhre and Stordal, 2001 | Fixed contrail altitude and monthly mean cloud data (ECMWF cloud coverage) | = | Effect in a 1% homogeneous contrail cover |
| Marquart et al., 2003 | MRO used for each vertical column | - | 10% reduction |
| Stuber and Forster, 2007 | No information on overlap model | - | 7% reduction |
| Rädel and Shine, 2008 | Random overlap | - | 8% reduction |
| Myhre et al., 2009 | Experiments testing various RT models | - | Effect in a 1% homogeneous contrail cover (see Table S1) |
| Rap et al., 2010 | Random overlap | - | Reduction up to 40% with on-line model |
| Frömming et al., 2011 | MRO used for each vertical column | - | Largest impacts occurring with few clouds |
| Markowicz and Witek, 2011a | 15 cloud overlap scenarios | +/- | Effect varies with assumed crystal shape in contrails |
| Schumann et al., 2012 | Parametric RF model used with CoCiP, calculates RF as a function of upward fluxes below the contrail and optical depth of clouds above the contrail | +/- | Effect varies with type of cloud |
| Yi et al., 2012 | Experiments testing sensitivity to overlap assumption | - | 7% reduction in the random overlap case |
| Spangenberg et al., 2013 | Aqua MODIS 1 km data (Minnis et al. 2008) used to classify cloudy pixels | +/- | Effect varies with type of cloud |

Studies have used observational data to quantify the effect of natural overlap on contrail RF. Spangenberg et al. (2013) found a reduction in both $|RF_{LW}|$ and $|RF_{SW}|$ from contrails in the presence of natural clouds,

with $|RF_{SW}|$ falling by 30% [40%] in the presence of ice [water] clouds. This is in part because of the optical properties of the clouds, but also because of the different thicknesses, temperatures, and altitudes of the observed clouds. The difference in shortwave effects resulted in a decrease in net RF when overlapping with ice clouds, but an increase when overlapping with liquid clouds – demonstrating the difficulty of evaluating the impact of natural clouds on the net contrail RF. Another assessment using a simple model of contrail coverage based on observational data indicated that, while low-level marine clouds could significantly increase contrail net RF, cirrus clouds could have the opposite impact by more significantly reducing $RF_{LW}$ than $RF_{SW}$ (Minnis et al., 1999).

Single-column analyses have also been performed. An estimate using fixed global contrail coverage for a single month from Myhre and Stordal (2001) found that the net impact of cloud overlap on contrail RF is close to zero, as the effect on $RF_{LW}$ and $RF_{SW}$ was similar. They performed no specific evaluation of the dependence on local conditions and cloud properties. Another study by Myhre et al. (2009), comparing multiple radiative transfer models, found a consistent reduction in contrail RF due to natural clouds, with a maximum decrease of 14%. Meerkötter et al. (1999) also compared radiative transfer models, including the effect of crystal shape and optical depth. They found that the presence of low-level clouds increases the net radiative forcing due to contrails.

A parameterization for line–shaped contrails in a general circulation model (GCM) was presented by Ponater et al. (2002) for ECHAM4 (version 4 European Center/Hamburg General Circulation Model). A later amendment suggested that the assumption of maximum random overlap can cause $RF_{LW}$ to be underestimated by 70% when using certain radiative transfer parameterizations (Marquart et al., 2003, Marquart and Mayer, 2002). This indicates the extent of the sensitivity of contrail RF to the assumed overlap scheme.

Marquart et al. (2003), again using ECHAM4, estimated a 10% reduction in linear contrail RF due to the presence of natural clouds. Frömming et al. (2011), using the same model, found the largest radiative

impact to occur over regions with few natural clouds. Stuber and Forster (2007) similarly found a 7% reduction in contrail RF due to cloud overlap when accounting for diurnal variations in air traffic.

Both Rädel and Shine (2008) and Rap et al (2010) found a reduction in global net RF of approximately 10%, with both $|RF_{LW}|$ and $|RF_{SW}|$ reduced by up to 40% due to cloud masking effects. Rap et al. (2010), adapting Ponater et al.'s (2002) contrail parameterization scheme to the UK Met Office climate model, also found a correlation between contrail and natural clouds, showing the importance of using accurate (and consistent) natural cloud cover data. Markowicz and Witek (2011a) extended these results by
evaluating the role of crystal structure. While still finding a mean net impact on global contrail RF of less than 10%, they also found that this impact changes sign depending on the assumed contrail crystal habit.

CAM5 (Community Atmospheric Model version 5) has also been used to estimate global contrail RF (Yi et al., 2012; Chen and Gettelman, 2013). In Yi et al. (2012) they assess the sensitivity to the assumed
form of overlap. Global contrail net RF is reduced by 15% when switching the cloud-contrail overlap assumption from random to maximum random (Geleyn and Hollingsworth, 1978). This shows that the choice of overlap scheme can significantly modify the estimated global RF.

Lagrangian models have also been used to simulate contrails, including CoCiP (Schumann, 2012) and
CERM (Caiazzo et al., 2017). Both models compute the RF of a single contrail using a parametrization which takes into account changes in contrail RF caused by clouds below the contrail. It incorporates contrail properties (temperature, optical depth, ice particle effective radius, and ice particle habit), upward radiative fluxes from below each contrail, the solar constant for the given time of year, the solar zenith angle, and the optical depth of clouds above the contrail. Using this approach, Schumann et al (2012)
concluded that net RF may increase if contrails overlap with low-level clouds, but may change sign if passing underneath natural cirrus clouds. This again demonstrates the need to accurately model natural clouds when simulating contrails. However, simulations of single contrails using this approach cannot easily account for multiple-contrail radiative interactions.

Approaches estimating the impact of cloud overlap on contrail radiative forcing and their results are summarized in Table 2. The disagreement in these estimates is in large part due to the nature of competing longwave and shortwave components, but also due to uncertainty regarding the role that specific cloud properties and parameters might have in changing the effect of overlap on contrail RF. We here aim to provide additional insight into these relationships through a parametric analysis (Section 4), extending

from a single column up to the global-scale effects of cloud-contrail overlap on contrail RF.

## 2.2 Previous examples of contrail-contrail overlap modeling

**Table 3.** Existing methods for modeling contrail-contrail overlap when estimating global contrail RF. MRO: Maximum-Random Overlap, defined by Geleyn and Hollingsworth (1978) as assuming that clouds in adjacent layers maximally overlap while clouds separated by one or more clear layer randomly overlap.

| Source | Model used to represent contrail-contrail overlap |
| --- | --- |
| Minnis et al., 1999 | No overlap considered (fractional coverage from observations) |
| Marquart et al., 2003 | MRO in the vertical for each column |
| Rädel and Shine, 2008 | Random overlap |
| Rap et al., 2010 | Random overlap |
| Frömming et al., 2011 | MRO in the vertical for each column |
| Burkhardt and Kärcher, 2011 | MRO in the vertical for each column |
| Chen and Gettelman, 2013 | Zero contrail-contrail overlap in grid box |
| Schumann et al., 2013 | Linear RF addition |
| Bock and Burkhardt, 2016 | MRO in the vertical for each column |


When contrails are simulated in global climate models, contrails (and contrail overlaps) are treated in several different ways (see Table 3). Contrail parametrizations have been developed for ECHAM4 (Ponater et al., 2002; Burkhardt and Kärcher, 2009) in which maximum random overlap is assumed between contrail and cloud layers (Marquart et al., 2003; Frömming et al., 2011; Burkhardt and Kärcher,

2011; Bock and Burkhardt, 2016). Rädel and Shine (2008) and Rap et al. (2010) also employ this parameterization, calibrating the results using satellite observations. Chen and Gettelman (2013) also implemented contrails in the CAM5 model, representing them as an increase in the 3-D cloud fraction. However, they assumed zero overlap between linear contrails if located in same vertical level (~1 km).

Finally, the CoCiP Lagrangian contrail model (Schumann, 2012) indirectly models contrail-contrail
overlaps by linearly summing the RF of all contrails while accounting for any cirrus which was observed
above the simulated contrail. However, this does not explicitly account for overlap between simulated
contrails.

Differences can be observed in the way contrail-contrail overlaps are modeled in literature. While the
optimal approach is not clear, no study to date has quantified the effect of contrail-contrail overlap on
global contrail RF. Assuming continued growth in the aviation industry, more instances of contrail
overlap can be expected to occur. Better understanding of the magnitude and behavior of contrail-contrail
overlap is therefore needed. In this work, we aim to provide insight into the factors which affect the sign
and magnitude of changes in contrail RF due to contrail-contrail overlap. We also provide a first
quantification of the current-day magnitude of its effect on global contrail RF.

## 3 Method

The modeling approach is based on a radiative transfer model previously developed to simulate natural
clouds, which we extend to simulate multiple contrail cloud layers. Section 3.1 describes the model and
compares the results against existing approaches, and Section 3.2 describes the input data. Using this
model, we develop a series of simulations - described in Section 3.3 - which quantify the net radiative
forcing impacts of contrail-contrail overlaps and cloud-contrail overlaps under different conditions.

### 3.1 The radiative forcing model

The net radiative forcing (RF) from contrails is the sum of two components: longwave (LW) and
shortwave (SW). Shortwave radiation is the incoming radiation flux from the sun, which typically
undergoes scattering and reflection with minimal atmospheric absorption. Longwave ("terrestrial")
radiation is the emission of longer-wavelength infrared radiation by the Earth, which undergoes minimal
scattering or reflection but is strongly absorbed by clouds before being re-emitted. Contrail cloud layers
induce a negative shortwave RF during the day since they reflect incoming solar radiation, slightly
increasing the global mean albedo. However, as in the case of natural cirrus clouds, the longwave RF

impacts of contrails during both day and night are positive. This is because they absorb terrestrial radiation and re-emit it at the lower temperatures of the upper troposphere (Penner et al., 1999).

In this study we extend and use a cloud radiative transfer model first described by Corti and Peter (2009) which can be applied to both natural or artificial cloud layers (e.g. contrails). This model calculates the

cloud-induced change in outgoing longwave and shortwave radiation based on simulated or observed surface conditions (albedo and surface temperature), outgoing longwave flux, meteorological data (ambient temperature), and cloud coverage. The radiative forcing (RF) attributable to a single cloud layer is calculated using two simulations: one with the cloud layer present, and one without. The instantaneous RF of a cloud layer is then defined as the difference between the net radiative flux at the top of the

atmosphere with and without the layer (IPCC, 2013), so a positive net radiative forcing impact implies an increase in the net energy of the Earth-atmosphere system.

### 3.1.1 Summary of the single cloud layer RF model

We calculate a single contrail's radiative forcing as the sum of $RF_{LW}$ and $RF_{SW}$. These terms are calculated

as

$$\mathbf{RF_{LW}} = \boldsymbol{\varepsilon} \times \mathbf{OLR_{clear}} - \boldsymbol{L_c} = \boldsymbol{\varepsilon} \times \mathbf{OLR_{clear}} - \boldsymbol{\varepsilon \sigma^* T_c}^{k*} \tag{1}$$

$$\mathbf{RF_{SW}} = -\boldsymbol{S} \cdot \boldsymbol{t} \cdot (\mathbf{1} - \boldsymbol{\alpha}) \left( \frac{\boldsymbol{R_c} - \boldsymbol{\alpha R_c'}}{\mathbf{1} - \boldsymbol{\alpha R_c'}} \right) \tag{2}$$

where $OLR_{clear}$ is the outgoing longwave radiation from the surface of the Earth (in W/m$^2$); $L_c$ is the total outgoing longwave radiation from the cloud (in W/m$^2$); $T_c$ is the cloud temperature (in K); $\varepsilon$ is the contrail

emissivity; and $\sigma^*$ (the adjusted Stefan-Boltzmann constant, in W/m$^2$/K$^{-2.528}$) and $k^* (= 2.528)$ are constants (Corti and Peter, 2009). $S$ is the incident solar radiation (in W/m$^2$); $R_c$ is the cloud reflectance for direct radiation; $R_c'$ is the cloud reflectance for diffuse radiation; $\alpha$ is the albedo of the Earth; and $t$ is the atmospheric transmittance above the cloud.

A more detailed description of the single contrail radiative forcing model, which is an extension of that described by Corti and Peter (2009), is provided in Appendix A. This includes a description of the calculations of key parameters, model assumptions, definitions of the required input data, such as satellite atmospheric data and contrail coverage data (see Section A3), and a discussion of the merits and issues with using clear-sky OLR as opposed to (for example) all-sky OLR.


The performance of this model for simulations of single contrails is evaluated in Appendix B. Model outputs are compared with other two existing and widely used radiative transfer models, FL (Fu and Liou, 1992, 1993; Fu, 1996; Fu et al., 1997) and CoCiP (Schumann, 2012). We obtain, for $RF_{SW}$, a difference of less than 15% for $\theta < 80°$ (with smaller differences at smaller solar zenith angles). At high solar zenith 255 angles ($\theta > 80°$), this difference can grow to up to 20%, while the difference in $RF_{LW}$ is always within 10%.

### 3.1.2 Extension to multiple layers

To quantify the effect of cloud-contrail or contrail-contrail overlaps, we extend the model to account for multiple overlapping layers. Computation of longwave RF is accomplished by working outwards from 260 the Earth's surface, as shown in Fig. 1, with each layer absorbing some fraction $\varepsilon_i$ of the incident longwave radiation while re-emitting a total flux of $\varepsilon_i \sigma^* T_i^{k*}$. This approach assumes each cloud layer to be at the temperature of the surrounding atmosphere, so that temperature feedbacks can be disregarded and longwave radiation absorption and re-emission is derived from local temperature and surface temperature. Downward fluxes are not shown because the approach neglects temperature feedbacks. As 265 a result, only outgoing radiation is used in our RF calculations. As in the model used by Corti and Peter (2009), applying this approach for a single cloud layer produces a longwave RF which is proportional to the temperature difference between the cloud and the ground. Finally, based on the approach followed in Schumann et al. (2012), we assume $RF_{LW}$ to be always nonnegative, setting the longwave radiative forcing of a contrail to be zero when its temperature is higher than lower layers.

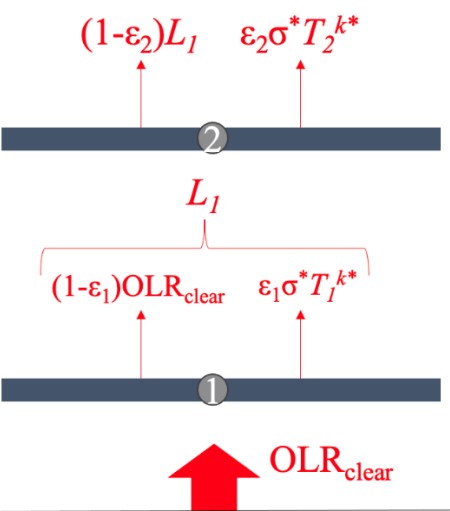

$(1-\varepsilon_2)L_1 \quad \varepsilon_2\sigma^*T_2^{k*}$

②

$L_1$

$(1-\varepsilon_1)\text{OLR}_{\text{clear}} \quad \varepsilon_1\sigma^*T_1^{k*}$

①

$\text{OLR}_{\text{clear}}$


**Figure 1**. Schematic of longwave RF calculation in a two-layer overlap. Arrows represent emitted or transmitted longwave radiation. $\text{OLR}_{\text{clear}}$ is the longwave emission from the Earth's surface, while $L_i$ is the longwave emission from layer $i$. $\varepsilon_i$ and $T_i$ are emissivity and temperature of each of the layers.

As in Corti and Peter (2009), to calculate the shortwave RF we start by estimating the shortwave radiation

impact of each cloud layer. Per unit of direct incident shortwave radiation, a fraction $R_c$ of shortwave radiation is reflected and $(1 - R_c)$ is transmitted (absorption of shortwave radiation is assumed to be negligible). The same approach is taken for diffuse shortwave radiation, this time using the parameter $R'_c$. The parameters $R_c$ and $R'_c$ are calculated as

$$R_c = \frac{\tau/\mu}{\gamma + \tau/\mu} \tag{3}$$

$$R'_c = \frac{2\tau}{\gamma + 2\tau} \tag{4}$$


where $\tau$ is the optical depth of the cloud layer, $\mu$ the cosine of the solar zenith angle $\theta$, and $\gamma = 1/(1-g)$ where $g$ is the layer asymmetry parameter.

Due to the high degree of forward scattering of clouds and contrails (Baran, 2012; Nousiainen and

McFarquhar, 2004; Yang et al., 2003; Kokhanovsky, 2004), we further assume that (i) shortwave radiation, which has not yet impinged on the Earth's surface, is direct; and (ii) any shortwave radiation

reflected from the Earth's surface is diffuse (Corti and Peter, 2009). With these assumptions, the total radiative forcing of two overlapping layers with identical asymmetry parameters is analytically equal to the radiative forcing of a single layer with an optical depth equal to the sum of that from both layers. A full derivation of this result is given in Section C1 for any number of layers.

To model the shortwave radiation impacts of multiple layers, we then collapse the cloud layers into an equivalent single effective layer. To characterize this layer, we derive the effective asymmetry parameter of the overlapping system (Section C2). For $N$ overlapping layers, this is calculated using the optical depth-weighted average value of the gamma function

$$\gamma_w = \left(\prod_{i=1}^{N} \gamma_i\right) \frac{\sum_{i=1}^{N} \tau_i}{\sum_{i=1}^{N} \prod_{j \neq i} \gamma_j \tau_i} \tag{5}$$

where $\tau_i$ and $\gamma_i$ are the optical depth and gamma function $(1/(1-g_i))$ respectively for each individual layer. Using the effective gamma function, we can then derive $R_c$ and $R_c'$ as shown in Eq. (3) and (4) for the full stack of overlapping layers. Substituting (3), (4) and (5) back into Eq. (2), we obtain the radiative forcing components for $N$ overlapping cloud layers as

$$RF_{SW,O} = -S \cdot t \cdot (1 - \alpha) \frac{R_c - \alpha R_c'}{1 - \alpha R_c'} \tag{6}$$

and then, this can be combined with the previously mentioned procedure for $RF_{LW}$ (Fig. 1) applied to $N$ overlapping cloud layers

$$RF_{LW,O} = OLR_{clear} - \left[ OLR_{clear} \prod_{i=1}^{N} (1 - \varepsilon_i) + \sum_{i=1}^{N} \left[ \prod_{j=i+1}^{N} (1 - \varepsilon_j) \right] \varepsilon_i \sigma^* T_i^{k^*} \right] \tag{7}$$

## 3.2 Input data for the radiative forcing model

Section A3 defines the input data required for single-contrail RF calculations. This includes an estimate of global contrail coverage, generated for this study using the CERM contrail model (Caiazzo et al., 2017), and our use of CERES observations (NASA Langley Research Center Atmospheric Science Data Center, 2015) to provide estimates of atmospheric radiation fluxes and "natural" cloudiness. However, we include here a brief discussion of the definition of overlap and of some limitations in our use of the CERES dataset, due to their specific importance to this work.

### 3.2.1 Contrail-contrail and cloud-contrail overlap definition

CERM does not provide the position and orientation of contrails within each grid cell. As such, contrail overlap is computed by assuming the maximum possible overlap, which provides an upper-bound estimate of total overlap. This approach assumes that the smallest contrail (by area) in each vertical column is fully overlapped with all other contrails in the column, repeating the process for all subsequent contrails in the column. If clouds are present in a vertical column, we assume that they overlap with any contrails which are present, resulting in an upper bound estimate of overlap impacts.

A limitation of the CERM modeling approach is that contrails which form within the same hour, grid cell, and vertical layer (~350 m thick at cruise altitude) are aggregated into a single contrail layer. This means that overlap which would occur between contrails forming in close proximity is not included in our estimate of the effects of contrail-contrail overlap.

The approach used to model cloud-contrail overlap varies in literature, with most assuming random or maximum random overlap. We instead assume maximum overlap in our calculations. This approach was also used by (e.g.) Spangenberg et al (2013) and Schumann (2012), where it was implemented either by reducing radiation reaching contrail layers or by modeling contrails as an increase in cloud fraction (see Table 2). This is consistent with the fact that cloud coverage is in general larger than contrail coverage.

Contrail-contrail overlaps are modeled assuming maximum random overlap (see Table 3) in most climate models where contrails are implemented, compared to maximum overlap in this work and in CoCiP's RF calculations (Schumann et al., 2012). Ideally, additional information is provided regarding contrail orientation. In flight corridors where large numbers of aircraft pass within several hours of each other and

with similar (or opposite) headings, overlapping, aligned contrails may be more common. However, this might not happen in denser flight areas like mainland US. Using information on flight paths to include contrail orientation in contrail modeling tools would be useful to more accurately model the impact of contrail-contrail overlaps on contrail radiative forcing. This and other avenues for improvement, such as through the use of higher vertical resolution, are discussed in Section 5.

### 3.2.2 Natural cloud data

CERES instruments also provide data on natural cloud coverage, with cloud detection based on algorithms described by Minnis et al. (2008). These detections are divided into four vertical levels defined by pressure, and include cloud properties such as optical depth and temperature. We use this data to estimate natural cloud cover when calculating the impacts of contrails in 2015. The detection limit of the CERES

instruments has been estimated as approximately $\tau = 0.02$ (Dessler and Yang, 2003), although later studies have suggested it may be closer to $\tau = 0.05$ (Kärcher et al., 2009).

Since CERES instruments provide data on only the sum of detected clouds (including visible contrails), we may be double counting the influence of contrails. Four levels of clouds are given in CERES data,

defined by their pressure level and corresponding to the following altitudes: from 0 to 10,000 ft, from 10,000 ft to 16,500 ft, from 16,500 ft to 30,000 ft, and above 30,000 ft. Accordingly, most contrails would appear in the 4th level detection.

There is a high-level cloud in the same location as a "CERES-detectable" contrail (optical depth greater

than 0.02) in 58% of contrail cases, whereas only 6% of simulated contrails are found in the mid-level cloud attitude range (the 3rd CERES vertical level). There is in theory the possibility that ~60% of all

contrails are already accounted for in the CERES data. However, considering that the average optical depth from CERM for 2015 global contrails is 0.065, a significant fraction of the simulated contrails are not detectable by CERES, limiting the likelihood of double-counting. Additionally, satellite detection limits do not affect our contrail coverage data meaning that this study includes subvisible contrails in impact and RF calculations.

Finally, contrail cirrus may also modify "natural" cloud coverage by changing the availability of atmospheric water. Any such effects would be inherently included in observations, including those retrieved by CERES for year 2015. Our approach does not allow us to separate out the effect of this interaction, but its impact has previously been estimated to reduce global contrail radiative forcing by approximately a fifth in Burkhardt and Kärcher (2011), by 15% in Schumann et al. (2015) and by a local maximum of 41% in Bickel et al. (2020).

## 3.3 Experimental design

We analyze the radiative forcing impacts of cloud-contrail and contrail-contrail overlaps using a three-step approach.

In the first step, through a parameterized analysis, we quantify the effect of a two-layer overlap on total radiative forcing when compared to a case where the layers are assumed to be independent, calculating how the effect of overlap varies as a function of the layer properties and the local conditions. This analysis shows the conditions under which the RF of two overlapping contrails is significantly different to the total RF of two independent contrails.

In the second step, we evaluate the global sensitivity of contrail RF to cloud-contrail and contrail-contrail overlaps using 2015 atmospheric data (meteorology and natural clouds). We calculate the RF associated with one or two contrail layers at each global location for one day from each month of the year in order to capture seasonal variation. To demonstrate this, we simulate a case used previously in estimates of

contrail radiative forcing (Myhre et al., 2009; Schumann et al., 2012). The RF attributable to a
hypothetical contrail is calculated for each location globally assuming typical optical properties ($g = 0.77$),
optical depth (0.3), and altitude (around 10.5 km). In order to quantify the effect of cloud overlap we
evaluate radiative forcing with and without natural cloud cover ("all sky" vs. "clear sky"). By subtracting
the RF obtained in the "clear sky" scenario from the RF obtained in the "all sky" scenario, we obtain the
difference in contrail RF attributable to the presence of clouds. The results can then be linked to different
cloudiness conditions to systematically analyze the impact of cloudiness on contrail RF. In order to
quantify the global sensitivity to contrail-contrail overlaps we simulate a superposition of two contrail
layers at each location, separated by a vertical distance of approximately 0.5 km.

Finally, we quantify the effect of cloud-contrail and contrail-contrail overlap on the global contrail RF in
2015. We use year-2015 contrail coverage data obtained from CERM (Caiazzo et al., 2017) and analyze
the associated radiative forcing impacts for the four scenarios shown in Table 4.

**Table 4.** Scenarios analyzed for 2015 global contrails.

| | Cloudiness Assumption | |
| --- | --- | --- |
| **Contrail overlap assumption** | **Clear sky (no clouds) (C)** | **All sky (clouds) (A)** |
| *Independent (I)* | IC | IA |
| *Overlapping (O)* | OC | OA |

Global evaluations are performed using detailed contrail coverage estimates and meteorological data
described in Appendix B.

## 4 Results

### 4.1 The effect of overlap on contrail radiative forcing in a single column

In this section we evaluate the general effect of overlap on contrail RF through a parameterized analysis.
We simulate two overlapping layers with different optical depths ($\tau$) and temperatures (T) (either natural
cloud or contrail). By varying the layer properties, we are able to simulate both cloud-contrail and

contrail-contrail overlaps. We also evaluate the effect of solar zenith angle ($\theta$), estimated outgoing longwave radiation without clouds ($OLR_{clear}$), and Earth surface albedo ($\alpha$).

The contrail modeling and observation literature suggests that contrails are usually optically thin, with typical optical depths in the range 0 to 0.35 (see Table 1). They also form almost exclusively at cruise altitude. Natural clouds are located within a greater range of altitudes and can achieve greater optical depths. We simulate contrail layers over a range of depths ($0 < \tau < 0.5$), based on typical values, at low temperatures/high altitudes (210 – 230 K), and with an asymmetry parameter of 0.77, representative of mature contrails (Heymsfield et al. 1998; Febvre et al., 2009; Markowicz and Witek, 2011a; Gayet et al., 2012; Schumann et al., 2017; Sanz-Morère et al, 2020). Cloud layers are simulated as being thicker ($0 < \tau < 4$), at higher temperatures/lower altitudes (215 – 280 K), and with an asymmetry parameter of 0.85, corresponding to low level clouds. When not otherwise specified, we assume each contrail layer to have an optical depth $\tau$ of 0.3 and temperatures of 215 K (upper) and 220 K (lower). This optical depth is at the upper bound of literature estimates of typical values for contrails (Voigt et al., 2011). For this analysis natural cloud layers are assumed to have an optical depth $\tau$ of 3 and a temperature of 260 K. The prescribed outgoing longwave radiation in this single-column analysis is 265 W/m$^2$ (consistent with a ~288 K surface temperature), with an albedo $\alpha = 0.3$ and solar zenith angle $\theta = 45°$.

The total forcing for the combined, overlapping layers is calculated as shown in Section 3.1.2. We calculate the "independent" forcing as the RF that would have been calculated by adding together the RF from each layer independently, without accounting for any overlap. We evaluate the effect that overlap has on the contrail net radiative forcing in both systems (cloud-contrail and contrail-contrail) as a function of each parameter (Section 4.1.1). We then calculate the error in estimated RF that results if overlap is ignored (Section 4.1.2). We also evaluate contrail RF when surrounded by cirrus clouds (Section 4.1.3) and finally, we compare our overlap model (Section 4.1.4) with the FL model described in Appendix B (Fu and Liou, 1992, 1993; Fu, 1996; Fu et al., 1997).

### 4.1.1 Parametric analysis of cloud-contrail and contrail-contrail overlap effects on contrail net RF

The effect of overlap on contrail RF depends both on cloud layers' properties and on local conditions. We first evaluate how the effect of overlap varies with cloud layer properties, including thickness of the two layers. We then quantify the effect of local conditions: solar zenith angle ($\theta$), estimated outgoing longwave radiation in clear sky conditions ($OLR_{clear}$), and Earth surface albedo ($\alpha$).

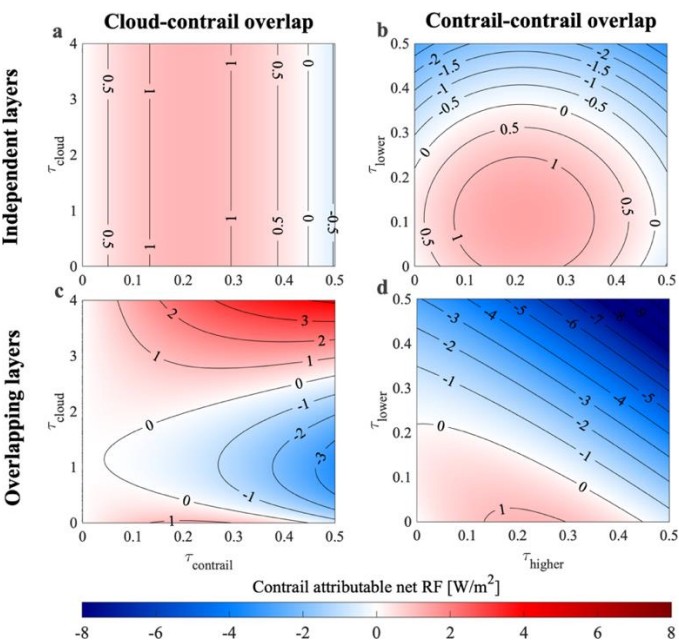

**Figure 2.** Effect of overlap between two layers on the contrail net RF as a function of optical depth $\tau$. Left: RF attributable to a single contrail when overlapping with a natural cloud layer. Right: total RF in a system of two overlapping contrails. Top: contrail RF estimated when treating the layers as independent and summing individual contributions. Bottom: contrail RF estimated in a single calculation which accounts for overlap. Negative RF is shown in blue and positive RF is shown in red. Contrail properties are: asymmetry parameter of 0.77, temperature of 220 K and 215 K respectively. Cloud properties are: asymmetry parameter of 0.85, temperature of 260 K. The solar zenith angle $\theta = 45°$ for all calculations. An additional version of this figure, calculated using a solar zenith angle $\theta = 30°$ and covering a greater range of optical depths, is provided as Figure S3 for comparison to other literature.

We evaluate the effect of overlap on contrail net RF for both cloud-contrail (with the contrail at 215 K) and contrail-contrail (at 215 K and 220 K) systems. The variation in contrail net RF with optical depth of either layer is shown in Figure 2. A decomposition of the results in terms of longwave and shortwave components can be found in the SI (Figures S1 and S2). The panels on the left show the effects of cloud-contrail overlap, while those on the right show the effects of contrail-contrail overlap. The upper row shows the net RF when the layers are considered to be independent, while the bottom row shows the RF

when accounting for overlap between the two. Each panel shows the net, contrail RF of the system (i.e. subtracting only any RF which is calculated when no contrails are simulated).

The RF attributable to a single contrail (no overlap) as a function of its optical depth is shown in the upper left panel (Fig. 2a). This is because, when overlap is ignored, the contrail RF of a cloud-contrail system is equal to the RF of the contrail alone. The RF increases from zero to a maximum of ~1.2 W/m$^2$ as the optical depth increases to ~0.2, after which increasing depth instead results in reduced RF. This is due to

460 the compensation of the increase in absorption by the increase in reflectance with increasing optical depth. The lower left panel (Fig. 2c) then shows how the presence of a cloud layer affects contrail RF as a function of the optical depth of each layer. The presence of a (lower) natural cloud layer can either increase or decrease the contrail RF depending on the optical depth of the cloud layer. Thin clouds can transform a warming contrail into a cooling one by absorbing part of the longwave radiation that

previously reached the contrail. Thick clouds can transform a cooling contrail into a warming one (from a net RF of -0.54 W/m$^2$ to +4.1 W/m$^2$ at a contrail optical depth of 0.5) by mitigating the shortwave cooling of the contrail. These results explain the existing uncertainty related to the effect of natural clouds on contrails' radiative impact. If overlap between the layers is ignored (Fig. 2a), these features are not captured.

Figure 3d shows the effect of contrail-contrail overlaps on contrail RF. The effect of each contrail individually can be seen on the values along the left and lower edges. The lower contrail, due to its higher temperature (less LW absorption), becomes cooling at a lower optical depth of ~0.22 (compared to ~0.45 for the upper contrail). The effects of overlap are similar to the effects obtained when a thin cloud ($\tau \sim$

0.1) is overlapping with a contrail: the net effect of increasing the optical depth of the contrail is to make the system more cooling (Fig. 2d). However, since both layers are thin (contrails), increasing the optical depth of either layer yields a more negative RF, unlike the case of a thick natural cloud with a thin contrail. This is because the shortwave cooling attributable to contrails increases regardless of which layer is providing the shortwave cooling. This results in a monotonic decrease in warming (increase in cooling)

attributable to the net contrail RF, from +1.2 W/m$^2$ for a single contrail of optical depth 0.25, to -10 W/m$^2$

for two contrails both of optical depth 0.5. For comparison, Figure 2b (upper right panel) shows the result when RF is calculated based on the independent combination of each contrail's RF. Independent calculation gives the wrong response by neglecting the screening effect on longwave radiation by the lower contrail. This error is small for low contrail thicknesses, with a maximum difference of -1.0 W/m$^2$ for a total contrail-contrail system thickness below approximately 0.15. However, for thicker contrail layers, both the sign and magnitude of the net effect can be incorrectly predicted when overlap is neglected. This analysis also confirms the findings of Kärcher and Burkhardt (2013) with regards to the overestimation of contrail RF by prescribing a mean optical depth. As an example, two simulated overlapping contrails of optical depths 0.1 and 0.2 result in ~0.8 W/m$^2$ of radiative forcing, but two overlapping contrails of optical depth 0.15 result in a forcing of 1.1 W/m$^2$.

The altitude (temperature) of each layer also affects the effect that overlap has on the net contrail RF. Net attributable RF of a contrail-contrail system decreases as contrail altitude decreases (increasing temperature), due to the increase in the temperature of re-emission. For a cloud-contrail system, the contrail RF is most sensitive to the altitude (temperature) of the natural cloud. The absolute difference varies from +6.1 W/m$^2$, for warmer (lower-altitude) clouds, to -12 W/m$^2$, for cooler (higher) clouds, assuming an optical depth of 3 for the natural cloud layer (see Fig. S4 in the SI).

The radiative forcing attributable to contrails (as well as the effect of overlap) also varies as a function of local conditions, such as the outgoing longwave radiation (related to surface temperature), surface albedo, and solar zenith angle. The greatest contrail warming occurs for high values of outgoing (terrestrial) longwave radiation, and high surface albedos. This is due to the combination of increased longwave radiative forcing and the reduced shortwave cooling from the contrail. We also find that the net RF of the contrail-contrail system is reduced as the solar zenith angle increases. As $\theta$ increases from 0° to 75°, the maximum net RF (at maximum OLR$_{clear}$ and $\alpha$) decreases from 27 W/m$^2$ to 8.0 W/m$^2$. This effect, driven by changes in the shortwave cooling, is explored in more detail in Section C3. The relative effect of overlap on both the warming and cooling components of contrail RF is, in relative terms, insensitive to outgoing longwave radiation and albedo. Due to the low absolute values of |RF$_{SW}$| at maximum $\alpha$ and

high values of $|RF_{LW}|$ at maximum $OLR_{clear}$, maximum absolute net RF decrease happens in those areas. For a deeper analysis, Figure S5 in the SI shows the variation of net RF in a contrail-contrail overlap event, with $OLR_{clear}$ and $\alpha$.

## 4.1.2 Parametric analysis of contrail-contrail overlap specific impact on $RF_{SW}$ and $RF_{LW}$

We now evaluate the error in both $RF_{SW}$ and $RF_{LW}$ which results from ignoring the effect of contrail-contrail overlap. We use $RF_O$ to denote the RF when overlap is treated explicitly and $RF_I$ to denote when overlap is ignored ("independent"), in which case the total RF is the sum of the RF from each cloud layer. The relative change in the estimated RF impact of the system is then

$$D = \frac{(RF_I - RF_O)}{|RF_O|} \tag{8}$$

where a positive value of D indicates that the assumption of independence results in an overestimate of warming effects (LW) or an underestimate of cooling effects (SW). Equivalently, a positive value means that accounting for overlap results in a decrease in the RF of the system relative to the independent calculation.

Figure 3 shows the percentage bias resulting from ignoring overlap when quantifying the RF of a contrail-contrail system. This is quantified as a function of each contrail's optical depth and of the local solar zenith angle ($\theta$). In each case, the upper and lower contrail have identical physical properties, as described in Section 4.1.1. We find that accounting for overlap consistently results in a reduced longwave RF for two overlapping contrail layers. This means that, if overlapping contrails are considered as independent, their longwave RF is overestimated by up to 16% (for contrails with optical depth of 0.5). This effect is independent of the solar zenith angle.

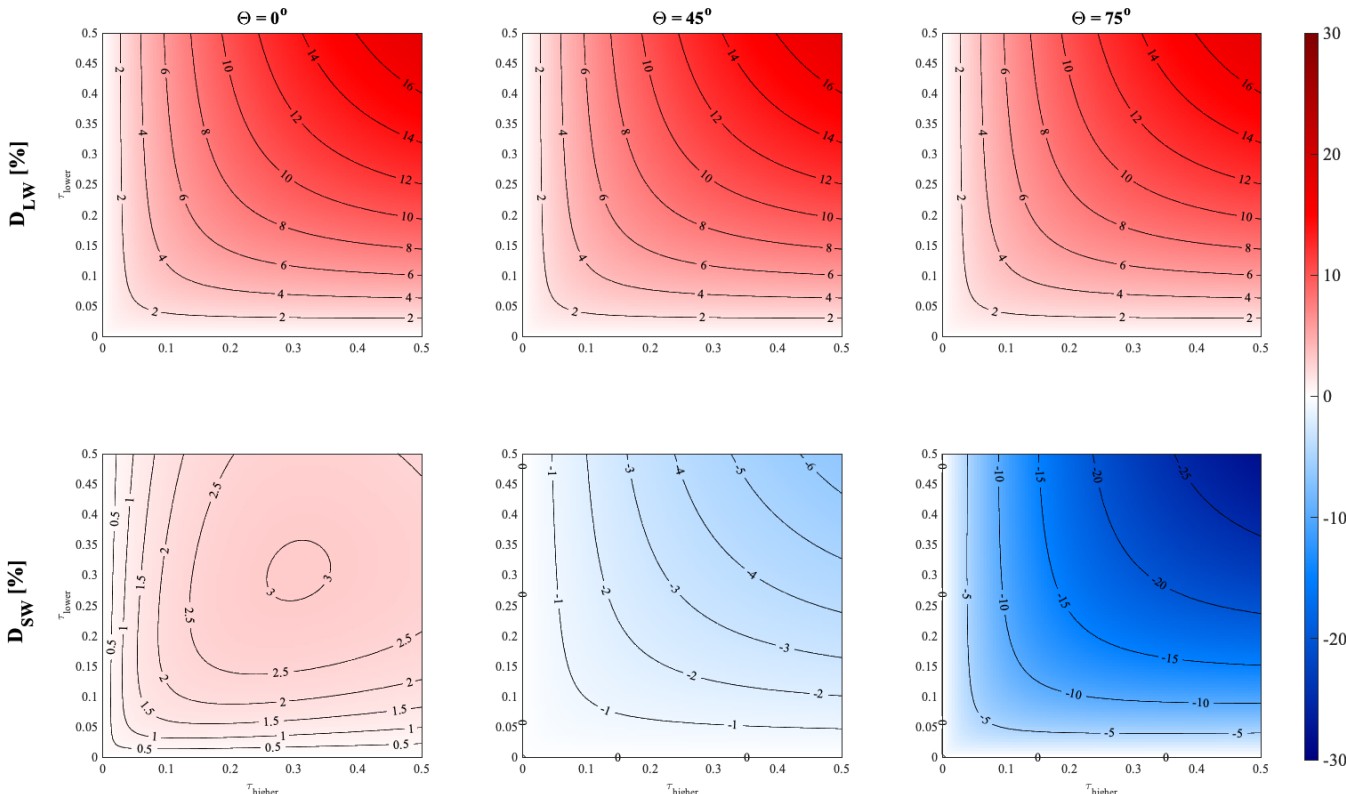

**Figure 3.** Error in estimated RF for two overlapping contrails when ignoring overlap, as a function of τ and θ. The solar zenith angle increases from the left-most to right-most panels. The upper panels show longwave RF error, while the lower panels show shortwave RF error. Positive (red) values indicate that the independent assumption results in an overestimate of warming effects (or underestimate of cooling effects). Negative (blue) values indicate that the independent assumption results in an overestimate of cooling effects (or underestimate of warming effects). An additional version of this figure, including calculations using a solar zenith angle θ = 30° and covering a larger range of contrail optical depths, is provided as Figure S6 for comparison to other literature.

For shortwave RF, the error resulting from independent calculation is sensitive to the solar zenith angle. In most cases, the total shortwave ("cooling") RF is smaller in magnitude when correctly accounting for overlap, relative to the independent calculation. This corresponds to an overestimate of the total reflectance if contrails are treated as independent. The magnitude of this error generally increases with contrail optical depth. Near sunrise or sunset (θ ≈ 75°), accounting for overlap reduces the calculated cooling effect by 25% for τ = 0.5. However, we observe a change in the sign of the error at zenith angles below ~25°. At noon (θ = 0°), assuming independent effects results in a slight underestimate of the cooling effect for any optical depth between 0 and 0.5, up to a value of 3.2%. The cause for the change in sign at very low solar zenith angles is investigated in detail in Appendix D.

The effect on total net RF depends on the tradeoff between the effects on both $RF_{LW}$ and $RF_{SW}$. At low solar zenith angles, neglecting contrail-contrail overlaps results in an overestimation of net RF. Due to the changes in sign of the error for shortwave RF, and the fact that the magnitude of each of the two components varies based on different factors, the effect on net RF at high solar zenith angles will depend on factors such as the location, time, and properties of each contrail.


In summary, we find that the net radiative forcing due to contrails may include a significant non-linear term due to overlap which is not captured in existing models. For contrails with optical depths of up to 0.5, we find that failing to account for this non-linearity could result in an overestimate of both the longwave warming (up to 16%) and the shortwave cooling (up to 25%). The sign and magnitude of the

effect on the system net RF is highly dependent on layers' properties, local conditions, and the solar zenith angle. The total effect of overlapping on a single contrail is therefore dependent on the solar zenith angle (time), temperature (altitude), and geographic location in which the contrail is formed.

**4.1.3 Parametric analysis of radiative impact from a contrail located in-between cirrus clouds**

We also model the case of a single contrail located between two natural cirrus cloud layers. We simulate a single contrail with the same properties as were used in the previous section (temperature of 215 K, optical depth of 0.3, and asymmetry parameter of 0.77). This is bracketed by two cirrus clouds, 500 m above and below the contrail, with optical depths of up to 1.5 and an asymmetry parameter of 0.75 (Kokhanovsky, 2004).


Figure 4 shows how the single contrail RF varies as a function of the optical depth of both natural cirrus clouds and as a function of solar zenith angle. For reference, the estimated RF for the contrail at a solar zenith angle of 45° in the absence of clouds is +27.9 W/m$^2$ (longwave) and -26.9 W/m$^2$ (shortwave), resulting in a net forcing of 1.0 W/m$^2$.

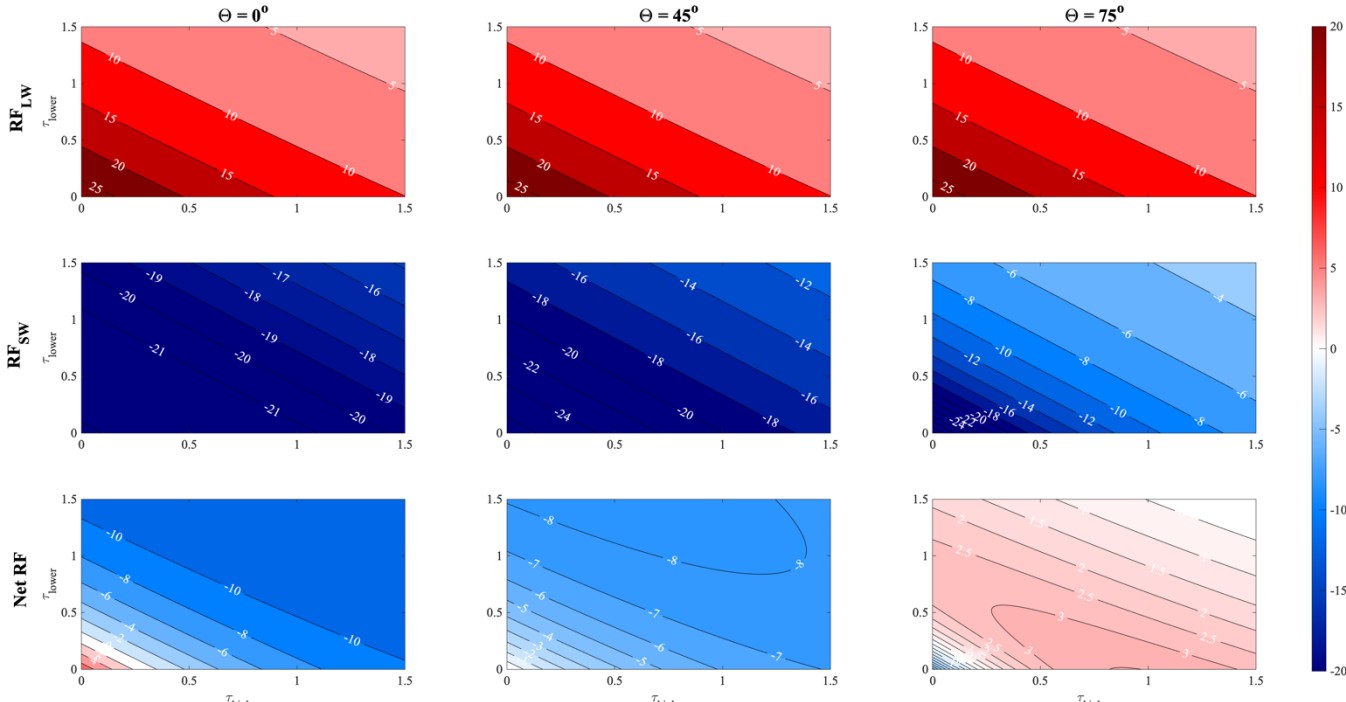


**Figure 4.** Radiative forcing [W/m²] due to a single contrail between two cirrus cloud layers. Radiative forcing is shown as a function of the solar zenith angle (increasing from left to right) and the optical depth of the lower (Y-axis) and upper (X-axis) natural cloud optical depths. From top to bottom: longwave; shortwave; and net radiative forcing. Contrail optical depth $\tau = 0.3$. An additional version of this figure, including calculations using a solar zenith angle $\theta = 30°$ and using a smaller contrail optical depth, is provided as Figure S7 for comparison to other literature.


The presence of either cloud layer alone decreases both the longwave and shortwave RF attributable to the contrail, as previously discussed. Except at high solar zenith angles, increasing the optical depth of either cloud layer reduces the net RF of the contrail layer. This is because the contrail's longwave RF falls rapidly, while the shortwave RF is less affected. The contrail's longwave radiative forcing decreases by up to a factor of seven when the surrounding clouds are sufficiently thick ($\tau = 1.5$), while the shortwave radiative forcing is only reduced by a factor of three. However, at high solar zenith angles, this situation is reversed (see Figure C2 in Appendix C), this means that the contrail RF instead initially increases with increasing cloud thickness.


### 4.1.4 Comparison of the overlap model results to existing models

In addition to evaluating the model for the purposes of simulating a single contrail (see Appendix B), we also compare the model's estimates of the effect of two-layer overlap to estimates from an existing radiative transfer model - the previously-described Fu-Liou radiative transfer model (FL). FL uses solid hexagonal columns to represent ice clouds, which have previously been found to be best represented in the Corti and Peter model by assuming an asymmetry parameter $g = 0.87$ (Corti and Peter, 2009). Figure 5 shows the error resulting from considering overlapping contrails as if they were independent, for both longwave and shortwave components, in both models. All simulations are performed using identical radiation data (outgoing longwave radiation and land albedo) and contrail properties. More information is provided in Appendix B.

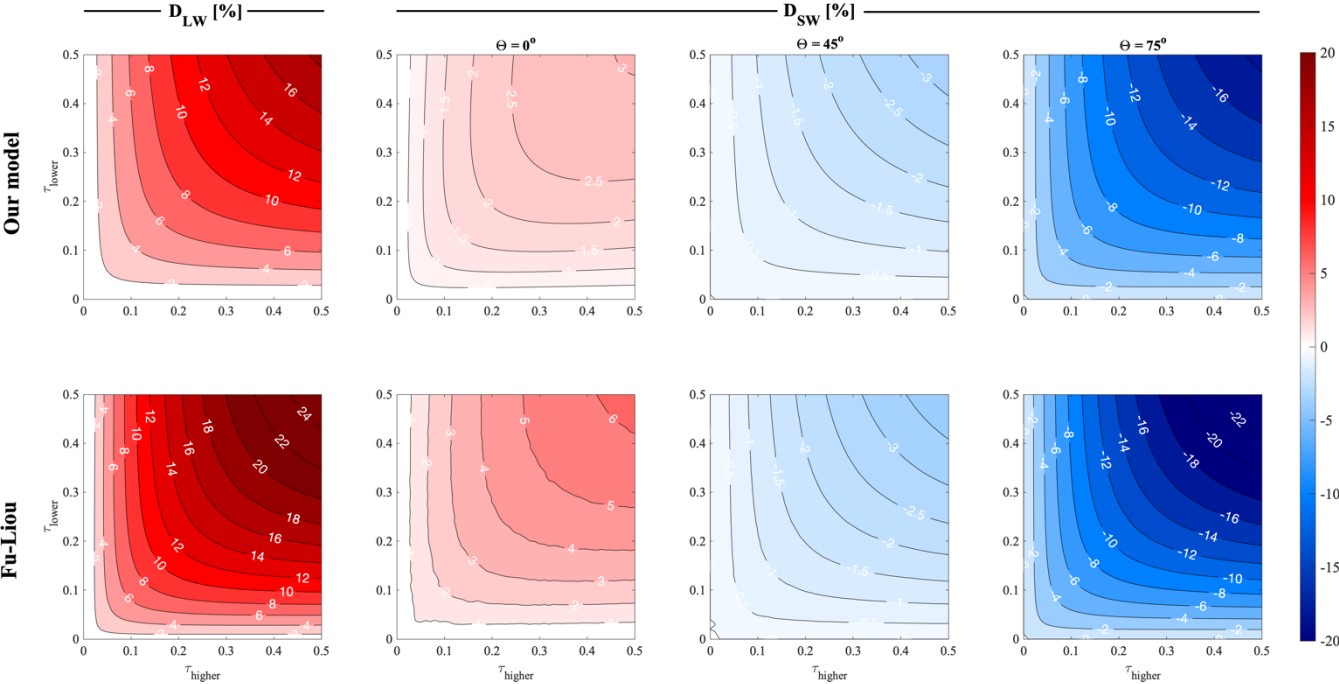

**Figure 5.** Error in estimated RF for two overlapping contrails when ignoring overlap, as a function of $\tau$ and $\theta$, for both our model (upper row of panels) and FL (lower row of panels). The first column shows error in longwave RF, while the remaining columns show error in shortwave RF at different solar zenith angles. Positive (red) values indicate that the independent assumption results in an overestimate of warming effects (or underestimate of cooling effects). Negative (blue) values indicate that the independent assumption results in an overestimate of cooling effects (or underestimate of warming effects). An additional version of this figure, including calculations using solar zenith angles $\theta = 15°$, $30°$ and $60°$ is provided as Figure S8 for comparison to other literature.

Qualitatively, the behavior is consistent between the two models. Both models estimate that the discrepancy in simulated longwave and shortwave RF (comparing the "overlap" to "independent" cases) increases with the increasing optical depth of each cloud layer. We also observe the same reversal of sign in the shortwave error at very low solar zenith angles. FL finds that both errors increase more quickly with optical depth than is estimated by our model, finding a maximum error in longwave RF of 25% (17%

in our model) and in shortwave RF of 24% (18% in our model). This indicates that our model correctly represents overlapping behavior but might underestimate the effect on both terms. The net RF difference is always lower than 30% and varies with solar zenith angle. At low solar zenith angles, we underestimate net RF (both for two independent and overlapping contrails). At $\theta = 45°$ we obtain the best agreement, with differences lower than 10% and at $\theta = 75°$ we overestimate net RF by up to 30% at an optical depth,

for both contrails, of 0.5 (at the upper end of current contrail optical depth estimates). These differences must be considered in the context of the global net RF results presented in Section 4.3.

## 4.2 Global sensitivity of cloud-contrail and contrail-contrail overlap to location and season

We next quantify the variation of contrail radiative forcing as a function of geographic location and time

of year. This captures the primary drivers in variations regarding the effects of overlap, as identified previously. As stated earlier, this work provides an upper bound for the effects of overlap by assuming maximum overlap between layers.

To obtain these sensitivities, we run a global simulation using 2015 atmospheric data (including radiation

and natural cloudiness data as described in Sections A3 and 3.2.2) in which we simulate the presence of a contrail layer in each location across the globe. We here assume that, in each grid cell, 1% of the total area is covered by contrail, reproducing an analysis performed by Schumann et al. (2012). We evaluate the effect of both cloud-contrail and contrail-contrail overlaps on contrail RF. We also calculate the error which would be incurred by treating two overlapping contrails as independent.


Figure 6 shows the radiative forcing per unit of additional contrail optical depth at each location, under both "clear sky" and "all sky" conditions (without and with natural clouds respectively, for year-2015 natural cloud cover). The RF varies as a function of latitude, consistent with prior studies (Schumann et al., 2012). The longwave warming ($RF_{LW}$) is maximized in regions with higher surface temperatures such

as the equator. Cooling (negative $RF_{SW}$) is instead sensitive to surface albedo, being maximized over oceans and minimized over snow-covered or desert regions.

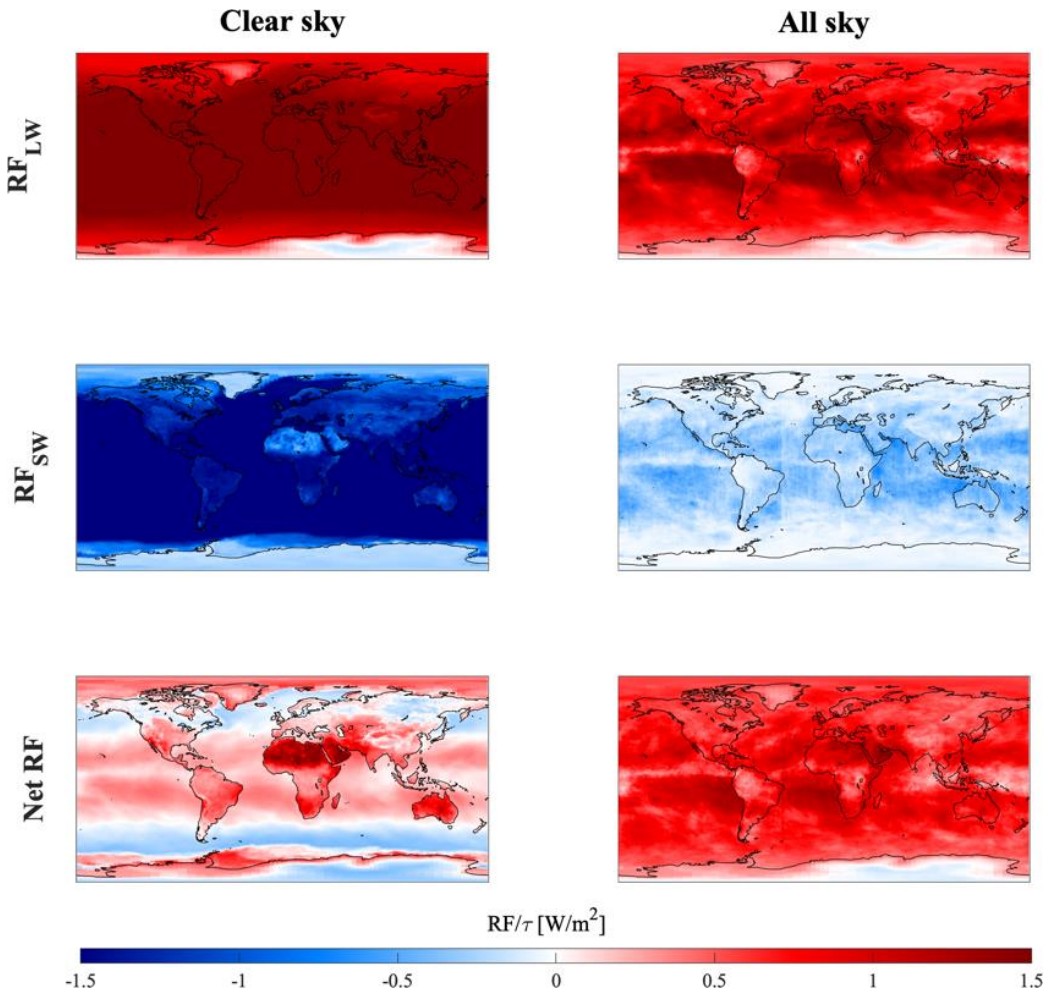

**Figure 6.** Hourly average radiative forcing per unit optical depth [W/m²] for a 1% contrail covering per 0.25°× 0.3125° cell and $g$=0.77
(2015 atmospheric data) From top to bottom: Longwave, shortwave, and net RF. Clear sky sensitivities are shown on the left, and all sky calculations on the right. Small discontinuities in shortwave cooling for all-sky conditions (e.g. over the North Atlantic Ocean) are the result of data artifacts in the CERES satellite data, which is a composite of observations from multiple observation platforms.

By comparing the "all-sky" and "clear-sky" simulation results, we find that the absolute value of both components of radiative forcing is reduced by the presence of clouds. The global mean reduction in shortwave forcing (~83%) exceeds the reduction in longwave forcing (~42%), meaning that cloud overlap causes a more than three times increase in the global, area-weighted average, contrail net RF, from +27.8 mW/m$^2$ to +107.1 mW/m$^2$ per unit of contrail optical depth. These values are consistent with prior studies 655 (e.g. Schumann et al. 2012). A detailed comparison with those prior studies can be found in the SI in Table S1, including for both clear-sky and all-sky conditions. We find that our estimated clear-sky RF results are consistent with literature results. Although our estimated all-sky net RF results are also consistent, we find that our estimated component (longwave and shortwave) RF results are smaller in magnitude. This is potentially due to our use of the maximum overlap assumption.


Our assumed asymmetry parameter for each contrail layer ($g = 0.77$) corresponds to a greater backscatter than is the case in previous studies (Fu and Liou, 1996; Myrhe et al., 2001; Schumann et al., 2012). This explains the low global sensitivity obtained in clear-sky conditions. For comparison, using an asymmetry parameter of $g = 0.9$ (typical of regular, spherical particles) results in a global mean, clear-sky sensitivity 665 of +144.3 mW/m$^2$, reducing cloud-contrail global impact. A deeper analysis of uncertainty related to microphysics and resulting global sensitivity to contrail is the subject of a complementary work (Sanz-Morère et al., 2020).

At night the effect of clouds on global contrail RF reverses, as the reduction in reflected shortwave 670 radiation is lost while the reduction in absorbed longwave radiation remains. The global, area-weighted average nighttime contrail RF is therefore reduced by 42% when accounting for the presence of clouds. However, these effects vary significantly with geographic location.

The depth, frequency, and altitude of natural cloud cover all vary as a function of location, resulting in a 675 geographical dependence of the sensitivity of contrail RF with respect to clouds. Thick, low altitude clouds are more common at midlatitudes, while higher, thinner cirrus clouds are more common in the tropics (Warren et al., 1988; Sassen et al., 2008; Marchand et al., 2010). The effect of these clouds on

contrail RF is shown in Figure 7. In the tropics (TROP, 30°S - 30°N), contrail RF is 1.5 times higher in the presence of clouds. However, in the midlatitudes (MLAT, 30°N - 60°N), the thicker, warmer clouds have a greater effect. Overlap with midlatitude clouds increases the net RF attributable to a contrail by more than a factor of six, from 8.7 mW/m$^2$ to 66 mW/m$^2$. This result is consistent with the analysis given in Section 4.1.1, and is due to the high reflectivity of the thick, low-altitude clouds.

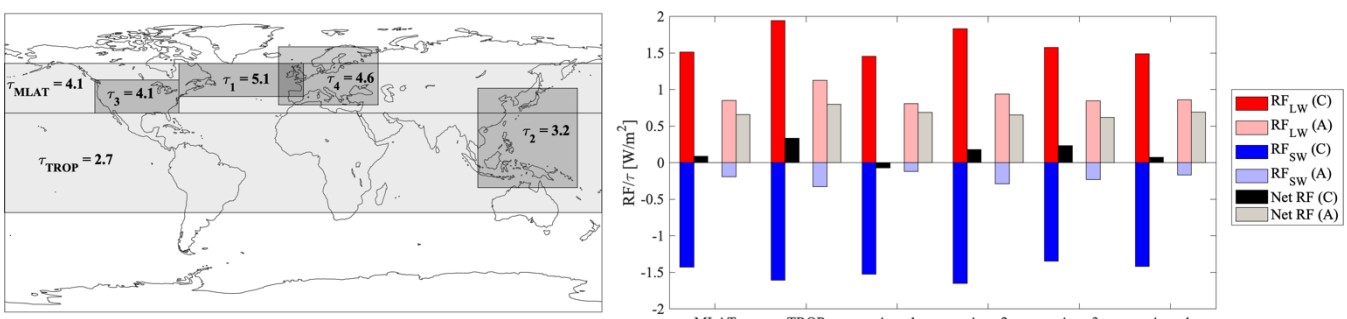

**Figure 7.** Contrail RF per unit of contrail optical depth for 6 different global areas: MLAT (northern midlatitudes), TROP (tropics), and subregions 1-4. Left panel: latitudinal and longitudinal limits and average natural cloud optical depth of each area. Right panel: average RF per unit of optical depth per area (A: all sky, C: clear sky).

We also quantify the sensitivity of contrail RF to overlap in four different geographical subregions: area 1, representing the North Atlantic corridor; area 2, which includes parts of Asia; area 3, approximately representing the continental United States; and area 4, approximately representing Europe (see Figure 7). These areas include ~51% of all passenger traffic in 2019 (Boeing, 2020) and differences in sensitivity for each region provide insights into the effects of future growth.

In all four regions, clouds have a greater relative and absolute effect on shortwave RF than on longwave RF (Figure 7). In area 3, clouds reduce the longwave RF per unit contrail optical depth by 46%, while reducing the shortwave RF by 83%. This results in 2.3 times increase in the net RF relative to the clear-sky case. By contrast, in the North Atlantic corridor (area 1), clouds reduce the longwave RF by 44%, but the shortwave RF is reduced by 99%. This changes a cooling effect of 70 mW/m$^2$ into a warming of 690 mW/m$^2$. The effects of cloud overlap in areas 2 and 4 lie in between these two extremes.

These variations are driven by differences in natural cloud coverage (primarily due to latitude) and surface albedo (e.g. land vs. sea). In the case of area 1, contrails are mostly forming over water, which has a very

low albedo. As a result, there is a larger shortwave cooling, and therefore a greater increase in the net RF when this cooling is mitigated by overlap with clouds. By contrast, over area 3 there is a greater land fraction and the clouds are thinner, resulting in a smaller overlap effect. These results suggest that avoiding overlap of contrails with clouds will yield the greatest RF reduction on midlatitude, oceanic routes, whereas the advantages of doing so over land and/or at lower latitudes will be smaller.

Contrail RF, and its sensitivity to clouds, also varies by season. Under all-sky conditions, in the Northern Hemisphere, the net contrail sensitivity is globally 15% lower in local winter than in local summer. This is because the reduction in longwave RF due to cooler surface temperatures exceeds the reduction in shortwave RF from shorter days (less insolation). However, this varies significantly by latitude because of the effect of changes in day length.

Climate change is likely to affect these results due to its effects on global cloud cover (Norris et al., 2016). Current satellite data show that cloud top heights are gradually increasing, which will likely decrease contrail net RF due to the resulting decrease in cloud top temperature. It is also anticipated that the tropics will expand (Kim et al., 2017). This will mean that more contrails are overlapping with high-altitude clouds, resulting in a reduced sensitivity to cloud overlap as discussed earlier.

We also evaluate how the effect of contrail-contrail overlap on contrail RF varies by location. This is quantified by simulating two contrail layers at each location, first treating them as independent and then calculating the total RF when accounting for overlap. The layers are simulated as being separated by 500 m. We find that correctly accounting for overlap results in a decrease in both the cooling and warming effects, relative to the "independent" calculation. The percentage decrease in each component is approximately uniform across all locations (consistent with Section 4.1.1). Since the components are of opposite sign, this results in a non-uniform effect on total net RF. Contrail overlap has the greatest effect on the net RF when contrails are located in hot, equatorial areas (increased longwave RF) with high albedo (reduced negative shortwave RF), as is the case in low-latitude desert areas such as the Sahara. This results in a maximum contrail net RF reduction by contrail-contrail overlapping in the tropics (TROP), where

we find a reduction from an average sensitivity of 1.6 W/m$^2$ (per unit of optical depth) for two "independent layers" to an average sensitivity of 0.6 W/m$^2$ for two "overlapping layers". Global sensitivity maps to contrail-contrail overlap are shown in Figure S9 of the SI.

## 4.3 Effect of cloud-contrail and contrail-contrail overlaps on net 2015 global radiative forcing attributable to contrails

Finally, we quantify the net effect of cloud-contrail and contrail-contrail overlap for existing aircraft traffic patterns. We use year-2015 contrail coverage data as estimated using CERM (see Section A3.1). The RF impacts of contrails are presented in Table 5, under all-sky and clear-sky conditions, and with and without explicit treatment of contrail-contrail overlap. For the given estimate of contrail coverage and optical depth, our assumption of maximum overlap means that these results provide an upper bound on the magnitude of the effect due to overlap (see Section 3.2.1).

### 4.3.1 Cloud-contrail overlaps

For 2015, we find that approximately 75% (by area) of contrails overlap with mid-level clouds. We compare results calculated under all-sky and clear-sky conditions (scenarios OA and OC) to quantify the effect of cloud-contrail overlap on contrail RF.

Figure 8 shows the effect of cloud-contrail overlaps on the shortwave and longwave radiative forcing due to contrails. We find a 66% decrease in net global cooling attributable to contrails as a result of cloud cover, accompanied by a 37% decrease in warming. Accounting for cloud overlaps therefore results in more than ten times greater net contrail warming. As a consequence, the annual average, global net RF changes from +0.7 mW/m$^2$ under clear sky conditions to +9.7 mW/m$^2$ when including clouds ("all-sky"). Overlap with clouds is found to reduce the global longwave RF of contrails by 37%, and the shortwave RF by 66%. At night, contrails over natural clouds have a lower net RF due to the lack of any shortwave effect. As a result, the presence of natural clouds during nighttime reduces the net RF of contrails by 37% as the only effect that clouds can have at this time is to mitigate the contrail longwave RF.

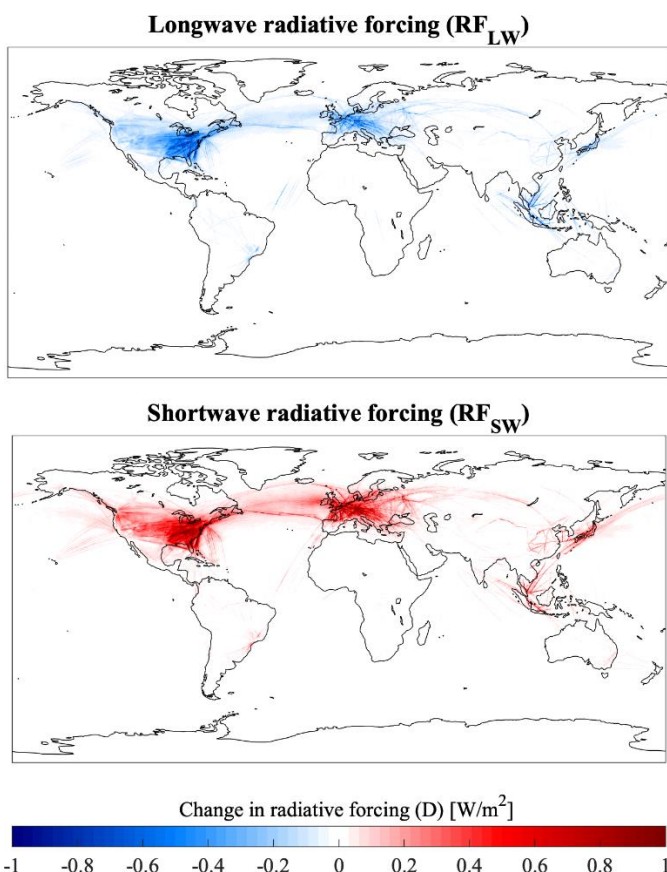

**Figure 8.** Change in annual-average RF [W/m$^2$] due to the presence of clouds from global flights in 2015. Upper panel: Longwave RF (blue corresponds to negative, meaning that clouds reduce the warming effect of contrails). Lower panel: Shortwave RF (red corresponds to positive, meaning that clouds reduce cooling effect of contrails).

### 4.3.2 Contrail-contrail overlaps

An analysis of year-2015 global contrail coverage simulated at a resolution of 0.25°×0.3125° using the CERM modeling tool (Caiazzo et al., 2017) provides an estimate of overlap frequency. Assuming maximum overlap by area (i.e. all contrails in a given column overlap to the greatest possible extent - see Section 3.2.1), up to 15% of all contrail area includes overlap with other contrails (Fig. 9, lower plot). More details on this assumption and the CERM modeling tool are given in Section 3.2 and A3. The majority of this overlap occurs for contrails which are no longer line-shaped, and which may appear to be natural cirrus when viewed from the ground. If we exclude contrails which are more than an hour old or which are "subvisible" for human eye, having an optical depth below 0.03 (Kärcher, 2002; Kärcher, 2018), this fraction falls to 2.2%.

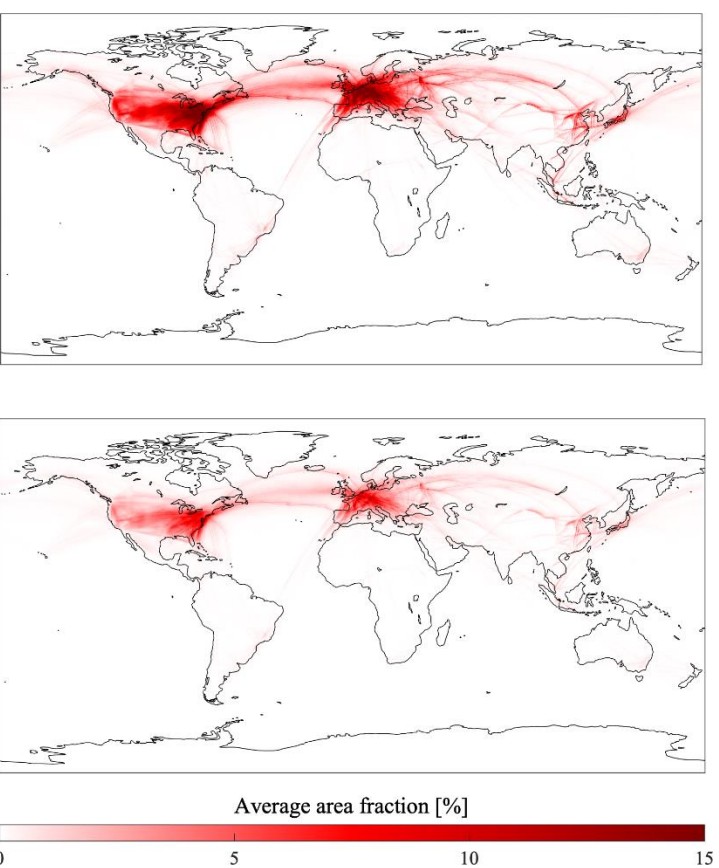

**Figure 9.** Estimated annual mean global contrail coverage for 2015. Upper panel: yearly average contrail coverage (in %), assuming no contrail-contrail overlap. Lower panel: yearly average coverage (in %), assuming "maximum overlap" such that all contrails in a single column are centered in each 0.25°×0.3125° grid cell (in %). Contrail data were generated using the CERM global contrail modeling tool (Caiazzo et al., 2017), which provides contrail quantities and properties discretized to the aforementioned global grid. More information on CERM can be found in Appendix A3. Maximum contrail overlap assumes that all contrails in a single vertical grid column overlap to the greatest possible extent by area. This estimate includes contrails which are diffuse and/or "sub-visible" (optical depth < 0.03).

Under an upper-bound assumption for the total area of contrail overlaps, we find that 15% of all modeled contrail area overlaps with other contrails at different altitudes. If the effect of cloud-contrail overlap is ignored, the maximum contrail-contrail overlap results in a more than three times increase in the contrail net radiative forcing. This is made up of a 21% reduction in longwave warming but a 38% decrease in shortwave cooling. However, if cloud-contrail overlap is accounted for, the net impact of contrail-contrail overlap is instead a 3.0% reduction in net contrail RF. The reduction in longwave warming is 2.0%, exceeding the 1.8% reduction in shortwave forcing. This difference is due to the strong mitigation of shortwave forcing (approximately 1/3 of that under clear sky conditions) by existing clouds, and is

consistent with the global sensitivity to contrail-contrail overlaps demonstrated in Section 4.2. The majority of contrail-contrail overlap occurs in low-albedo areas such as the North-Atlantic corridor (area 1 of Figure 7) or at high latitudes (areas 3 and 4 of Figure 7), resulting in a small absolute effect on net RF (-0.3 mW/m$^2$). Another contributing factor may be regional variations in the fraction of contrail coverage which results in a longer-term increase in overall cloud coverage. Bock and Burkhardt (2016) found that this fraction varies significantly, and is around half the global average in the Northern Atlantic and Northern Pacific.

These results are sensitive to the assumptions regarding the degree of overlap in each model column. We assume that all contrails in a given model column overlap to the maximum extent, providing an upper bound for the total effect of contrail overlap. If we instead assume minimum overlap – where each contrail in the column "avoids" overlap until there is no remaining uncovered area – then contrail-contrail overlap only occurs for 2% of all modeled contrail area. This limitation is explored further in Section 5.

### 4.3.3 Overall impact of cloud-contrail and contrail-contrail overlap on global RF

**Table 5.** Contrail global average radiative forcing (daytime value) in mW/m$^2$ under each set of assumptions (IC: independent contrails; clear-sky OC: overlapping contrails clear-sky; IA: independent contrails all-sky; OA: overlapping contrails all-sky).

|          | IC     | OC     | IA     | OA     |
|----------|--------|--------|--------|--------|
| **RF$_{LW}$** | +33.3  | +32.6  | +21.0  | +20.6  |
| **RF$_{SW}$** | -33.1  | -31.9  | -11.0  | -10.8  |
| **Net RF** | +0.2   | +0.7   | +10.0  | **+9.7**  |

Table 5 shows the total contrail RF with and without clouds, and either accounting for or neglecting the effects of contrail-contrail overlap. We find that contrails induce a net RF of 9.7 mW/m$^2$ for 2015. This result includes a 3% reduction in overall RF from contrail-contrail overlap, but most of it (93%) is due to overlap with clouds.

Assuming that these impacts are an upper bound, these results suggest that the impacts of cloud-contrail overlap are significant, but that contrail-contrail overlap can likely be neglected in radiation modeling studies under current conditions. However, our result of +9.7 mW/m$^2$ for the net impact of contrails is at

the low end of existing literature estimates (see Table 1). This is due to uncertainties in contrail coverage, contrail optical depth, and contrail optical properties. The global CERM simulation output has an average

optical depth per contrail of 0.065 and a global coverage of 0.39% by area, both of which are at the lower end of literature estimates (see Table 1). As a sensitivity test, if we increase the optical depth of all contrails from the CERM output data by a factor of four to give the same average per-contrail optical depth as Schumann et al. (2013), which found a net RF of 49.2 mW/m$^2$, we find a global net contrail RF of 32.6 mW/m$^2$. Under these conditions, we find that contrail-contrail overlaps decrease the simulated

global RF by 8%.

## 5 Limitations

### 5.1 Radiative transfer model

Our radiative forcing model is an extension to an existing, single layer cloud model (Corti and Peter, 2009). Corti and Peter's model was previously compared to the widely used radiative transfer library libRadtran (Mayer and Killing, 2005) for single contrail radiative forcing (Schumann et al., 2012). Appendix A additionally provides an independent comparison of its performance against a set of existing radiative transfer models for the purposes of simulating single contrails, and Section 4.1.4 performs a

comparison for simulating multi-layer overlap. Based on the results of these comparisons, we here describe some of the limitations of this model, our estimate of their effect and importance, and possible opportunities for future improvements.

When calculating the total outgoing longwave radiation for each layer, the model includes an estimate of

absorption by atmospheric $CO_2$ and water vapor. Estimates for multiple overlapping layers may therefore double-count this contribution. Additionally, cloud emissivity is estimated as only a function of the cloud optical depth. This expression has been previously used as a parameterization of cloud longwave radiative transfer (Stephens et al., 1990), but it is unclear how this will affect estimates of the effects of overlap on contrail RF. Our model also neglects scattering of longwave radiation, based on longwave radiative

transfer formulations from Stephens et al. (1990) and Corti and Peter (2009). This effect has been ignored

in several climate models, and previous studies have estimated the error resulting from this assumption in the context of natural clouds (Ritter and Geleyn, 1992; Stephens et al., 2001; Costa and Shine, 2006). They obtain a global underestimation of OLR of between 3 and 8 W/m$^2$, leading to a potential underestimate of cloud RF$_{LW}$ of approximately 10% (Costa and Shine, 2006). These limitations may partially explain some of the differences in the calculated outgoing longwave radiative forcing between this model and the Fu-Liou radiative transfer model, which includes longwave scattering (Fu et al., 1997; Gu, 2019), as discussed in Section 4.1.4. Implementation of longwave scattering is therefore a potential avenue of future research, based on existing parameterizations (Chou et al., 1999; Tang et al., 2018). Our longwave radiative forcing model also assumes all layers to be in equilibrium, and does not account for local temperature feedbacks due to the presence of artificial cloud layers. Finally, we do not account for 3-D effects. Cloud layers are assumed to be vertically homogeneous and edge effects are ignored, as in the reference model. A previous investigation of contrail radiative forcing found that 3-D effects could change simulated radiative forcing by ~10% (Gounou and Hogan, 2007).

Regarding shortwave radiative forcing, we do not account for inhomogeneity in the above-cloud atmospheric transmittance of shortwave radiation, instead considering it to be constant at 73%. Shortwave radiative interactions between contrails and other constituents (such as tropospheric aerosols and water vapor) are also not explicitly accounted for. The model also uses an isotropic wavelength-independent two-stream approximation of radiative transfer (Coakley and Chylek, 1975). This has been shown to give accurate results (errors of approximately less than 15% in estimated SW reflectance) at optical depths below ~1 and solar zenith angles below 75°. Errors are expected to be larger outside of this range, as shown by comparison to other models (Appendix B). It is difficult to provide a quantitative estimate of the effect that such errors might have on the overall results, including the weaker dependence of our model's calculated RF and overlap impacts on solar zenith angle when compared to the FL model. However, we find that our model estimates a smaller RF than the FL model at low solar zenith angles. Annually, the solar zenith angle is between 75 and 90° for 16% of the time globally, and 14.5% of the time at latitudes covering the majority of current commercial flights (30°N - 60°N). This may therefore result in an underestimate of overall contrail RF by our model.

The two-stream approximation used in this model is most accurate for low optical depths. This is appropriate for contrails and thin natural cirrus, but lower-altitude natural clouds can be much thicker. For this reason, we use an asymmetry parameter for high altitude clouds and contrails based on direct observations (Sanz-Morère et al., 2020), while using an asymmetry parameter similar to that suggested by Corti and Peter (2009) for low altitude clouds.

An additional concern is discussed by Rap et al. (2010). They showed that a correlation exists between the existence of contrails and natural clouds. This could result in bias when the method used to simulate or estimate natural cloud cover is not consistent with that used for contrail estimation. This is a difficult issue to address for a Lagrangian approach such as ours, and may result in an unquantified bias in our
estimated contrail radiative forcing. Future research using the model presented here may therefore wish to perform additional model comparison or calibration to ensure that colocation of contrails and natural clouds is correctly captured.

### 5.2 Input data

Due to the lack of additional input information, and to provide a conservative estimate, we assume that all contrails overlap maximally within a column. This assumption would not be necessary if additional information was supplied by the base contrail model. For instance, the Lagrangian mentioned model CoCiP (Schumann, 2012) includes additional information on contrail location and orientation that could
be used to improve overlap modeling. Currently, we instead assume maximum possible overlap. This provides an upper bound on the impact of multiple cloud layer overlap on contrail RF – which is significant since we find only a small effect due to contrail-contrail overlap. However, a more accurate assessment would be possible using the aforementioned orientation data.

Additionally, contrail coverage could be constrained or calibrated by satellite measurements. Some studies (Kärcher et al., 2009; Iwabuchi et al., 2012) have combined satellite imagery (e.g. from MODIS)

with observed cloud coverage data to provide an improved estimate of contrail coverage. The combination of these data with single-contrail modeling tools (such as CERM) may help to improve the accuracy of estimated contrail coverage. However, there remain significant uncertainties due to the non-detection of very thin contrails (Kärcher et al., 2009), as well as the difficulty of distinguishing between long-lived contrails and natural cirrus clouds in observational data.

Finally, the natural cloud data provided by CERES is coarsely resolved with only four layers in the vertical dimension, averages every three hours, and lacking some additional useful information. The vertical resolution of CERES is also a challenge. Hogan and Illingworth (2000) found that (for cloud layers more than 4 km apart) overlap is essentially random, but this information is difficult to incorporate given the low vertical resolution of the CERES product. Alternatives to CERES like CALIPSO or CloudSat (Iwabuchi et al., 2012; Tesche et al., 2016) may provide a useful alternative, as they include both more precise estimates of cloud altitude and additional optical properties of the cloud layers.

These results are also sensitive to the optical depth of the simulated layers. Contrails simulated by CERM have a mean contrail optical depth of 0.065, at the lower end of a significant uncertainty range based on existing literature (see Table 1). Since the effects of overlap increase non-linearly with optical depth, estimates based on models which predict thicker contrails may find a significantly greater impact of overlap. Finally, there remain significant uncertainties in contrail coverage. The usage of reanalysis data (from GEOS-FP) as a meteorological data source has been found to overestimate humidity (Jiang et al., 2015; Davis et al., 2017), likely resulting in an overestimate in contrail coverage and lifetime. Improved estimates of contrail lifetime and formation frequency could significantly affect the frequency, and therefore total impact on contrail-related RF, of cloud-contrail and contrail-contrail overlap.

### 5.3 Priorities for future work

In light of the limitations outlined above, there are some future research directions which could significantly improve the accuracy of the results from this approach.

Firstly, a more detailed dataset of contrail coverage, including continuous information on contrail position and orientation, would remove the need to assume maximum overlap with natural clouds. Greater model spatial and temporal resolution, using real flight movement data, would reduce or even eliminate the need for a fractional cloud cover scheme.

Secondly, multiple improvements can be made with regards to the simulation of natural clouds. Finer vertical and temporal resolution would enable better representation of both natural and artificial cloud overlap. Our results are also sensitive to the properties prescribed for natural clouds. Incorporation of natural cloud datasets which estimate or infer cloud properties on a case-by-case basis would be useful in

providing a more accurate estimate of the effects of multi-layer overlap.

Finally, the radiative forcing model used here does not account for 3-D effects. A more accurate estimate of overlap impacts, in particular those associated with contrail-contrail overlap, would benefit from incorporating these details into their calculations. This is especially true for shortwave interactions.

## 6 Conclusions

We develop and apply a radiative transfer model to estimate the effect of cloud-contrail and contrail-contrail overlap on the net radiative forcing from contrails. The results will improve our understanding of the factors which contribute to global contrail RF, and help existing models such as CERM to produce

more accurate estimates (Caiazzo et al, 2017).

We find that overlap between contrails and natural cloud layers can cause a non-linearity in the net radiative forcing. In most cases, overlap between a contrail and a second cloud layer reduces both the cooling (negative shortwave RF) and warming (positive longwave RF) effects of the contrail. This effect

is sensitive to the optical depth of each cloud layer. We find a net increase in radiative forcing when contrails overlap with thick clouds ($\tau > 0.5$), but a net decrease when contrails overlap with thinner clouds.

However, overlap between two contrails is in general beneficial for climate, decreasing the total contrail RF. The magnitude of this effect is sensitive to local conditions, including surface albedo, solar zenith angle, and surface temperature. Under night-time conditions, overlapping between contrails and any other cloud layer consistently reduces the net contrail RF due to the lack of competing shortwave effects.

The radiative forcing attributable to a contrail layer increases by a factor of three due to the presence of natural clouds on a global mean basis, but this varies by region. Clouds have a greater effect on midlatitude contrail radiative effects than in the tropics due to the general trend of greater thickness and lower altitude, while other parameters like atmospheric composition and incoming solar radiation may also play a role. They also have greater effects over oceanic routes. We find that contrails over the North Atlantic corridor have, on average, a small cooling effect under clear-sky conditions (-0.07 W/m$^2$ per unit of optical depth) but cause warming (+0.69 W/m$^2$ per unit of optical depth) in cloudy conditions. This suggests that avoiding cloud-contrail overlaps in this region could yield climate benefits, although implementing a contrail avoidance strategy is itself a non-trivial task (e.g. Teoh et al., 2020). This sensitivity also varies by season, with a 15% decrease in RF per unit of optical depth in the Northern Hemisphere from summer to winter.

For year-2015 atmospheric data and flight activity, we calculate an upper bound for the effect of multiple layer overlap on contrail radiative forcing. We find that the presence of natural clouds reduces global contrail longwave radiative forcing by 37%, and the shortwave radiation reflectance by 66%. This is found to result in a net increase in global contrail RF. Global contrail net RF potential instead decreases by 3% when accounting for contrail-contrail overlap. However, the magnitude of this effect is dependent on the optical thickness of the contrails, which remains highly uncertain (global estimations of average contrail optical depth can vary from ~0.065 to ~0.3).

**Acknowledgments**

The GEOS-FP meteorological data used in this study have been provided by the Global Modeling and Assimilation Office (GMAO) at the NASA Goddard Space Flight Center. This work was also partially

funded by two separate grants: from the "la Caixa" Foundation through their Full Graduate Fellowship,
and through NASA grant NNX14AT22A. Global coastline data are provided as part of MATLAB'S
Mapping Toolbox, and are derived from the Advanced High Resolution Radiometer sensor on the NOAA
Polar Orbiter satellites, NOAA-7, -9 and -11

## Code and Data availability

The codes and data used to produce this work are available from the authors upon request.

## Author contributions

SB, RS, and SDE were responsible for the conceptual design of the study. FA, RS, and SDE provided
continuous review of the study progress and direction during execution. ISM performed all model
simulations and wrote the manuscript. All authors contributed to manuscript editing.

## Competing interests

The authors declare no competing interests.

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

## Appendix A: Extended description of the radiative transfer model

### A1 Summary of the model for a single contrail

The radiative forcing model quantifies the instantaneous RF per unit area of cloud layer. A full description is given in the original model description paper (Corti and Peter, 2009), but we give a brief summary here.

In the original model, the longwave RF is calculated in $W/m^2$ for a single cloud layer as

$$RF_{LW} = L - L_c = \varepsilon\sigma^*(T_{\mathrm{srf}}^{k*} - T_c^{k*})$$
(A1)

where $L$ is the outgoing longwave radiation from the surface of the Earth (in $W/m^2$); $L_c$ is the total outgoing longwave radiation from the cloud (in $W/m^2$); $T_{\mathrm{srf}}$ is the temperature of the Earth's surface (in K); $T_c$ is the cloud temperature (in K); $\varepsilon$ is the contrail emissivity; and $\sigma^*$ (the adjusted Stefan-Boltzmann

 constant, in $W/m^2/K^{-2.528}$) and $k^*$ ($= 2.528$) are constants and based on clear sky simulations combining results from a high-fidelity radiative transfer model and ECMWF ERA-40 atmospheric profiles (Fu and Liou, 1993; Corti and Peter, 2009). Therefore $\varepsilon\sigma^*T_c^{k*}$ represents the longwave radiation emitted by the cloud (in $W/m^2$) accounting for $CO_2$ and water vapor absorption from the atmosphere (Corti and Peter, 2009). This model includes various assumptions. The double-counting of atmospheric absorption is

 inherent to the original model (see Section 5.1). Additionally, longwave emissivity is assumed to be only function of cloud optical depth (Corti and Peter, 2009; Stephens et al., 1990). However, Corti and Peter (2009) report that a 10% change in this function increases longwave radiative forcing error in comparison with radiative transfer calculations only by about 1%, indicating that the assumption can be retained in our model. Finally, the reference model neglects longwave radiation scattering based on assumptions

 from Stephens et al. (1990). This is further commented in the limitations section (Section 5).

The shortwave RF adapted model uses an isotropic wavelength-independent two-stream approximation of radiative transfer (Coakley and Chylek, 1975). $RF_{SW}$ is therefore calculated as

$$RF_{SW} = -S \cdot t \cdot (1 - \alpha)\left(\frac{R_c - \alpha R_c'}{1 - \alpha R_c'}\right)$$
(A2)

where $S$ is the incident solar radiation (in W/m$^2$); $R_c$ is the cloud reflectance for direct radiation; $R_c'$ is the cloud reflectance for diffuse radiation; $\alpha$ is the albedo of the Earth; and $t$ is the atmospheric transmittance above the cloud, assumed constant at a value of 0.73 (Corti and Peter, 2009). The daily mean atmospheric transmittance ($t$) is based on clear sky simulations combining results from a high-fidelity radiative transfer model and ECMWF ERA-40 atmospheric profiles (Fu and Liou, 1993; Corti and Peter, 2009). Assuming

a constant transmittance may result in some bias, as the parameter $t$ would likely vary with location, time and atmospheric composition, including column concentrations of water vapor and aerosols (Schwarz et al. 2020). Additionally, potential uncertainties resulting from the two-stream approximation can be found in Section 5.

While most of the parameters previously mentioned describe the atmospheric conditions, three parameters describe the interaction between clouds and radiation: longwave emissivity ($\varepsilon$) and shortwave reflectances ($R_c$ and $R_c'$). All three are dependent on the layer optical depth $\tau$. Shortwave reflectances, representing cloud interaction with sunlight, are additionally dependent on cloud layer microphysics through the asymmetry parameter $g$ and $R_c'$ is additionally dependent on the solar zenith angle. A full description of

this derivation is given in Corti and Peter (2009).

    The optical properties of contrail ice crystals are represented in the model by the asymmetry parameter $g$ of the layer. $g$ measures the degree of anisotropy of scattering and is dependent on the radius and shape of the particle mixture. It ranges from -1 (total backscatter) to +1 (total forward scatter), while equaling

0 for perfect isotropic scattering (Stephens et al., 1990). Ice cloud particles have complex scattering phase functions (Liou et al., 1998; Baran, 2012) but typically fall into the Mie scattering regime with a dominant forward scattering peak, corresponding to an asymmetry parameter between 0.7 and 0.9 (Baran, 2012; Nousiainen and McFarquhar, 2004; Yang et al., 2003). The effect of uncertainty in the asymmetry parameter on contrail RF is investigated in a complementary study (Sanz-Morère et al., 2020). We here

assume an average contrail asymmetry parameter, based on in situ measurements, of 0.77 with an increase for the first hour to account for short-term changes in crystal shape ($g = 0.78$) (Febvre et al., 2009; Gayet

et al., 2012; Bedka et al., 2013; Minnis et al., 2013; Schumann et al., 2017; Sanz-Morère et al., 2020). For natural clouds, the asymmetry parameter is calculated as a function of altitude only. We assume that clouds below 8 km have an asymmetry parameter of 0.85 (typical of liquid water clouds); that clouds above 10 km have an asymmetry parameter of 0.7 (typical of long-lived cold cirrus clouds); and that clouds between 8 and 9 km have an asymmetry parameter of 0.8 (Gerber, 2000; Jourdan, 2003; Kokhanovsky, 2004; Schumann et al., 2017).

## A2 Modification, limitations, and comparison of the radiative transfer model

We have modified the original approach described by Corti and Peter (2009) to account for limitations highlighted by Lolli et al. (2017). Firstly, the original model estimates outgoing longwave flux at the surface by applying a fixed relationship between surface temperature and emitted radiation, based on data from the ECMWF ERA-40 meteorological product. Lolli et al. (2017) found that, below surface temperatures of 288 K, this yielded results that agreed (within 6%) with those from the more complex Fu-Liou-Gu radiative transfer model (Fu and Liou, 1992, 1993; Fu et al., 1997; Gu, 2019). However, they also found that for surface temperatures greater than 288 K, this approach is inaccurate and results in radiative forcing errors of approximately 65%. They identified that the source of this error was the regression used by Corti and Peter (2009) to estimate longwave emissivity in the context of high surface temperatures.

To overcome this issue, we instead use a "top-down" approach in which radiative forcing (longwave) is calculated as the difference between the estimated top-of-atmosphere longwave flux under "clear sky" conditions (without clouds), and the longwave flux perturbed by cloud layer(s). This removes the need to use the regression from Corti and Peter (2009) when calculating surface emissivity, which is the most likely context in which such high temperatures will be encountered. The estimated outgoing longwave radiation in the absence of clouds ($OLR_{clear}$) is provided in the CERES data product (see Section A3.2). This value of outgoing longwave radiation is estimated to have an annual global mean error of approximately 1.7%, while local biases can reach values up to 10 W/m$^2$ (Loeb et al., 2018). However, we

do not propagate this error further through our calculations. Hence, we calculate longwave radiative forcing due to contrails as

$$RF_{LW} = \varepsilon OLR_{clear} - L_c = \varepsilon OLR_{clear} - \varepsilon \sigma^* T_c^{k*} \tag{A3}$$

with all other terms as described in Equation (A1). An alternative approach to overcome the errors in
estimated clear-sky radiation from satellite data is proposed by Schumann et al. (2012), by reducing the usage of satellite data in the radiative forcing model to top of the atmosphere irradiances.

Shortwave radiative forcing is calculated by assuming a constant atmospheric transmittance above the cloud layer, which may result in inaccuracy when considering clouds at different altitudes. This constant
value is calculated based on average estimates from a high-fidelity radiative transfer model (Fu and Liou, 1993). Lolli et al. (2017) found that the error due to this assumption was negligible, and so we retain it in our model.

We also assume that cloud layers are of sufficient horizontal extent that 3-D effects (due to horizontal
propagation of radiation) are negligible. The effect of this assumption has been investigated in detail previously (Gounou and Hogan, 2007; Davis and Marshak, 2010; Barker et al., 2012; Hogan and Shonk, 2013). Due to the low thickness of contrails, the resulting error in $RF_{SW}$ and $RF_{LW}$ is expected to be on the order of 10% (Gounou and Hogan, 2007).

To ensure that our conclusions are realistic, we also compare the model to two existing radiative transfer models developed for cirrus clouds: the "Fu-Liou" model (hereafter FL) (Fu and Liou, 1992, 1993; Fu, 1996; Fu et al., 1997) and CoCiP (Schumann, 2012). We calculate the radiative forcing due to an isolated contrail layer while varying multiple parameters: contrail optical depth, surface albedo, and solar zenith angle (with fixed radiation data). A full description and evaluation is given in Appendix B. Each of the
three models uses a different approach to represent the optical properties of the ice crystals, so initial comparisons are performed by comparing the results when sweeping over a range of input parameters.

We find that, for the radiative forcing due to a single contrail, our results match those from CoCiP, with differences of less than 10% for both $RF_{LW}$ and $RF_{SW}$. Qualitatively, for the same range of particle sizes, FL shows similar behavior. However, the magnitude of the calculated radiative forcing differs between our model and FL, with inconsistencies of up to 40% in $RF_{SW}$.

Due to the strong dependence of $RF_{SW}$ on crystal size and shape (Markowicz and Witek, 2011b; Sanz-Morère et al., 2020), and due to the different treatment of these properties in the three models tested, we conduct a deeper analysis on the resulting difference on $RF_{SW}$. We choose a specific crystal size in FL and compare the simulated RF against results from our model using an "equivalent" asymmetry parameter (more information can be found in Appendix B). For a given surface albedo, we find differences of less than 15% at low solar zenith angles, increasing up to 20% at solar zenith angles greater than 80°. This is consistent with prior evaluations of the two-stream approximation used in our model (Coakley and Chylek, 1975; Corti and Peter, 2009), which has reduced accuracy at high solar zenith angles (see Section 5.1). The dependence of $RF_{SW}$ on albedo is also evaluated in each model. Qualitatively the three models show the same behavior with changing albedo, optical depth and solar zenith angle. For albedos below 0.3 the models agree to within 10%, and for albedos below 0.5 the maximum difference is less than 30%. The percentage difference is insensitive to optical depth (see Fig. B2).

## A3 Input data required for single contrail RF calculation

### A3.1 CERM modeling tool

An hourly map of contrail optical depth, coverage and lifetime in 2015 is estimated using a global version of CERM (Caiazzo et al., 2017). CERM follows a bottom-up approach for simulating contrails by combining externally-provided meteorological and atmospheric data with flight track data.

With an hourly time-discretization, and a 0.25°×0.3125°×22 global grid, CERM estimates individual contrail properties (including optical depth and size) for all flights in a year using flight track and atmospheric composition data. CERM models contrails from formation to sublimation based on the physical evolution defined in Schumann (2012). Therefore, it is in theory capable of capturing linear

contrails and contrail cirrus. Two contrails allocated in the same grid cell are assumed to be a single contrail while no overlap is assumed between contrails located at different vertical levels. Additionally, physical interactions between simulated contrails and natural clouds are not considered by CERM. This is in part because the contrails may form in the "non-cloudy" parts of grid cells, and in part because of uncertainty over contrail formation when flying through (for example) subvisible cirrus. We use meteorological reanalysis data from the GEOS forward processing (GEOS-FP) product, supplied by the NASA Global Modeling and Assimilation Office. Flight track and emissions data are calculated using the open-source Aviation Emissions Inventory Code (AEIC) (Simone et al., 2013).

The CERM version used to create the input data for this analysis incorporates new capabilities compared to previous versions (Caiazzo et al., 2017): a higher-resolution vertical grid (22 layers instead of 10 layers); a $4^{th}$ order Runge-Kutta advection scheme; and an improved ice crystal coagulation model (Schumann, 2012).

### A3.2 Atmospheric radiation data

All atmospheric data required in the radiative forcing model are taken from observations by CERES instruments on three orbiting platforms (NASA Langley Research Center Atmospheric Science Data Center 2015). CERES data are provided on a $1°\times1°$ resolution global grid at three-hour intervals. No interpolation is performed between estimates.

The terrestrial, longwave radiation flux is simulated using the estimated "clear-sky" outgoing longwave flux provided by CERES. The "clear sky" flux is the estimated flux in the absence of clouds. This removes the need to estimate outgoing fluxes based on indirectly-observed surface temperatures. Longwave emission from cloud layers is calculated as described in Corti and Peter (2009). The total incident shortwave radiation $S$ is computed using the solar zenith angle calculated based on time and geographic location (Kalogirou, 2014) as

$$S = S_0 \left(1 + 0.033 \cos\left(2\pi \cdot \frac{J}{365}\right)\right) \mu, \tag{A4}$$

where $S_0$ is the solar constant (1366.1 W/m$^2$), $\mu$ is the cosine of the solar zenith angle $\theta$, and $J$ is the Julian Day.

### Appendix B: Comparison of the single layer model

We compare the simulated radiative forcing for a single contrail layer against the existing FL (Fu and Liou, 1992, 1993; Fu, 1996; Fu et al., 1997) and CoCiP (Schumann, 2012) cirrus cloud radiative transfer

models. Fu and Liou model version 200503 is openly distributed by NASA Langley (Fu and Liou, 1993; Kato et al., 2005; Rose et al., 2006; NASA Langley Fu & Liou Radiative Transfer Code) while CoCiP radiative transfer model is in detail described in Schumann (2012).

Section B1 describes the different inputs for the three models, and demonstrates how the RF simulated

by each model varies as a function of the chosen input parameters. In Section B2, we simulate the change in shortwave RF as a function of surface albedo in all three models. In the comparison case we obtain, for $RF_{SW}$, a difference of less than 15% for $\theta < 80°$ (with smaller difference at smaller solar zenith angles). At high solar zenith angles ($\theta > 80°$), this difference can grow to up to 20%, while the difference in $RF_{LW}$ is always within 10%.

### B1 Comparison of our model against existing approaches

Each of the three models uses a different representation of ice particle optical properties. FL uses a "generalized diameter", assuming hexagonal ice columns (Fu and Liou, 1993). CoCiP can simulate a

number of different ice particle shapes for a given ice particle effective radius. Our model requires instead the asymmetry parameter of the layer. To enable reasonable comparisons, we start from the most complex of the three models, FL. This represents the ice crystals using a "generalized diameter" that we choose between 20 and 130 μm. We use data from table 1 of Fu (1996) to deduce the effective radius used for

CoCiP (21-112 μm). We finally use Fig. 5 from Key et al. (2002) to estimate the asymmetry parameter
corresponding to each given particle radius (0.75 – 0.92).

To test the level of agreement, we simulate a single contrail layer under clear-sky conditions. We use a
fixed contrail altitude (11 km), a fixed albedo of 0.3, a fixed outgoing longwave radiation flux of 278
W/m$^2$ and same TOA outgoing solar radiation, with no aerosol layers. We simulate multiple optical depths
between 0.01 and 0.5, and simulate the effect for solar zenith angles of 0 to 90°. Figure B1 shows the RF
components simulated by each model when sweeping across the given range of optical properties. The
positive values are the longwave component, while the negative values are the shortwave.

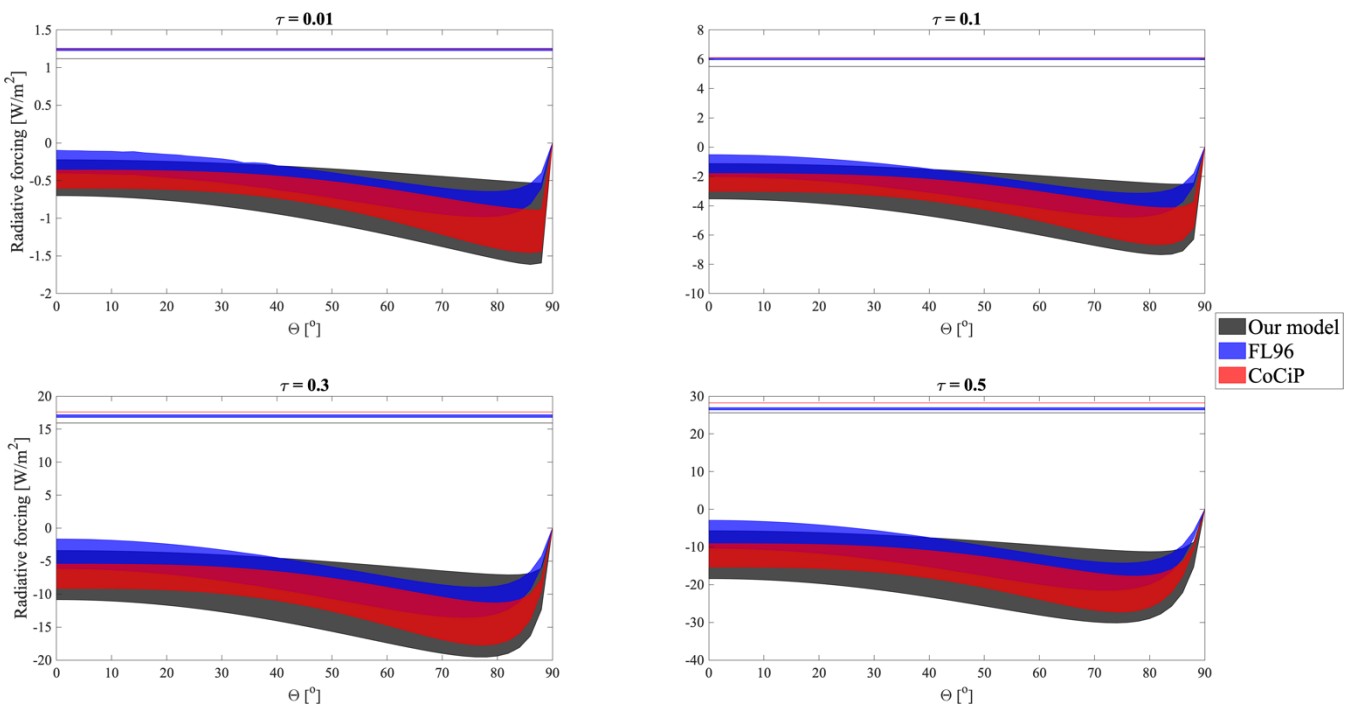

**Figure B1.** Longwave (positive) and shortwave (negative) radiative forcing ranges (varying particle size) in W/m$^2$ with the three models
tested here: our model (based on Corti and Peter), FL, and CoCiP. Variations with optical depth and solar zenith angle are shown.

Qualitatively, the models behave similarly. Variation in longwave radiative forcing in response to
changing optical properties is negligible in all three models, but our model consistently estimates a lower
RF$_{LW}$ than is estimated by CoCiP. This error is maximized at low optical depths, reaching ~10%. The

estimate from FL varies, agreeing more closely with CoCiP at low optical depths and more closely with our model at high optical depths. This error might be due to the longwave scattering neglection from the original Corti and Peter model (Corti and Peter, 2009), further commented in the limitations Section (Section 5).

Shortwave radiative forcing varies significantly with changes in optical properties in all three models. The range of asymmetry parameters simulated by our model results in a greater overall variation than is observed in the range of properties tested for FL or CoCiP. Qualitatively, the behavior of our model as the solar zenith angle ($\theta$) increases matches that of CoCiP closely. $RF_{SW}$ increases slowly with $\theta$, before reaching a peak between $\theta = 75°$ (high optical depths) and $88°$ (low optical depths). At values of $\theta$ beyond

this peak, $RF_{SW}$ falls rapidly to 0. In FL, the shape of the relationship is similar at all optical depths, with the minimum value occurring approximately at $75°$. We also evaluate the difference in average value for each of the models, within the ranges of comparable microphysical properties. We obtain an average difference of less than 10% between CoCiP and our model, with the greatest error being ~20% at $\theta > 80°$, expected from the here used two-stream approximation and mentioned in the limitations section. We find

a greater average difference of ~40% between our model and FL.

To perform more quantitative analysis and comparison, we select a specific value of the relevant optical parameter for each model. For this purpose, we choose an effective radius for ice of 45 µm. This is consistent with a natural ice cloud at an altitude of ~11 km, based on published parameterizations

(Heymsfield and Platt, 1984; Corti and Peter, 2009; Lolli et al, 2017; Heymsfield et al., 2014). A prior analysis by Corti and Peter (2009) found that an asymmetry parameter $g$ of 0.87 gave results which most closely matched those from FL, and as such we use that value here. Key et al. (2002) also confirmed that this is consistent with solid columns of the mentioned size. Using the approach outlined earlier, this crystal size is represented in CoCiP using an effective radius of 45 µm and in FL using a generalized diameter

of 46 µm. This specific single-contrail experiment results in differences in $RF_{SW}$ between our model and FL which are below 15%.

## B2 Comparison of albedo effect on single contrail RF$_{SW}$

To evaluate the consistency of our single contrail radiative forcing model, we simulate the effect of changes in surface albedo on single contrail shortwave radiative forcing and compare the results to both FL and CoCiP. Figure B2 shows the variation of RF$_{SW}$ with albedo in each model at three different optical depths. As previously explained, CoCiP uses an effective radius of 45 μm and FL uses a generalized diameter of 46 μm, while our model is using an asymmetry parameter of $g = 0.87$.

We observe the same qualitative behavior in all 3 models. Neglecting the already mentioned differences at high solar zenith angles ($\theta > 80°$), FL and CoCiP quantitatively agree best with our model at low albedos ($\alpha < 0.5$), with overall differences below 30%. Our model predicts a higher cooling impact at low solar zenith angles, and a lower cooling at high solar zenith angles. Maximum differences are found at high albedos ($\alpha > 0.6$) and high solar zenith angles ($\theta > 50°$), where our model significantly underestimates RF$_{SW}$, with differences of approximately 50%. However, these differences are less than 2 W/m$^2$ in absolute terms. The best agreement is found at an albedo of 0.3, the global Earth average albedo, with less than 10% difference. Finally, there are significant differences with CoCiP at low solar zenith angles and high albedos due to the forced negative sign of RF$_{SW}$ with that model. All percentage differences between the models are insensitive to changes in optical depth.

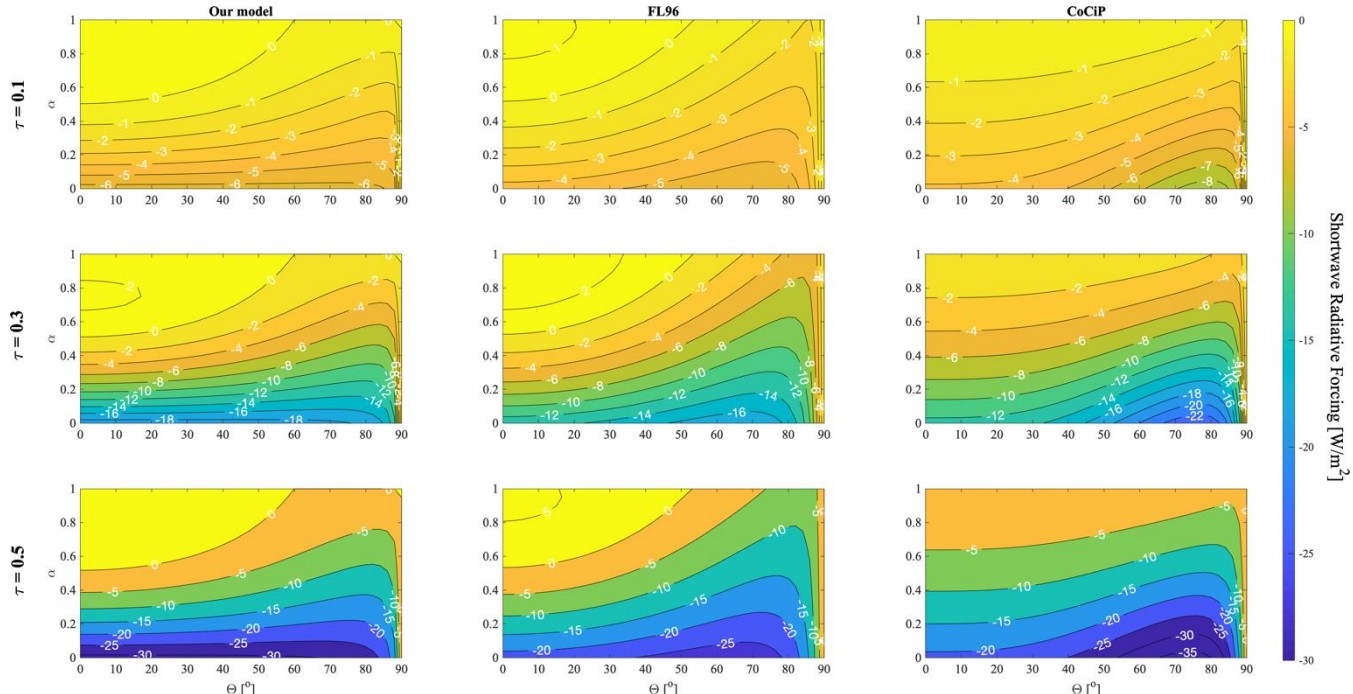

**Figure B2.** Shortwave radiative forcing in W/m² of a single contrail as a function of surface albedo (Y-axis) and solar zenith angle (X-axis). Each column corresponds to a different model, and each row corresponds to a different contrail optical depth.

## Appendix C: Shortwave RF model for overlapping layers

### C1 Simplification of reflections between two infinite layers

Different formulations have been developed to address radiation transfer between multiple layers, solving problems from very diverse topics: from estimating scattering in layered surfaces, through 1D transport theories (Hanrahan and Krüger, 1993) or by the transport matrix method (Byrnes, 2016), to representing cloud overlap with an effective decorrelation length (Barker, 2008). The simple expression of reflectance from Coakley and Chylek (1975), used in Corti and Peter model, allows us to develop our own formulation.

In this section we develop the formulation for calculating shortwave radiative forcing for a 2- and 3-layers overlap and deduce a formulation applicable to an N-layers overlap. We start by recalling single contrail

RF$_{SW}$ equation (Section C1.1), defined in main paper. We then develop the formulation for a 2-layers overlap (Section C1.2) and finish by extend the formulation to an N-layers overlap (Section C1.3), resulting in a simple formula easily applicable to our contrail coverage data.

### C1.1 Single layer RF$_{SW}$

When evaluating shortwave radiative forcing of $N$ infinite overlapping layers, we have to consider all the interactions between layers including reflectance and transmittance. We assume that cloud layers reflect shortwave radiation without diffusing it, whereas the Earth's surface diffuses incoming radiation in every direction (Corti and Peter, 2009). Using these assumptions, we can decompose mathematically all the

radiation interactions between layers.

As given in Appendix A, the shortwave radiative forcing of a single contrail can be expressed as

$$RF_{SW} = -S \cdot t(1-\alpha)\left(\frac{R-\alpha R'}{1-\alpha R'}\right) \tag{C1}$$

where $S$ is the solar constant, $\alpha$ is the Earth's surface albedo, $t$ is the mean atmospheric transmittance, and $R$ and $R'$ are the direct and diffuse reflectances of the contrail. This expression can be rewritten as

$$RF_{SW} = S \cdot t(\alpha - \alpha_1) = S \cdot t(\alpha - \alpha_{11} - \alpha_{12}) = S \cdot t\left(\alpha - R - \alpha\frac{TT'}{1-\alpha R'}\right) \tag{C2}$$

with $(T, T')$ being the direct and diffuse transmittances ($T = 1 - R$, $T' = 1 - R'$). We can divide the shortwave RF of a single contrail (contrail $i$) into two different components, $\alpha_{i1}$ and $\alpha_{i2}$. The first, $\alpha_{i1}$ is equivalent to the contrail "albedo" for direct radiation, and is in this case simply $R$. The second, $\alpha_{i2}$ can then be thought of as the contrail "albedo" for diffuse radiation – in this case, $\alpha\frac{TT'}{1-\alpha R'}$.

### C1.2 Two-layer RF$_{SW}$

Now consider a situation with two overlapping cloud layers, whose optical properties are fully captured by their individual reflectances ($R_1$ and $R_2$ for direct, $R'_1$ and $R'_2$ for diffuse). Under the assumptions listed above, and ignoring edge effects, Fig. C1 diagrams the how one incoming unit of radiation ($S = 1$) will interact with these two layers.

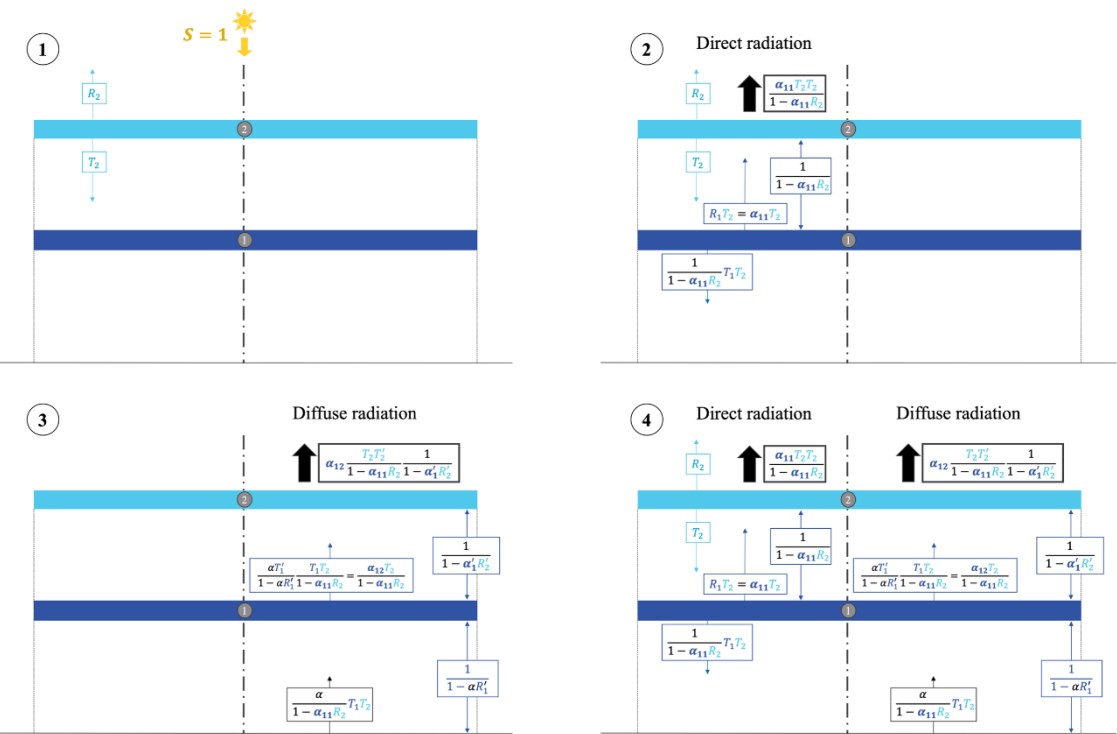

**Figure C1.** Decomposition of interactions between two cloud layers when receiving a single unit of shortwave radiation ($S = 1$). Sub-panels 1, 2, and 3 show three successive steps in the calculation as referred to in the text. $R_1$: direct reflectance, $T_1$: direct transmission ($=1-R_1$); $R'_1$: diffused reflectance, $T'_1$: diffused transmission ($1-R'_1$). Subscripts (1 and 2) indicate the layer number. $\alpha$ = Earth albedo. Text color indicates the layer which most recently interacted with the radiation, with radiation from layer 1 in dark blue, from layer 2 in light blue, and from the Earth (reflected) in black.

Subpanel 1 shows the initial interaction between incoming (direct) radiation and the upper contrail (contrail number 2). This contrail reflects a proportion $R_2$ of the incoming direct light and transmits (allows to pass through) a fraction $T_2$, equal to $1-R_2$. This would show all direct radiation interactions if there were no lower contrail, as light reflecting off the Earth's surface is assumed to be diffuse.

Subpanel 2 shows the full set of interactions for the incoming, direct, radiation when including both contrails. The fraction which passed through the upper contrail, $T_2$, now undergoes an infinite number of reflections between contrails 1 and 2. On each reflection, some fraction ($T_1$ and $T_2$, respectively) of the reflected light passes through. This results in a geometric series, which can be summed to yield the total radiation which passes through the upper contrail (back to space) or lower contrail (towards the ground).

Ignoring these reflections, the radiation passing to the ground would be simply $T_1T_2$; the reflections increase this by a factor of $\frac{1}{1-\alpha_{11}R_2}$, such that $\frac{T_1T_2}{1-\alpha_{11}R_2}$ is the total radiation heading towards the surface. The total which leaves upwards, back to space, is then $R_2 + \frac{\alpha_{11}T_2T_2}{1-\alpha_{11}R_2}$.

 

    Subpanels 3 then shows how diffuse radiation, reflecting off the Earth's surface, interacts with the system.

As shown in Subpanel 2, the total direct radiation which reaches the ground is $\left(\frac{T_1T_2}{1-\alpha_{11}R_2}\right)$, of which only a fraction $\alpha$ is reflected back upwards as diffuse radiation. There are now two sets of infinite reflections to consider. The first is between the Earth and lower contrail, resulting in a geometric series which can be summed to $\frac{1}{1-\alpha R_1'}$ - now using the diffuse reflectance $R_1'$ instead of the direct reflectance $R_1$. The second is between the two contrails, and can be expressed using the effective "albedo" of the lower contrail $\alpha_1' (=$

$\alpha_{11}' + \alpha_{12}' = R_1' + R_2')$. This geometric series can then be expressed as $\frac{1}{1-\alpha_1'R_2'}$. From these equations, it becomes clear that the effect of additional contrails is to have additional "albedos", each of which modifies the total radiation which is either reflected to space or eventually absorbed by the Earth's surface (through repeated reflections).

 

The combination of the direct and diffuse radiation fluxes can then be seen in Subpanel 4; each upwards arrow from contrail 2 represents a separate component which will escape back to space. Adding these together and subtracting from the radiation which would be reflected to space under a clear-sky scenario (i.e. the Earth's albedo), the total shortwave RF can be summarized as

$$RF_{SW} = S \cdot t(\alpha - \alpha_2) = S \cdot t(\alpha - \alpha_{21} - \alpha_{22}) = S \cdot t\left[\alpha - R_2 - \alpha_{11}\frac{T_2 T_2}{1 - R_1 R_2} - \alpha_{12}\left(\frac{T_2 T_2'}{1 - R_1 R_2}\right)\left(\frac{1}{1 - \alpha_1' R_2'}\right)\right] \tag{C3}$$

where we have now combined the terms into two effective "albedos". These terms allow us to treat the combined layer pair as if it were a single contrail. Specifically, we have $\alpha_{21}$ $(= R_2 + \alpha_{11}\frac{T_2 T_2}{1-R_1 R_2})$ being the "albedo" of the layer pair to direct radiation, and $\alpha_{22}$ $(= \alpha_{12}\frac{T_2 T_2'}{1-R_1 R_2}\frac{1}{1-\alpha_1' R_2'})$ being the "albedo" of the layer pair to diffuse radiation.

### C1.3 *N*-layer RFsw

This approach extends from 2 to *N* layers by following the same mathematical logic (see Table C1), using as "albedo" values $(\alpha_i)$ the direct and diffuse "albedos" of the (N-1) layers below the top one.

Table C1. Developed expression of RFsw/*St* for multiple layers overlaps

| # of layers | RFsw expression $(= RF_{SW}/St)$ |
|---|---|
| 1 | $\alpha - \alpha_1 = \alpha - \alpha_{11} - \alpha_{12} = \alpha - R_1 - \dfrac{T_1}{1 - \alpha R_1'}\alpha T_1'$ |
| 2 | $\alpha - \alpha_2 = \alpha - \alpha_{21} - \alpha_{22} = \alpha - R_2 - \dfrac{T_2}{1 - \alpha_{11}R_2}\left(\alpha_{11}T_2 + \dfrac{\alpha_{12}T_2'}{1 - \alpha_1' R_2'}\right)$ |
| 3 | $\alpha - \alpha_3 = \alpha - \alpha_{31} - \alpha_{32} = \alpha - R_3 - \dfrac{T_3}{1 - \alpha_{21}R_3}\left(\alpha_{21}T_3 + \dfrac{\alpha_{22}T_3'}{1 - \alpha_2' R_3'}\right)$ |
| N | $\alpha - \alpha_N = \alpha - \alpha_{N1} - \alpha_{N2} = \alpha - R_N - \dfrac{T_N}{1 - \alpha_{(N-1)1}R_N}\left(\alpha_{(N-1)1}T_N + \dfrac{\alpha_{(N-1)2}T_N'}{1 - \alpha_{(N-1)}' R_N'}\right)$ |

The resulting "albedos" for direct $(\alpha_{N1})$ and diffuse $(\alpha_{N2})$ radiation in a N-layer overlap are then the following:

$$\alpha_{N1} = R_N + \alpha_{(N-1)1}\frac{T_N T_N}{1 - \alpha_{(N-1)1}R_N} \tag{C4}$$

$$\alpha_{N2} = \alpha_{(N-1)2} \frac{T_N T_N'}{1 - \alpha_{(N-1)1} R_N} \cdot \frac{1}{1 - \alpha'_{(N-1)} R_N'} \tag{C5}$$

This calculation means that we can collapse the effect of $N$ different layers on shortwave radiation into the effect of a single, combined layer, as long as we know the direct and diffuse reflectances of each layer.

If we assume that all layers have the same optical properties (identical asymmetry parameter $g$ and therefore identical optical parameter $\gamma$) we can simplify this further. Using the definition of $R$ from the main text, we find that the direct albedos for 2 and 3 layers can be written as $\alpha_{21} = \frac{(\tau_1 + \tau_2)/\mu}{\gamma + (\tau_1 + \tau_2)/\mu}$ and $\alpha_{31} = \frac{(\tau_1 + \tau_2 + \tau_3)/\mu}{\gamma + (\tau_1 + \tau_2 + \tau_3)/\mu}$. Extrapolating to an arbitrary $N$ layers, we find that

$$\alpha_{N1} = \frac{\sum_{i=1}^{N} \tau_i/\mu}{\gamma + \sum_{i=1}^{N} \tau_i/\mu} \tag{C6}$$

Therefore, the direct "albedo" from an $N$-layer overlap of similar layers (same optical properties) is equal to the direct reflectance of a single layer with the same total (summed) optical depth. The same logic can be applied to the diffuse albedo.

If the overlap occurs between layers of different optical properties, the same method can be applied as long as a single "effective" asymmetry parameter $g_e$ can be used for all layers. A method to find this parameter is derived below (Section C2). Once this parameter is known, RF$_{SW}$ for multiple overlapping contrails can be reduced to that for a single layer, i.e.

$$RF_{SW} = S \cdot t \left( \alpha - R_e - \alpha \frac{T_e T_e'}{1 - \alpha R_e'} \right) = -S \cdot t (1 - \alpha) \left( \frac{R_e - R_e'}{1 - \alpha R_e'} \right) \tag{C7}$$

with $R_e = \frac{\sum_{i=1}^{N} \tau_i/\mu}{\gamma_e + \sum_{i=1}^{N} \tau_i/\mu}$ and $\gamma_e = \frac{1}{1 - g_e}$.

**C2 Derivation of weighted asymmetry parameter**

As outlined above, the calculation of shortwave radiative forcing for an $N$-layer overlap can be simplified significantly if a single, "effective" asymmetry parameter can be identified which characterizes the entire system. To calculate this effective optical parameter, we first determine what would be the effective optical parameter so that the direct radiation "albedos" are equal in both cases. We then show that matching this albedo is sufficient to ensure that the overall radiative forcing (accounting for both diffuse and direct radiation) matches between the "simplified" case and one in which each layer is treated independently.

In an $N$-layer overlap, a proportion $\alpha_{N1}$ of incoming direct radiation is reflected. The single effective layer reflects radiation through the factor $R_e$. The equality that must hold is then

$$\alpha_{N1} = R_N + \frac{\alpha_{(N-1)1} T_N T_N}{1 - \alpha_{(N-1)1} R_N} = R_e \tag{C8}$$

As for Eq. (C6), developing the expression of direct radiation albedo for a 2 and 3-layers overlap we obtain $\alpha_{21} = \frac{\gamma_1 \tau_2/\mu + \gamma_2 \tau_1/\mu}{\gamma_1 \gamma_2 + \gamma_1 \tau_2/\mu + \gamma_2 \tau_1/\mu}$ and $\alpha_{31} = \frac{\gamma_1 \gamma_3 \tau_2/\mu + \gamma_2 \gamma_3 \tau_1/\mu + \gamma_1 \gamma_2 \tau_3/\mu}{\gamma_1 \gamma_2 \gamma_3 + \gamma_1 \gamma_3 \tau_2/\mu + \gamma_2 \gamma_3 \tau_1/\mu + \gamma_1 \gamma_2 \tau_3/\mu}$ . If we assume that $\alpha_{(N-1)1} = \frac{\left(\sum_{i=1}^{N-1} \prod_{j\neq i} \gamma_j \tau_i/\mu\right)}{\left(\prod_{i=1}^{N-1} \gamma_i\right) + \left(\sum_{i=1}^{N-1} \prod_{j\neq i} \gamma_j \tau_i/\mu\right)}$ , we obtain $\alpha_{N1} = \frac{\left(\sum_{i=1}^{N} \prod_{j\neq i} \gamma_j \tau_i/\mu\right)}{\left(\prod_{i=1}^{N} \gamma_i\right) + \left(\sum_{i=1}^{N} \prod_{j\neq i} \gamma_j \tau_i/\mu\right)}$ . This then yields the following expression, for any N:

$$\alpha_{N1} = \frac{\left(\sum_{i=1}^{N} \prod_{j\neq i} \gamma_j \, \tau_i/\mu\right)}{\left(\prod_{i=1}^{N} \gamma_i\right) + \left(\sum_{i=1}^{N} \prod_{j\neq i} \gamma_j \, \tau_i/\mu\right)} \tag{C9}$$

Equalizing expression (C9) with the effective reflectance of direct radiation $\left(R_e = \frac{\sum_{i=1}^{N} \tau_i/\mu}{\gamma_e + \sum_{i=1}^{N} \tau_i/\mu}\right)$ we find an expression for the effective optical parameter of the entire layered system:

$$\gamma_e = \left(\prod_{i=1}^{N} \gamma_i\right) \frac{\sum_{i=1}^{N} \tau_i/\mu}{\sum_{i=1}^{N} \prod_{j\neq i} \gamma_j \, \tau_i/\mu} \tag{C10}$$

If we use expression (C10) to calculate the effective diffuse radiation albedo $\left(\alpha \frac{T_e T_e'}{1 - \alpha R_e'}\right)$ and expand each term, it results in the same formula for $\alpha_{N2}$ as is shown in Equation (C5). Since both the diffuse and direct

albedos of the system are now matched, the total $RF_{SW}$ of the contrail layer system can be calculated by treating it as a single layer with the optical parameter shown in Eq. (C10).

### C3 Variation of scattering with solar zenith angle


The objective of this section is to assess how the solar zenith angle ($\theta$) affects the potential cooling impact from contrails. In Section 4.1.1 we stated that increasing the solar zenith angle $\theta$ also decreases the (net positive) contrail radiative forcing. This is because of an increase in shortwave cooling, since longwave radiation is not affected. Figure C2 shows how the total upscattered fraction of radiation is affected by

changes in solar zenith angle. $\theta$ varies from 0 (noon) to 90° (sunset), moving anti-clockwise from the top left figure and shown as a black arrow. The dotted horizontal line represents the horizon. F and B represent downward (towards Earth) and upward (back to space) scattering. We assume that 90% of incident radiation is scattered forward, with 10% scattered backwards, representing the high forward scattering fraction of ice particles (high asymmetry parameter $g$). As the solar zenith angle increases, a greater

fraction of the forward scattering peak is directed towards space (greater upscatter). This results in an increase cooling effect near sunrise or sunset compared to noon time.

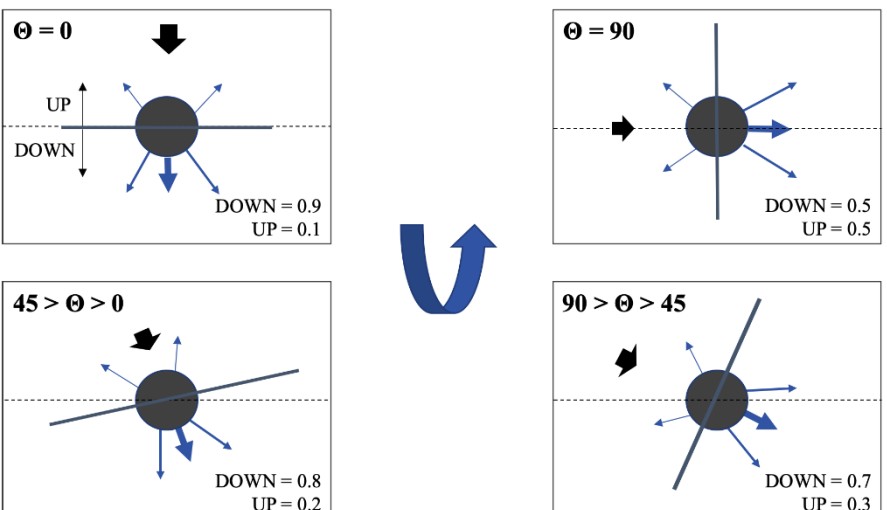

**Fig C2.** Change in particle reflection when variation in $\theta$ (DOWN: downscatter towards Earth; UP: upscatter back to space)

**Appendix D: Theoretical explanation of a decrease in cooling when accounting for overlap**

This appendix mathematically explains an interesting feature obtained in Section 4.1.2 related to the effect of two overlapping layers on the shortwave radiation reflectance. This specific result is interesting but does not significantly affect the overall impacts attributable to contrails.


In Section 4.1.2 we found that, in a small interval of low optical depths and at high solar zenith angles, the amount of radiation reflected when overlapping is higher than the amount of radiation reflected if the two layers were independent, resulting in a higher absolute value of the shortwave RF (Fig. 3). This is anomalous since two overlapping layers would be expected to reflect less sunlight due to the reduction of

covered area.

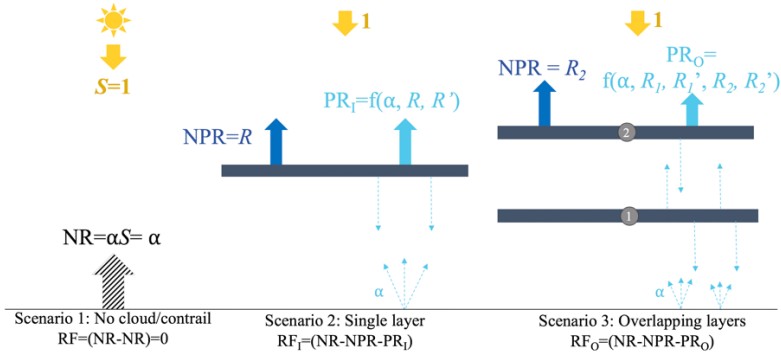

**Figure D1.** Components of response to a unit of incident light for 3 scenarios: no cloud/contrail, single layer, overlapping layers

In order to explain this, we decompose the fraction of the incident SW radiation flux S reflected by layers of clouds or contrails into non-participating radiation ("NPR") and participating radiation ("PR") (Fig.

D1). NPR is the light which is reflected into space from the upper contrail and therefore does not participate in scattering. In turn, it increases with $R_c$, which rises with optical depth. PR is the remaining outgoing shortwave radiation. Since all light included in PR was reflected or diffused, i.e. it has "participated" in scattering between the layer(s) and the Earth's surface, PR is driven by both direct reflectance $R_c$ and diffused reflectance $R_c'$. PR decreases with increasing optical depth. Finally, NR is the

natural reflectance of light by the surface of the Earth, proportional to its albedo. We note that in the "clear sky" scenario, the total outgoing shortwave radiation is NR = $\alpha S$.

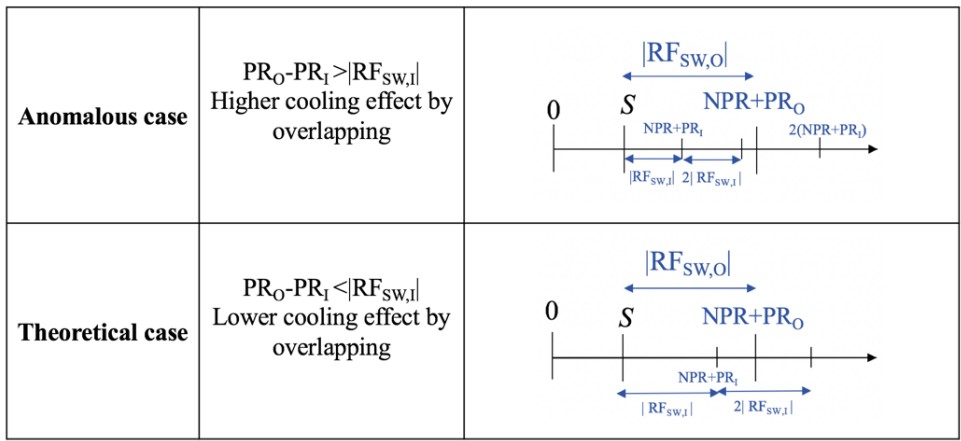

**Figure D2.** Comparison scheme of reflection components in an overlap (Upper case: increase in cooling; lower case: decrease in cooling)

With this decomposition, we can compute the shortwave RF of a single layer per unit of incoming radiation by comparing the outgoing shortwave radiation with no cloud ($\alpha$) to that with a cloud layer (NPR – PR). This yields $RF_{SW,I} = (\alpha - NPR - PR_I)$. For two overlapping layers, the shortwave RF is $RF_{SW,O} = (\alpha - NPR - PRO)$. We then compare this finding to previous work which treats the two layers independently, so that $RF_{SW,2I} = 2 \times (\alpha - NPR - PR_I)$.

First, we can see from the $RF_{SW,2I}$ expression, that the "clear sky" reflection is accounted for twice, which doesn't reflect the reality when two layers overlaps. This indicates that considering two independent layers for calculating RF when these two-layers overlap is not a correct assumption. Additionally, the absolute value of the shortwave RF, or cooling effect, of overlapping contrails will exceed the independently computed cooling effect of two overlapping layers if $(PR_O - PR_I) > |RF_{SW,I}|$, shown schematically in Figure D2. Although $PR_O$ is always higher than $PR_I$, due to the additional upscatter from the lower layer, this explains why the net cooling is only increased by overlapping (compared to two independent layers) for small optical depths.

As a conclusion, under specific circumstances (low optical depth and solar zenith angle), two contrails overlapping will reflect more radiation (higher cooling effect) than if they were independent,

compensating the higher covered area. However, the difference in $RF_{SW}$ for these cases is small enough that it has no noticeable effect on global average values.