# Peer review of "Impacts of multi-layer overlap on contrail radiative forcing"

_Atmospheric Chemistry and Physics, 2020_

## Referee Comment (RC1) · Anonymous Referee #1 · 25 May 2020

The paper discusses the impact of overlap of contrails with other clouds and of contrails with other contrails for horizontally homogeneous clouds and contrails.

The paper offers a simplified radiative transport model to account for RF changed for verlapping cloud layers.

The main conclusions are: 1) Contrail-cloud overlap is important. Contrails over clear sky and contrails over other clouds have far different radiative forcings (RF).

True. However, that finding is not surprising and not new. It is not surprising because the clouds below and above contrails change the reflected solar radiation (local Earth albedo) and the outgoing longwave radiation. It is not new since you find that in several previous papers. See, for example Fig. 6, column 6, in Meerkötter et al. (1999; cited in

the paper). That paper discussed the sensitivity of RF to various contrail and ambient parameters, including a cloud layer below the contrail cirrus and the optical depth of "background clouds". It showed that the net RF can increase from close to zero to a large positive value when background clouds get included. The value (94%) stated in the present paper here has no significance because it depends on the reference value and may vary from minus infinity to plus infinity when the net RF happens to be close to zero for the reference case considered. That makes no sense.

The present paper mentions Minnis et al (1999, cited) and Myhre et al. (2009, cited) and other studies, who discussed contrail-cloud overlap. The abstract and conclusions stress the importance and uncertainties of contrail-cloud overlap, which is correct, but report the findings as if that would be new, which is not correct. Apparently this discussion still reflects the history of the present paper, which apparently started with the Corti-Peters model with just one cloud layer (contrails or cirrus) over Earth surface and where the inclusion of other clouds changed the results considerably. This needs to be fully revised.

2) Contrail-contrail overlap depends on the number and proximity of contrails. For present traffic, contrail-contrail overlap occurs on average over the globe but only rarely. This overlap may occur more frequently for increased traffic and under special flight track conditions.

The treatment of contrail-contrail overlap is interesting. In areas with dense traffic, many contrails overlap with each other and with other clouds to different degrees. It would be good to have an efficient and still accurate method to account for the climate impact of contrails in such situation.

Contrail-contrail overlap may not be the most important uncertainty in contrail RF modelling. More important parameters may include the amount of ice supersaturation available in the atmosphere, the growth of the contrail cross-section by mixing with ambient air and the life time of contrails depending on many parameters (Schumann and

Heymsfield, 2017, cited, and the references cited therein).

Still, an investigation of contrail-contrail overlap effects and their modelling is of interest.

Comments on the approach and results:

The radiative transfer model used to account for multiple layers (Section 2.1.3) looks interesting. It seems to have similarities with older theories; see, e.g., Hansen and Travis (1974) and Minnis et al. (1993); Minnis et al. (1998). A paper much cited in this respect is that of Ritter and Geleyn (1992). This part needs review by experts in this specfic field.

Section 3.1.4 compares results for this model with the Fu&Liou model. That is certainly an acceptable approach, as long as one can justify the plane-parallel cloud representation. Only few details are given on how the code was applied, for example with respect to the background atmosphere and aerosols and the specific model parameters. The comparison shows qualitative agreements with the multilayer model derived, but significant quantitative differences. So, how can we be sure that the results are correct? So uncertainties remain.

Eq. (3) needs a bit more discussion: As it is written, this equation does not guarantee that RF_LW is positive. How often are negative values occurring?

Is Eq. (9) correct? 360? is that 360° (2 pi)?

The discussion of the range of optical depth values (tau below 0.3) might be reasonable for global mean value, but locally the variability can be far larger (Atlas and Wang 2010).

The paper presents contrail results from the model CERM. As stated, CERM does not account for contrail orientation and does not account for contrail position in a grid cell. How can one compute the contrail-contrail overlap effects without knowing the degree of overlap? The mentioned model CoCiP includes such geometry in more detail.

How good do the meteorological data used represent humidity? Which time period is

covered by the data, what is the spatial and temporal resolution of the data, and what is the vertical resolution in meters near the mean flight level height (around 10 to 12 km asl)? What is the fraction of ice supersaturated air masses in these data and how does this compare to published findings.

I assume, the model uses gridded emission source rates as provided by the FAA's Aviation Environmental Design Tool, but the paper gives no details on this. Are these data accessible to the community? Otherwise the results cannot be checked by other scientists.

Very little is said about the satellite data CERES. The paper cites NASA Langley Research Center Atmospheric Science Data Center 2015 as reference and says that the data are provided at three-hour intervals. How can one derive hourly average values from 3-hourly data? How well do they represent the diurnal cycle and how sensitive are they to cirrus clouds and to geometrically thin contrails? How uniform is this sensitivity spatially and temporally, e.g. over land and oceans?

The paper says that the "CERES instruments observe both contrails and natural cirrus clouds". I assume this means that CERES provides information only on the sum of contrails and other clouds. That should be clarified.

The conclusion claims that the results "help to inform policymakers and researchers to identify technical, operational, and regulatory means to reduce these impacts." I think, based on the information given, the paper is still quite far away from this goal.

The conclusion "The radiative forcing attributable to a contrail layer increases by a factor of three due to the presence of natural clouds on a global mean basis, but this varies by region" could be formulated inversely, e.g. "if a model would ignore other clouds the results could be wrong by a factor of three", but the conclusion should also make clear that this is not state of the art. Other models do account for ambient clouds.

In the abstract, the growth rate of air traffic is cited. I agree, growth rates of 4.5 %

each year over the next 20 years have been estimated in the past by industry, as cited. However such trend values have large uncertainty and I would recommend omitting such uncertain values from the abstract.

Unfortunately, the paper is not really clear and understandable and the conclusions are overselling the findings. The subject of the paper and some of the results are interesting but the approach and its presentation require major improvements.

I suggest splitting the paper into two parts: One on radiation transfer and one on the application. The first one should describe the model for the impact of cloud overlap on radiative forcing as a purely technical paper, with full validation. That paper should be reviewed by radiation transfer modelling experts. The other paper might then deal with the consequences of contrail-contrail overlap for climate forcing of aviation, addressing the corresponding community.

References

Atlas, D., and Z. Wang, 2010: Contrails of small and very large optical depth. J. Atmos. Sci., 67, 3065-3073, doi: 10.1175/2010JAS3403.1.

Hansen, J. E., and L. D. Travis, 1974: Light scattering in planetary atmospheres. Space Sci. Rev., 16, 527-610,

Minnis, P., K.-N. Liou, and Y. Takano, 1993: Inference of cirrus cloud properties using satellite-observed visible and infrared radiances. Part I: Parameterization of radiance fields. J. Atmos. Sci., 50, 1279-1304,

Minnis, P., D. P. Garber, D. F. Young, R. F. Arduini, and Y. Takano, 1998: Parameterizations of reflectance and effective emittance for satellite remote sensing of cloud properties. J. Atmos. Sci., 55, 3313-3339,

Ritter, B., and J. F. Geleyn, 1992: A comprehensive radiation scheme for numerical weather prediction models with potential applications in climate simulations. Mon. Wea. Rev., 120, 303-325, doi: 10.1175/1520-

0493(1992)120<0303:ACRSFN>2.0.CO;2.

---

## Referee Comment (RC2) · Anonymous Referee #2 · 29 Jun 2020

The paper investigates the impact of cloud-contrail and contrail-contrail overlap on the radiative forcing due to contrail cirrus. The authors use a radiative transfer model of Corti and Peter that they modify in order to study the impact of cloud overlap on contrail radiative forcing. They consider two options no overlap and maximum overlap and study the difference when using those two assumptions. Cloud and contrail properties are varied systematically and the impact on LW and SW RF is analyzed. Cloud properties are prescribed using observed natural clouds (from Satellite) and CERM simulated contrails. Whereas, in principal the impact of cloud overlap is an interesting topic, the authors' extreme assumptions (no or maximum overlap) limit the relevance of the paper. They wrongly claim that other studies assume no or maximum overlap between contrails and clouds and contrails and contrails and claim that the sensitivity that they

see is a measure for the bias of contrail cirrus RF published in the literature. Relevant literature that discusses the impact of overlap on RF in detail has not been discussed and comparisons with the results in the literature only partly made.

I suggest presenting the work as a sensitivity study, including a detailed comparison with the many results in the literature and removing text stating that the present work improves contrail RF estimates and estimates the uncertainty in contrail RF in the literature or that the results help to inform policymakers and similar claims.

Major comments:

1. The sensitivity of contrail RF on the overlap analyzed in this paper is not a measure for the uncertainty in contrail RF in the literature. Both assumptions used in this paper, no or maximum overlap, are very extreme whereas in the literature mostly maximum-random overlap for contrail-cloud and contrail-contrail overlap has been used. The statements that in the literature mainly maximum overlap has been used (e.g. line 346-348) or that random overlap has been used for contrail-contrail overlap (line 350) the authors partly contradict their own table 1.

2. The estimate for contrail RF is not an improved estimate relative to the estimates in the literature

a. Neither maximum nor no overlap are good assumptions. Maximum overlap is certainly an upper bound for the overlap but far away from the truth.

b. Using only cloud data with 3 hourly temporal resolution does not allow for a realistic representation of contrail-cloud overlap or a realistic estimation of overlap frequencies. The low temporal resolution cannot resolve the correlation between cloud and contrail frequency.

c. Overlap assumptions have been shown to be dependent on vertical resolution. Even if vertically extended clouds are assumed the vertical overlap decreases strongly with layer depth. When levels are separated by more than 4km the overlap is essentially

random (Hogan and Illingworth, 2000). At a low vertical resolution random overlap is a good assumption while maximum overlap is realistic (for vertically extended clouds) at high vertical resolution. At lower horizontal resolution the arguments must include a discussion of synoptic situations and the resulting vertical and horizontal statistics of the moisture field. Using observed cloud statistics aggregated in only 4 atmospheric layers and calculating the overlap with contrails the assumption of maximum overlap is far from realistic.

d. Assuming maximum overlap between contrails and contrails is not realistic. Even if planes would follow each other on the same flight track advection would mean that contrails don't maximally overlap. But this topic does not seem to be very promising as contrails have a low optical depth and the overlap between contrails does not impact the radiative forcing strongly.

e. As the authors say whether a contrail cools or warms depends on the height of the clouds that may be vertically overlapping this contrail. The cloud height cannot be properly represented using cloud observations aggregated on only 4 levels.

3. Comparison of the results to the previous publications should be improved.

a. The publications of Markowicz and Witek (2011 a,b) have not be cited. They discuss the impact of contrail cloud overlap in great detail e.g. the dependence on particle habit. They also show that contrail RF turns negative for all considered ice crystal shapes at much higher optical depth (at zenith angle $30°$) then shown in fig. 3a. As a zenith angle of $30°$ is often used in the literature it would be good to supply results for that angle in order to allow for comparison.

b. The results in Fig. 7 should be compared with results of e.g. Schumann et al. 2012 and Myhre et al 2009 in detail. Why are absolute values of contrail RF so different from previous results?

4. The result that cloud-contrail overlap is responsible for 93% of the net radiative

forcing attributable to contrails in 2015 relies on the assumption that the authors have simulated the 'true' overlap between contrails and natural clouds which is not the case. Instead they should say that overlapping contrails maximally with clouds instead of prescribing no overlap leads to an increase in radiative forcing by xx%. The same is true for the statements about the importance of contrail-contrail overlap. Note also that those values will be resolution dependent so that their significance is very limited and should not appear in the abstract!

5. Line 891-892: How do you suggest avoiding cloud contrail overlap? Contrails mostly form close to natural clouds which means that they often overlap with other clouds. Cloud-free areas are mostly dry and therefore persistent contrails cannot exist.

Minor comments:

1. Table 1 is incomplete, contains mistakes and is misleading: The table serves to show the great scatter in the contrail RF estimates but it omits to say

- that the Marquart et al and the Frömming et al estimates are for line-shaped contrails only whereas the other estimates are for contrail cirrus.

- the main difference between the contrail cirrus modeling studies are the different ways of treating contrails, keeping them separate from natural clouds or treating them with the cloud scheme, and the contrail initialization. The Chen and Gettelman study follows a very different approach from the others.

- That there is another estimate for contrail cirrus RF (Bock and Burkhardt, 2016 – which is already in the literature list) that lies in between the Schumann et al. and the Burkhardt and Kärcher estimate. This means that 3 of 4 estimates lie close together.

- Chen and Gettelman include contrail-contrail overlap since overlap is dealt with in the cloud scheme. The model assumes maximum random overlap for clouds. Only at the initialization stage contrails (age of up to ∼30 min.) do not overlap but that is not the same as no overlap between contrails in general.

That means that the scatter between contrail cirrus RF estimates is much smaller than suggested by the current table 1 and that the scatter is not due to the overlap scheme. Even though the uncertainty in the overlap between clouds and contrails leads to an uncertainty in contrail cirrus RF, the results from the literature do not demonstrate this fact as they tend to use the same overlap scheme. The range of net RF due to contrail cirrus encompasses the estimate of this study only because estimates for line-shaped contrails are included here.

2. Biofuels have little impact on contrail formation likelihood but instead on soot number emissions, ice nucleation and contrail life times and optical depth (Moore et al. 2017, Kärcher et al. 2018, Burkhardt et al. 2018)

3. Marquart et al (2003) and Frömming et al. (2011) both use the contrail parameterization of Ponater et al. (2002) and simulate line-shaped contrails of varying optical depth including those with an optical depth smaller than 0.02. They both calculate the fractional increase in cloudiness due to contrails. Only the overlap was calculated differently.

4. Chen and Gettelman do not assume zero overlap between linear contrails when calculating radiation. Instead they use the overlap scheme of CAM (maximum random) in order to take care of overlap between different clouds.

5. Page 6: You cite papers that determine if cloud-contrail overlap reduces or enhances contrail RF and conclude that this is a major uncertainty instead of mentioning the known fact that the effect depends on the contrail and cloud properties and temperature/height.

6. On page 7 you talk about contrail-contrail interaction. Please note that interaction can happen due to a number of processes and does not necessarily have to do with radiation.

7. On page 11 the error of the model compared to FL96 is determined. In that context

the systematic bias, the underestimation of the dependency on the zenith angle, should be discussed and its impact on the results of this paper.

8. Line 341-344: All 3 sentences are unclear. It is not clear what it means to aggregate single contrails into one layer! Are coverages due to single contrails added up or not? 'Overlap between . . . is therefore not explicitly resolved': What does that mean – minimum overlap? 'the same approach . . ... as if clouds were centered in the grid cell'. What does that have to do with overlap. Equations would make this text easier to understand.

9. Line 349: what does '(linear in most cases)' refer to?

10. Line 349: I assume you mean to say that contrail area 'overlaps' and not 'interacts'

11. Line 353: what is 'potential maximum overlap'

12. Line 355-357: even more useful would be a higher temporal and vertical resolution. The lower the resolution the larger the overlap.

13. In order to correct for errors in the model the authors use for the clear-sky estimates CERES data. Without any discussion where the errors are coming from it is difficult to understand if this approach is acceptable. The radiation emitted from the cloud could still be affected by this error which would mean that the estimated contrail RF would include this model error.

14. The term 'independent overlap' needs to be defined.

15. Line 434: The asymmetry parameter in Schumann et al is 0.787 (and not 0.77) for older contrails and 0.827 for younger contrails.

16. Line 503: Figure 3b is the upper right panel

17. Line 854: You probably meant to cite Kärcher et al., 2009 and not 2002.

18. Line 886 – 888: Those conclusions depend very much on the type of clouds you

prescribed. In the tropics many very thick clouds can be found and differences between cloud and contrail top temperature can be very large as well.

Markowicz, K. M., and M. L. Witek (2011), Simulations of contrail optical properties and radiative forcing for various crystal shapes, J. Appl. Meteorol. Climatol., 50(8), 1740–1755, doi:10.1175/2011JAMC2618.1.

Markowicz and Witek Sensitivity study of global contrail radiative forcing due to particle shape. JGR, 116, doi:10.1029/2011JD016345 (2011)

Moore, R. H. et al. Biofuel blending reduces particle emissions from aircraft engines at cruise conditions. Nature 543, 411–415 (2017).

Kärcher, B. Formation and radiative forcing of contrail cirrus. Nature Comm.Communications 9, :1824, DOI: https://doi.org/10.1038/s41467-018-04068-0 (2018).

Burkhardt, U., L. Bock, A. Bier, Mitigating the contrail cirrus climate impact by reducing aircraft soot number emissions. npj Climate and Atmospheric Science 1:37, https://doi.org/10.1038/s41612-018-0046-4 (2018).

Kärcher B, Burkhardt U, Unterstrasser S, Minnis P (2009) Factors controlling contrail cirrus optical depth. Atmos Chem Phys 9:6229–6254

---

## Short Comment (SC1) · 30 Jun 2020

Specific comments on the ACPD manuscript "Effect of contrail overlap on radiative impact attributable to aviation contrails" by I. Sanz-Morere et al.

Introductory remarks

I find this work of Sanz-Morere et al. quite interesting as it emphasizes an important aspect in contrail radiative impact studies that usually has not been investigated as systematically as is done here. In this respect, I feel the two official referees have taken a somewhat stern attitude towards the paper. I think that using a parameter scanning approach in assessing cloud-contrail and contrail-contrail overlap situations goes beyond what currently available studies have done. Figure 3 of the present paper is certainly worthwhile providing. Yet, the referees are certainly right when reminding the authors not to overreach their conclusions, given the limitations of the model framework used in the paper. It is also true, I agree, that some aspects (and even whole papers) of previous research work has been overlooked by the authors.

However, this specific comment is mainly written to provide additional information (including insider knowledge of previous papers) that can help to rectify some statements where the authors – in my view - have interpreted previous work inaccurately.

General comments

- Contrail-cloud overlap has been generally accounted for in previous contrail radiative impact studies in the way the contrail radiative forcing is usually given under clear-sky as well as all-sky conditions (Myhre and Stordal, 2001; Stuber and Forster, 2007; Yi et al., 2012, and others cited by the authors). It has been a common finding that both the shortwave and the longwave radiative forcing decrease in magnitude under cloudy-sky conditions. Often, but not always, the daily mean net radiative forcing gets more positive with natural clouds included.
- As rightly pointed out by referee 2, contrail-contrail overlap has usually been accounted for in contrail studies with global climate models (Marquart et al., 2003; Rap et al., 2010, Burkhardt and Kärcher, 2011; Bock and Burkhardt, 2016), except for when used in idealized setups like the GCMs (ECHAM and CNRM) contributing to Myhre et al. (2009). For illustration what situations can occur in climate models, I reproduce a figure from the PhD thesis of Marquart (2003, unfortunately only available in German language). This picture makes it clear that layers with contrails not only may overlap with layers containing natural cloud, but that situations with contrails and natural clouds existing side by side in the same grid box are also possible. That renders the overlap situations in models like that rather complicated, even if the overlap principle is straightforward. Anyway, it is clear that the climate model parameterizations can include the effect.

[Figure]

*Possible grid box column situations containing contrails (yellow) and natural clouds (grey) in the ECHAM4 model of Ponater et al. (2002), Marquart (2003), and Marquart et al. (2003). Adapted from Marquart, 2003, her Figure 2.7.*

Specific comments

- It is important to realize that the GCM studies (at least those based on the ECHAM climate model) use the maximum-random overlap scheme for calculating radiative fluxes in allsky columns. This refers to contrail-cloud overlap as well as to contrail-contrail overlap situations. See, for example, Figure 4 in Marquart and Mayer (2002).

- It may be of interest for the present study that the use of the maximum-random overlap principle has been shown to create severe problems when used in an un-favorable parameterization combination, as discussed by Marquart and Mayer (2002). This has caused the radiative forcing values given by Ponater et al. (2002) to be basically incorrect (amended in Marquart et al., 2003).

- It is not correct (as given in your Table 1) that the overlap assumption in the studies of Marquart et al. (2003) and Frömming et al. (2011) is different. Both use the maximum-random overlap principle in the radiative transfer calculations, as do Burkhardt and Kärcher (2011).

- Note that the ECHAM4 studies of Ponater et al. (2002), Marquart et al. (2003), and Frömming et al. (2011) mainly give mean optical depth values of *visible* contrails (i.e., averaged over those contrails that exceed a "visibility threshold" of

0.02), to enable comparison with observations. However, the "invisible"contrails are not excluded from the radiative forcing calculations. This may confuse an unaware viewer of your Table 1.

I also note that the visibility threshold has been a subject of debate. According to Kärcher et al. (2009) a threshold value of 0.05 is more appropriate and has been preferred in later studies (e.g., Bock and Burkhardt, 2016).

- The global mean optical depth value of 0.05 given in Table 1 for the Frömming et al. (2011) results seems to be incorrect. Table 2 in that paper provides the consistent value of 0.08 . The confusion may originate from the first paragraph of Frömming et al.'s section 3.1, where the optical depth of all contrails (including "invisible" ones below 0.02) is additionally given, and this one is indeed 0.05 .

- Finally I recommend to split your Table 1 into two parts, one referring to line-shaped contrails (first two rows), and one to contrail cirrus (last four rows). Otherwise any reader observing totally different radiative forcings for nearly the same air traffic volume will be misleaded and will mistakenly be tempted to attribute the difference to the cloud overlap assumptions!

- Line 89: As stated above, I disagree with the claim that Frömming et al. (2011) assume random overlap in the radiative transfer calculations.

- Line 239: Do you mean sufficiently *thick* or sufficiently *extended*? Yet, either assumption appears to be somewhat bold for contrails, I think.

- Line 256: "Due to the *known* strong dependence …"I think at this point the fundamental work of Markowicz and Witek (2010) on the subject ought to be acknowledged.

- Line 408: Here, or somewhat earlier, the notion of "quantifying the effect of cloud overlap by the difference of all-sky minus clear-sky" should be scrutinized a little bit. The point is that Rap et al. (2010) extensively discuss the potential effect of a correlation between contrails and natural clouds, increasing the frequency of all-sky situations with respect to clear-sky situations in comparison with a setup assuming climatological background (natural) clouds. My impression is that you do not account for this correlation in your calculation setup. If this is true, it might be fair to mention this as a caveat.

- Line 431: I think that there are also many contrails below 0.05 (see Kärcher et al., 2009). This only supports your approach to extend your parameter space down to tau = 0!

- Line 439: In situ measurements like the one cited here may not fully represent the parameter range of contrails, hence you might consider to add some citation of a satellite study such as Bedka  et al. (2012) or Minnis et al. (2013).

References

Bedka, S., et al., 2013: Properties of linear contrails in the Northern Hemisphere derived from 2006 MODIS observations, Geophys. Res. Lett., 40, 772-777.

Markowicz, K.M. and Witek, M., 2010: Sensitivity stud y of global radiative forcing due to particle shape, J. Geophys. Res., 116, D23203.

Marquart, S. and Mayer, B., 2002: Towards a reliable GCM estimate of contrail radiative forcing, Geophys. Res. Lett., 29, 1179.

Marquart, S., 2003: Klimawirkung von Kondensstreifen: Untersuchungen mit einem globalen atmosphärischen Zirkulationsmodell, DLR-Forschungsbericht 2003-16, https://elib.dlr.de/10016/ or https://edoc.ub.uni-muenchen.de/1341/ .

Minnis P., et al., 2013: Linear contrail and contrail cirrus properties determined from satellite data, Geophys. Res. Lett., 40, 3220-3226.

Myhre, G. and Stordal, F., 2001: On the tradeoff between the solat and the thermal infrared radiative impact of contrails, Geophys. Res. Lett., 28, 3119-3122.

Myhre, G. et al., 2009: Intercomparison of radiative forcing calculations for stratospheric water vapour and contrails, Meteorol. Z., 18, 585-596.

Stuber, N. and Forster, P., 2007: The impcts of diurnal variations of air traffic on contrail radiative forcing, Atmos. Chem. Phys., 7, 3153-3162.

Yi, B., et al., 2012: Simulation of the global contrail radiative forcing: a sensitivity analysis, Geophys. Res. Lett., 39, L00F03.

---

## Author Comment (AC1) · 31 Jul 2020

Massachusetts Institute of Technology 77 Mass. Ave. Office 33-322A, Cambridge MA 02139, USA seastham@mit.edu | http://lae.mit.edu | +1 617–452–2550

Editorial Office Atmospheric Chemistry and Physics

July 31st, 2020

Dear Editor,

**Re: Submission of "Effect of contrail overlap on radiative impact attributable to aviation contrails" to *Atmospheric Chemistry and Physics**

Thank you for arranging the review of our work. We thank the reviewers for their time and attention, and in particular for their comments regarding additional pertinent literature and for their suggestions regarding the paper's structure. We have now incorporated a deeper literature review, and have significantly restructured the paper to more clearly reflect the goals and novelty of the work. This includes the separation of the "Introduction" section into two components: Section 1, which motivates the work and explains our approach; and Section 2, which is now dedicated to providing a thorough overview of previous contributions in this field, and distinguishing our approach (and objectives) from theirs.

Please find below our responses to the comments from anonymous reviewers #1 and #2, as well as to the comment from Michael Ponater. In the responses below, we have listed the reviewer's comments in *italics* and our responses in **bold**. All line numbers refer to the revised manuscript, which will be submitted after this comment is posted as per *Atmospheric Chemistry and Physics* guidelines.

**Anonymous Referee #1**

The paper discusses the impact of overlap of contrails with other clouds and of contrails with other contrails for horizontally homogeneous clouds and contrails.

The paper offers a simplified radiative transport model to account for RF changed for overlapping cloud layers.

The main conclusions are:

1) Contrail-cloud overlap is important. Contrails over clear sky and contrails over other clouds have far different radiative forcings (RF).

True. However, that finding is not surprising and not new. It is not surprising because the clouds below and above contrails change the reflected solar radiation (local Earth albedo) and the outgoing longwave radiation. It is not new since you find that in several previous papers. See, for example Fig. 6, column 6, in Meerkötter et al. (1999; cited in the paper). That paper discussed the sensitivity of RF to various contrail and ambient parameters, including a cloud layer below the contrail cirrus and the optical depth of "background clouds". It showed that the net RF can increase from close to zero to a large positive value when background clouds get included. The value (94%) stated in the present paper here has no significance because it depends on the reference value and may vary from minus infinity to plus infinity when the net RF happens to be close to zero for the reference case considered. That makes no sense.

We agree that the phrasing of a 94% change was misleading in the context of an RF which is the result of changes in two components of different sign. Upon reflection we have changed the way that this is presented, providing information on the effect for both longwave and shortwave components in the abstract (lines 26-27) and conclusions (lines 1056) instead.

The present paper mentions Minnis et al (1999, cited) and Myhre et al. (2009, cited) and other studies, who discussed contrail-cloud overlap. The abstract and conclusions stress the importance and uncertainties of contrail-cloud overlap, which is correct, but report the findings as if that would be new, which is not correct.

Apparently, this discussion still reflects the history of the present paper, which apparently started with the Corti-Peters model with just one cloud layer (contrails or cirrus) over Earth surface and where the inclusion of other clouds changed the results considerably. This needs to be fully revised.

Based on the insights provided by all reviewers and commenters, we realized that the objectives of the paper were not written sufficiently clearly, and that the literature review did not do justice to the existing work in this field. We have therefore heavily reworked the introduction, splitting it into two sections as described in the opening text of this letter.

With regards to this specific comment, we have made a particular effort to clarify the novelty of this paper. We now separate out our motivation, including the specific aims of this work, in a dedicated Section 1. This is followed by a dedicated discussion of previous work on cloud-contrail overlaps (see Section 2.1) in addition to the approaches taken to account for contrail-contrail overlap in previous work, where relevant.

2) Contrail-contrail overlap depends on the number and proximity of contrails. For present traffic, contrail-contrail overlap occurs on average over the globe but only rarely. This overlap may occur more frequently for increased traffic and under special flight track conditions.

The treatment of contrail-contrail overlap is interesting. In areas with dense traffic, many contrails overlap with each other and with other clouds to different degrees. It would be good to have an efficient and still accurate method to account for the climate impact of contrails in such situation.

Contrail-contrail overlap may not be the most important uncertainty in contrail RF modelling. More important parameters may include the amount of ice supersaturation available in the atmosphere, the growth of the contrail cross-section by mixing with ambient air and the life time of contrails depending on many parameters (Schumann and Heymsfield, 2017, cited, and the references cited therein). Still, an investigation of contrail-contrail overlap effects and their modelling is of interest.

We agree with the reviewer that while this is a useful new possibility, it is certainly not the only (or even dominant) uncertainty in contrail modeling. We have made a renewed attempt to distinguish between physical uncertainties such as in available ice supersaturation or contrail ice microphysics (lines 69-71); modeling challenges such as the dependence of contrail-attributable radiative forcing on the representation of cloud-contrail overlap; and the unquantified effects of contrail-contrail overlap (lines 79-89).

**Comments on the approach and results:**

The radiative transfer model used to account for multiple layers (Section 2.1.3) looks interesting. It seems to have similarities with older theories; see, e.g., Hansen and Travis (1974) and Minnis et al. (1993); Minnis et al. (1998). A paper much cited in this respect is that of Ritter and Geleyn (1992). This part needs review by experts in this specific field.

Section 3.1.4 compares results for this model with the Fu&Liou model. That is certainly an acceptable approach, as long as one can justify the plane-parallel cloud representation. Only few details are given on how the code was applied, for example with respect to the background atmosphere and aerosols and the specific model parameters. The comparison shows qualitative agreements with the multilayer model derived, but significant quantitative differences. So, how can we be sure that the results are correct? So uncertainties remain.

Additional data have been added with regards to the experiment implementation and design (lines 703-711). We also now have a discussion emphasizing the variation of the error with solar zenith angle (lines 729-731). We have also expanded our discussion regarding how differences between our results and those using FL96 might propagate to our results (lines 948-954).

Eq. (3) needs a bit more discussion: As it is written, this equation does not guarantee that RF\_LW is positive. How often are negative values occurring?

We now clarify this on lines 346-348. As in Schumann et al. (2012), we treat negative values of  $RF_{LW}$  as being equal to zero.

Is Eq. (9) correct? 360? is that 360° (2 pi)?

This equation has been changed to instead use radians as suggested.

The discussion of the range of optical depth values (tau below 0.3) might be reasonable for global mean value, but locally the variability can be far larger (Atlas and Wang 2010).

This is correct. Our core results evaluate the impact of contrails with optical depths up to 0.5, at the upper end of contrails modeled by CERM. However we recognize that this does not cover the full range of possibilities. We have therefore generated alternative versions of Figures 2 and 3, showing results for contrail optical depths of up to 1.5 – potentially representative of more contrail-dense areas like Northern America or Europe. These figures can be found in the SI (Figures S3 and S4).

The paper presents contrail results from the model CERM. As stated, CERM does not account for contrail orientation and does not account for contrail position in a grid cell. How can one compute the contrail-contrail overlap effects without knowing the degree of overlap? The mentioned model CoCiP includes such geometry in more detail.

We apologize for the confusion on this point, and have modified the paper's introduction to clarify. As noted by the reviewer, the lack of orientation information from CERM is a limitation of this work, which we now discuss in Section 5. Accordingly, we assume maximum overlap between contrails, providing an upper bound on the effect of contrail-contrail impact with regards to radiative forcing. However, we agree that it would be an important improvement to either modify CERM to report orientation or to use results from a model such as CoCiP which already includes this data. We now state as much on lines 972-979.

How good do the meteorological data used represent humidity? Which time period is covered by the data, what is the spatial and temporal resolution of the data, and what is the vertical resolution in meters near the mean flight level height (around 10 to 12 km asl)? What is the fraction of ice supersaturated air masses in these data and how does this compare to published findings.

CERM uses a resolution of ~350 m at flight altitudes, but uses meteorological data from GEOS-FP which is provided at a slightly coarser resolution of ~500m at flight altitudes. We have an additional paper in preparation which assesses the magnitude of errors in relative humidity as calculated by GEOS, but this is out of the scope of this paper. We do however now mention known issues with relative humidity estimates in reanalysis (lines 1001-1005) including the possibility of a consistent humidity bias in reanalysis data (Jiang et al., 2015; Davis et al., 2017).

I assume, the model uses gridded emission source rates as provided by the FAA's Aviation Environmental Design Tool, but the paper gives no details on this. Are these data accessible to the community? Otherwise the results cannot be checked by other scientists.

We apologize for the oversight, and have added a sentence detailing the source of aircraft emissions data for this work on lines 399-400. Although CERM has previously been used with AEDT, we chose to instead use flight track data estimated by the Aviation Environment Inventory Code (AEIC) for this work (Simone et al., 2013). AEIC is an open-source tool, enabling independent validation of our results.

Very little is said about the satellite data CERES. The paper cites NASA Langley Research Center Atmospheric Science Data Center 2015 as reference and says that the data are provided at three-hour intervals. How can one derive hourly average values from 3-hourly data? How well do they represent the diurnal cycle and how sensitive are they to cirrus clouds and to geometrically thin contrails? How uniform is this sensitivity spatially and temporally, e.g. over land and oceans?

We now provide additional detail regarding the CERES satellite data on lines 447-473. As stated, CERES provides data once very three hours, and we assume constant cloud coverage over this period. Data are provided on a 1° by 1° global grid. We also discuss the likelihood of double-counting (i.e. contrails being observed by CERES while also being simulated by CERM) on lines 479-486. Sensitivity fo thin contrails depends on visibility thresholds. General visibility thresholds have been discussed lately and a common value is an optical depth of 0.05 (Kärcher et al., 2009), while a specific reference on MODIS instrument (used for CERES) mentions thresholds around 0.02 (Dessler and Yang, 2003). This is now mentioned in line 460-467.

The paper says that the "CERES instruments observe both contrails and natural cirrus clouds". I assume this means that CERES provides information only on the sum of contrails and other clouds. That should be clarified. **Correct. This is now clarified on lines 470-471.**

The conclusion claims that the results "help to inform policymakers and researchers to identify technical, operational, and regulatory means to reduce these impacts." I think, based on the information given, the paper is still quite far away from this goal.

Based on this comment and those from other reviewers, we have decided to remove this conclusion.

The conclusion "The radiative forcing attributable to a contrail layer increases by a factor of three due to the presence of natural clouds on a global mean basis, but this varies by region" could be formulated inversely, e.g. "if a model would ignore other clouds the results could be wrong by a factor of three", but the conclusion should also make clear that this is not state of the art. Other models do account for ambient clouds.

We agree that the previous framing was unclear regarding the goals of the paper, while also not providing enough context regarding the depth of existing literature. With that in mind, we now provide a dedicated discussion regarding our contribution on the aspect of cloud-contrail overlap – specifically, to provide a quantitative analysis of the factors which contribute to and determine the magnitude of the effect that clouds have on contrail radiative forcing (lines 73-84). We also reiterate this point in the conclusions (lines 1026-1029).

In the abstract, the growth rate of air traffic is cited. I agree, growth rates of 4.5 % each year over the next 20 years have been estimated in the past by industry, as cited. However, such trend values have large uncertainty and I would recommend omitting such uncertain values from the abstract. We have removed this statement from the abstract as suggested.

Unfortunately, the paper is not really clear and understandable and the conclusions are overselling the findings. The subject of the paper and some of the results are interesting but the approach and its presentation require major improvements.

I suggest splitting the paper into two parts: One on radiation transfer and one on the application. The first one should describe the model for the impact of cloud overlap on radiative forcing as a purely technical paper, with full validation. That paper should be reviewed by radiation transfer modelling experts. The other paper might then deal with the consequences of contrail-contrail overlap for climate forcing of aviation, addressing the corresponding community.

We hope that the reviewer finds that the revised structure of the paper is clearer, and more accurately conveys our intended message. We also hope that findings are no longer oversold. We believe that these revisions have helped to make a more coherent paper which is best packaged as a single unit, as the technical work (extending the Corti and Peter (2009) model to include multiple layers) was performed with the specific goal of enabling this comparison.

**Anonymous Referee #2**

The paper investigates the impact of cloud-contrail and contrail-contrail overlap on the radiative forcing due to contrail cirrus. The authors use a radiative transfer model of Corti and Peter that they modify in order to study the impact of cloud overlap on contrail radiative forcing. They consider two options no overlap and maximum overlap and study the difference when using those two assumptions. Cloud and contrail properties are varied systematically and the impact on LW and SW RF is analyzed. Cloud properties are prescribed using observed natural clouds (from Satellite) and CERM simulated contrails. Whereas, in principal the impact of cloud overlap is an interesting topic, the authors' extreme assumptions (no or maximum overlap) limit the relevance of the paper. They wrongly claim that other studies assume no or maximum overlap between contrails and clouds and contrails and contrails and claim that the sensitivity that they see is a measure for the bias of contrail cirrus RF published in the literature. Relevant literature that discusses the impact of overlap on RF in detail has not been discussed and comparisons with the results in the literature only partly made.

We agree that the treatment of literature in the previous version of this paper was insufficient. We have now dramatically expanded the literature review into a separate section (Section 2). This includes a dedicated review of previous attempts to quantify cloud-contrail overlap (Section 2.1) as well as the approaches used in other models to account for contrail-contrail overlap (Section 2.2).

I suggest presenting the work as a sensitivity study, including a detailed comparison with the many results in the literature and removing text stating that the present work improves contrail RF estimates and estimates the uncertainty in contrail RF in the literature or that the results help to inform policymakers and similar claims.

We have now softened our statements regarding policy applicability, including the removal of a sentence identified by Reviewer #1 as problematic in the conclusions. We have also worked to reframe the paper so that its objectives are clearer: to provide a first quantification of the effects of contrail-contrail overlap on net contrail radiative forcing, and to quantitatively investigate the relationship between physical

**parameters and the effect of contrail overlap (with natural or artificial clouds) on estimated contrail radiative forcing.**

Major comments:

1. The sensitivity of contrail RF on the overlap analyzed in this paper is not a measure for the uncertainty in contrail RF in the literature. Both assumptions used in this paper, no or maximum overlap, are very extreme whereas in the literature mostly maximum random overlap for contrail-cloud and contrail-contrail overlap has been used. The statements that in the literature mainly maximum overlap has been used (e.g. line 346- 348) or that random overlap has been used for contrail-contrail overlap (line 350) the authors partly contradict their own table 1.

This comment was very useful in directing our thoughts regarding uncertainty. We agree fully that what is being addressed here is not uncertainty so much as a previously unquantified component of contrail radiative forcing, and one where prior estimates of contrail radiative forcing had differed in their treatement. We have rephrased our study (Section 1) and have also separated out the previous treatments of contrail-contrail overlap (Table 2).

2. The estimate for contrail RF is not an improved estimate relative to the estimates in the literature We agree that this is not the purpose of this paper and have removed such claims from the text.

a. Neither maximum nor no overlap are good assumptions. Maximum overlap is certainly an upper bound for the overlap but far away from the truth.

We agree with this assessment. The purpose of the paper is to provide a quantitative estimate of 1) the effect that variations in key parameters (e.g. location, season) have on the effect of multi-layer overlap with regards to contrails; and 2) a quantitative assessment of the potential impact that contrail-contrail overlap might have on contrail RF. Since the CERM simulation does not provide contrail orientation (line 407-412), we instead quantify an upper bound using the maximum overlap assumption. This is now explicitly stated in the limitations section (lines 971-979).

b. Using only cloud data with 3 hourly temporal resolution does not allow for a realistic representation of contrail-cloud overlap or a realistic estimation of overlap frequencies. The low temporal resolution cannot resolve the correlation between cloud and contrail frequency.

We have added this to our section on limitations (line 989). Although CERES is a state-of-the-science observational product, we agree that its low temporal resolution will bias estimates of the effects of cloud-contrail overlap. Since we also find other limitations associated with the use of CERES data, we have listed the identification of alternative approaches to incorporate natural cloud data as a high priority for future work (lines 1015-1019).

c. Overlap assumptions have been shown to be dependent on vertical resolution. Even if vertically extended clouds are assumed the vertical overlap decreases strongly with layer depth. When levels are separated by more than 4km the overlap is essentially random (Hogan and Illingworth, 2000). At a low vertical resolution random overlap is a good assumption while maximum overlap is realistic (for vertically extended clouds) at high vertical resolution. At lower horizontal resolution the arguments must include a discussion of synoptic situations and the resulting vertical and horizontal statistics of the moisture field. Using observed cloud statistics aggregated in only 4 atmospheric layers and calculating the overlap with contrails the assumption of maximum overlap is far from realistic.

We agree that vertical resolution is an important factor, for multiple reasons. Although CERM is capable of simulating the formation and trajectory of contrails at essentially arbitrary vertical resolution, it is limited by the vertical resolution of the moisture field as simulated by the meteorological data (in our case GEOS-FP data have a resolution of approximately 500 m at flight altitudes). The coverage data used in this work, generated by CERM, have a vertical resolution of approximately 350 m at flight altitudes, by interpolating RH and temperature from GEOS-FP. As for the issue of the vertical resolution of clouds, we are grateful to the reviewer for pointing out the assessment by Hogan and Illingworth, which we now cite as part of a discussion of this issue on line 991. We state there that, although a flawed assumption, we believe that maximum overlap is still the most appropriate approach to estimate an upper bound on contrail overlap impacts. However we fully agree that an assessment which prioritizes a more nuanced description of layer overlap, including higher vertical and temporal resolution of natural cloud data, should be prioritized for future work (line 1015).

d. Assuming maximum overlap between contrails and contrails is not realistic. Even if planes would follow each other on the same flight track advection would mean that contrails don't maximally overlap. But this topic does not seem to be very promising as contrails have a low optical depth and the overlap between contrails does not impact the radiative forcing strongly.

It is true that planes following each other on the same flight path would not be expected to produce overlapping contrails, but patches of overlapping contrail are frequently observed in satellite imagery (e.g. Minnis et al., 2011) and model estimates (e.g. Figure 9). However, we realize that our goal of finding an upper bound on this effect is not clearly enough stated, and have made efforts to clarify this throughout the manuscript.

e. As the authors say whether a contrail cools or warms depends on the height of the clouds that may be vertically overlapping this contrail. The cloud height cannot be properly represented using cloud observations aggregated on only 4 levels.

As discussed in the prior responses, we have extended our discussion of the limitations associated with use of the CERES natural cloud observations (lines 989-995 and 1015-1019).

- 3. Comparison of the results to the previous publications should be improved.
- a. The publications of Markowicz and Witek (2011 a,b) have not be cited. They discuss the impact of contrail cloud overlap in great detail e.g. the dependence on particle habit. They also show that contrail RF turns negative for all considered ice crystal shapes at much higher optical depth (at zenith angle 30°) then shown in fig. 3a. As a zenith angle of 30° is often used in the literature it would be good to supply results for that angle in order to allow for comparison.

These references are now cited in Table 2 and Section 2.1. We have also regenerated Figures 2 to 5 using a solar zenith angle of 30° and added them to the supplementary information (Figures S3, S6, S7 and S8).

b. The results in Fig. 7 should be compared with results of e.g. Schumann et al. 2012 and Myhre et al 2009 in detail. Why are absolute values of contrail RF so different from previous results?

This comparison is included on lines 760-765. The resulting global sensitivity in all-sky conditions is of the same order as previous estimates (~100 mW/m2). The differences in clear-sky results might be due to the assumed asymmetry parameter, with a significant impact on global longwave radiative forcing in clear-sky conditions (Sanz-Morère et al. 2020).

4. The result that cloud-contrail overlap is responsible for 93% of the net radiative forcing attributable to contrails in 2015 relies on the assumption that the authors have simulated the 'true' overlap between contrails and natural clouds which is not the case. Instead they should say that overlapping contrails maximally with clouds instead of prescribing no overlap leads to an increase in radiative forcing by xx%. The same is true for the statements about the importance of contrail-contrail overlap. Note also that those values will be resolution dependent so that their significance is very limited and should not appear in the abstract!

We agree that this is misleading. We've removed the statement in question, instead providing the contribution to longwave and shortwave individually which should be more robust. We have also made this modification throughout the paper (e.g. lines 854-855).

5. Line 891-892: How do you suggest avoiding cloud contrail overlap? Contrails mostly form close to natural clouds which means that they often overlap with other clouds. Cloud-free areas are mostly dry and therefore persistent contrails cannot exist.

There have been several studies of possible contrail avoidance strategy which may be pertinent here, including recently by Teoh et al (2020). This suggests that, if high-accuracy meteorological data are available, it may be possible to avoid contrails through altitude adjustments. However, we do agree that contrail avoidance is a non-trivial task, and have added this caveat on line 1050.

**Minor comments:**

1. Table 1 is incomplete, contains mistakes and is misleading: The table serves to show the great scatter in the contrail RF estimates but it omits to say

- that the Marquart et al and the Frömming et al estimates are for line-shaped contrails only whereas the other estimates are for contrail cirrus.

- the main difference between the contrail cirrus modeling studies are the different ways of treating contrails, keeping them separate from natural clouds or treating them with the cloud scheme, and the contrail initialization. The Chen and Gettelman study follows a very different approach from the others.

- That there is another estimate for contrail cirrus RF (Bock and Burkhardt, 2016 – which is already in the literature list) that lies in between the Schumann et al. and the Burkhardt and Kärcher estimate. This means that 3 of 4 estimates lie close together.

- Chen and Gettelman include contrail-contrail overlap since overlap is dealt with in the cloud scheme. The model assumes maximum random overlap for clouds. Only at the initialization stage contrails (age of up to ~30 min.) do not overlap but that is not the same as no overlap between contrails in general. That means that the scatter between contrail cirrus RF estimates is much smaller than suggested by the current table 1 and that the scatter is not due to the overlap scheme. Even though the uncertainty in the overlap between clouds and contrails leads to an uncertainty in contrail cirrus RF, the results from the literature do not demonstrate this fact as they tend to use the same overlap scheme. The range of net RF due to contrail cirrus encompasses the estimate of this study only because estimates for line-shaped contrails are included here.

A deep analysis of previous literature has been done for this revision. Table 1 has been completely rebuilt, adding two new parts (now Tables 2 and 3), and each part reviewed for accuracy. Additionally, a whole section (Section 2) has been added to better address existing literature. This new section in particular benefited greatly from the comments of both reviewers and Michael Ponater, for which we are grateful. We no longer list the Chen and Gettelman (2013) estimate in Table 2, and instead discuss their approach on contrail-contrail overlaps on lines 185-188 of Section 2.2.

2. Biofuels have little impact on contrail formation likelihood but instead on soot number emissions, ice nucleation and contrail life times and optical depth (Moore et al. 2017, Kärcher et al. 2018, Burkhardt et al. 2018) We have adjusted this comment to make clearer that biofuels will affect the properties of contrails, and not necessarily their likelihood of formation (line 67).

3. Marquart et al (2003) and Frömming et al. (2011) both use the contrail parameterization of Ponater et al. (2002) and simulate line-shaped contrails of varying optical depth including those with an optical depth smaller than 0.02. They both calculate the fractional increase in cloudiness due to contrails. Only the overlap was calculated differently.

This has been corrected in Table 1.

4. Chen and Gettelman do not assume zero overlap between linear contrails when calculating radiation. Instead they use the overlap scheme of CAM (maximum random) in order to take care of overlap between different clouds.

This has been corrected in Table 3. Since Chen and Gettelman (2013) use a very different approach, we no longer list their estimate in Table 2, although it remains in the discussion (lines 185-188).

5. Page 6: You cite papers that determine if cloud-contrail overlap reduces or enhances contrail RF and conclude that this is a major uncertainty instead of mentioning the known fact that the effect depends on the contrail and cloud properties and temperature/height.

We agree that the prior phrasing was incorrect, and that this is a specific issue given that quantifying this dependence is a goal of our paper. We have rephrased the corresponding text in Section 2.1 (lines 169-174). We have also emphasized this in the Introduction in an attempt to better clarify the goals of the paper with regards to investigating this dependence (lines 73-84).

6. On page 7 you talk about contrail-contrail interaction. Please note that interaction can happen due to a number of processes and does not necessarily have to do with radiation.

We have removed this comment as part of the restructuring of Section 1 into two sections.

7. On page 11 the error of the model compared to FL96 is determined. In that context the systematic bias, the underestimation of the dependency on the zenith angle, should be discussed and its impact on the results of this paper.

We agree that this is important. We have extended the discussion in Section 4.1 to draw attention to the possibility of biases combining (lines 721-733). We have also extended the limitations section (Section 5.1) to discuss how these two sources of bias may affect our results (lines 951-954).

8. Line 341-344: All 3 sentences are unclear. It is not clear what it means to aggregate single contrails into one layer! Are coverages due to single contrails added up or not? 'Overlap between . . . is therefore not explicitly resolved': What does that mean – minimum overlap? 'the same approach . . ... as if clouds were centered in the grid cell'. What does that have to do with overlap. Equations would make this text easier to understand. We have completely rephrased this paragraph, and split it into two for clarity. The comment regarding some overlap not being "explicitly resolved" regarded cases where contrails might form in very close spatial and temporal proximity, an inherent limitation of the version of CERM used here.

9. Line 349: what does '(linear in most cases)' refer to? This clause has been deleted as it does not add meaning.

10. Line 349: I assume you mean to say that contrail area 'overlaps' and not 'interacts' **Thank you, this has been corrected.**

**11. Line 353: what is 'potential maximum overlap'**

This sentence has been changed to make the intended meaning clearer - that there may be more colinear contrail overlap in flight corriders, while overlaps over areas such as the mainland US may more often inolve unaligned contrails (lines 419-430).

12. Line 355-357: even more useful would be a higher temporal and vertical resolution. The lower the resolution the larger the overlap.

We agree. We have extended our discussion of the advantages of greater temporal and vertical resolution (lines 989-995).

13. In order to correct for errors in the model the authors use for the clear-sky estimates CERES data. Without any discussion where the errors are coming from it is difficult to understand if this approach is acceptable. The radiation emitted from the cloud could still be affected by this error which would mean that the estimated contrail RF would include this model error.

Lolli et al. (2017) found that the regression used by Corti and Peter (2009) to estimate longwave emissivity resulted in significant errors for lack-body temperatures greater than 288 K. We assume that those temperatures are most relevant to the surface, for which we now use the clear-sky CERES estimate. We have added clarifying sentences to this effect on lines 289-306.

**14. The term 'independent overlap' needs to be defined.**

In this work we distinguish between 'independent contrails' and 'overlapping contrails' with regards to how RF is calculated. This has been clarified in the caption for Table 4 (line 922). We compare 'independent' contrails to 'overlapping' contrails, with the former treating contrails as if there was no radiative interaction between them. A comparison between both cases can also be found in Section 4.1.1.

15. Line 434: The asymmetry parameter in Schumann et al is 0.787 (and not 0.77) for older contrails and 0.827 for younger contrails.

The choice of 0.77 is based on an analysis in a recent paper (Sanz-Morère et al 2020) which drew on the results from Schumann et al (2017) among others. We have extended the reference list appropriately and have added a brief explanation of how the number 0.77 was determined (lines 518-521).

16. Line 503: Figure 3b is the upper right panel **This has been corrected.**

17. Line 854: You probably meant to cite Kärcher et al., 2009 and not 2002. This has been corrected.

18. Line 886 – 888: Those conclusions depend very much on the type of clouds you prescribed. In the tropics many very thick clouds can be found and differences between cloud and contrail top temperature can be very large as well.

This is true. A comment to this effect has been added in the conclusions (line 1044).

**Specific comments from Michael Ponater**

**Introductory remarks**

I find this work of Sanz-Morere et al. quite interesting as it emphasizes an important aspect in contrail radiative impact studies that usually has not been investigated as systematically as is done here. In this respect, I feel the two official referees have taken a somewhat stern attitude towards the paper. I think that using a parameter scanning approach in assessing cloud-contrail and contrail-contrail overlap situations goes beyond what currently available studies have done. Figure 3 of the present paper is certainly worthwhile providing. Yet, the referees are certainly right when reminding the authors not to overreach their conclusions, given the limitations of the model framework used in the paper. It is also true, I agree, that some aspects (and even whole papers) of previous research work has been overlooked by the authors.

However, this specific comment is mainly written to provide additional information (including insider knowledge of previous papers) that can help to rectify some statements where the authors – in my view - have interpreted previous work inaccurately.

We would like to express our gratitude to Dr. Ponater for taking the time to read and comment on our manuscript. Based on this comment, and those of the reviewers, we have dramatically reworked our introductory section (now two sections), including an extended discussion of previous work which was previously not well represented.

**General comments**

• Contrail-cloud overlap has been generally accounted for in previous contrail radiative impact studies in the way the contrail radiative forcing is usually given under clear-sky as well as all-sky conditions (Myhre and Stordal, 2001; Stuber and Forster, 2007; Yi et al., 2012, and others cited by the authors). It has been a common finding that both the shortwave and the longwave radiative forcing decrease in magnitude under cloudy-sky conditions. Often, but not always, the daily mean net radiative forcing gets more positive with natural clouds included. We now dedicate a subsection (2.1) to better exploring how previous studies have quantified cloud-contrail overlap. We hope that this section provides a fairer evaluation of the literature, and better highlights the contribution of our work.

• As rightly pointed out by referee 2, contrail-contrail overlap has usually been accounted for in contrail studies with global climate models (Marquart et al., 2003; Rap et al., 2010, Burkhardt and Kärcher, 2011; Bock and Burkhardt, 2016), except for when used in idealized setups like the GCMs (ECHAM and CNRM) contributing to Myhre et al. (2009). For illustration what situations can occur in climate models, I reproduce a figure from the PhD thesis of Marquart (2003, unfortunately only available in German language). This picture makes it clear that layers with contrails not only may overlap with layers containing natural cloud, but that situations with contrails and natural clouds existing side by side in the same grid box are also possible. That renders the overlap situations in models like that rather complicated, even if the overlap principle is straightforward. Anyway, it is clear that the climate model parameterizations can include the effect. Possible grid box column situations containing contrails (yellow) and natural clouds (grey) in the ECHAM4 model of Ponater et al. (2002), Marquart (2003), and Marquart et al. (2003). Adapted from Marquart, 2003, her Figure 2.7.

Thank you very much for this insightful comment and for the useful illustration. We agree and acknowledge that our approach is an approximation only, and is most useful for quantifying an upper bound of multiple layers overlaps impacts on contrail attributable net RF. We have modified our literature review to more accurately reflect the approach taken in GCMs (Section 2), with references added in Tables 2 and 3.

Possible grid box column situations containing contrails (yellow) and natural clouds (grey) in the ECHAM4 model of Ponater et al. (2002), Marquart (2003), and Marquart et al. (2003). Adapted from Marquart, 2003, her Figure 2.7.

**Specific comments**

• It is important to realize that the GCM studies (at least those based on the ECHAM climate model) use the maximum-random overlap scheme for calculating radiative fluxes in all sky columns. This refers to contrail-cloud overlap as well as to contrail-contrail overlap situations. See, for example, Figure 4 in Marquart and Mayer (2002). This has been added in Table 2 as well as in section 2.1.

• It may be of interest for the present study that the use of the maximum-random overlap principle has been shown to create severe problems when used in an unfavorable parameterization combination, as discussed by Marquart and Mayer (2002). This has caused the radiative forcing values given by Ponater et al. (2002) to be basically incorrect (amended in Marquart et al., 2003).

We agree that this caveat is important. We have reviewed Marquart et al (2003), Marquart and Mayer (2002), and Ponater et al (2002), and have attempted to synthesize this in Section 2.1 (lines 135-140).

• It is not correct (as given in your Table 1) that the overlap assumption in the studies of Marquart et al. (2003) and Frömming et al. (2011) is different. Both use the maximum-random overlap principle in the radiative transfer calculations, as do Burkhardt and Kärcher (2011).

We have re-reviewed the appropriate literature and corrected our summary accordingly. This is now reflected in Table 2.

• Note that the ECHAM4 studies of Ponater et al. (2002), Marquart et al. (2003), and Frömming et al. (2011) mainly give mean optical depth values of visible contrails (i.e., averaged over those contrails that exceed a "visibility threshold" of 0.02), to enable comparison with observations. However, the "invisible" contrails are not excluded from the radiative forcing calculations. This may confuse an unaware viewer of your Table 1. I also note that the visibility threshold has been a subject of debate. According to Kärcher et al. (2009) a threshold value of 0.05 is more appropriate and has been preferred in later studies (e.g., Bock and Burkhardt, 2016). We have updated Table 1 accordingly, and attempted to added a brief discussion in Section 3.2.4 regarding assumptions on satellite contrail visibility, and how "invisible" contrails are treated in radiative forcing calculations (lines 447-474).

• The global mean optical depth value of 0.05 given in Table 1 for the Frömming et al. (2011) results seems to be incorrect. Table 2 in that paper provides the consistent value of 0.08. The confusion may originate from the first paragraph of Frömming et al.'s section 3.1, where the optical depth of all contrails (including "invisible" ones below 0.02) is additionally given, and this one is indeed 0.05. Table 1 has been corrected accordingly.

• Finally I recommend to split your Table 1 into two parts, one referring to lineshaped contrails (first two rows), and one to contrail cirrus (last four rows). Otherwise any reader observing totally different radiative forcings for

nearly the same air traffic volume will be misleaded and will mistakenly be tempted to attribute the difference to the cloud overlap assumptions!

We agree that this improves the paper significantly. We have split Table 1 as suggested.

• Line 89: As stated above, I disagree with the claim that Frömming et al. (2011) assume random overlap in the radiative transfer calculations. This has been corrected.

inis has been corrected.

• Line 239: Do you mean sufficiently thick or sufficiently extended? Yet, either assumption appears to be somewhat bold for contrails, I think.

We have clarified this on line 308. This is a significant assumption, and one which we would like to investigate in more detail in the future. The overall error for contrails has previously been estimated at around 10% (Gounou and Hogan 2007) (line 312), but may be greater for the purposes of investigating contrail overlap specifically. We state this in our Limitations section (lines 937-940) but now also include a statement that the inclusion of 3-D effects would be a productive topic for future work (line 1021).

• Line 256: "Due to the known strong dependence ..." I think at this point the fundamental work of Markowicz and Witek (2010) on the subject ought to be acknowledged. This reference has been added on line 326.

• Line 408: Here, or somewhat earlier, the notion of "quantifying the effect of cloud overlap by the difference of allsky minus clear-sky" should be scrutinized a little bit. The point is that Rap et al. (2010) extensively discuss the potential effect of a correlation between contrails and natural clouds, increasing the frequency of all-sky situations with respect to clear-sky situations in comparison with a setup assuming climatological background (natural) clouds. My impression is that you do not account for this correlation in your calculation setup. If this is true, it might be fair to mention this as a caveat.

We now include this in our opening literature review, but given the concerns regarding this work specifically we have also added a discussion on this topic in the Limitations section (lines 962-969).

• Line 431: I think that there are also many contrails below 0.05 (see Kärcher et al., 2009). This only supports your approach to extend your parameter space down to tau = 0! This is a good point – we have corrected line 516 to emphasize this strength.

• Line 439: In situ measurements like the one cited here may not fully represent the parameter range of contrails, hence you might consider to add some citation of a satellite study such as Bedka et al. (2012) or Minnis et al. (2013).

We have reviewed the studies suggested to ensure that our parameter range is reasonable, and have added them as references (section 3.1.1).

We would again like to thank the reviewers for their time and insight, and believe that their input during this review process has improved the paper substantially. Thank you again for considering our manuscript for publication in *Atmospheric Chemistry and Physics*, and we look forward to your response.

Best wishes,

Sebastian Eastham

**References**

Chen, C.-C. and Gettelman, A.: Simulated radiative forcing from contrails and contrail cirrus, Atmos. Chem. Phys., 13(24), 12525–12536, doi:10.5194/acp-13-12525-2013, 2013.

Corti, T. and Peter, T.: A simple model for cloud radiative forcing, Atmos. Chem. Phys., 9(15), 5751–5758, doi:10.5194/acp-9-5751-2009, 2009.

Davis, S. M., Hegglin, M. I., Fujiwara, M., Dragani, R., Harada, Y., Kobayashi, C., Long, C., Manney, G. L., Nash, E. R., Potter, G. L., Tegtmeier, S., Wang, T., Wargan, K., and Wright, J. S.: Assessment of upper tropospheric and stratospheric water vapor and ozone in reanalyses as part of S-RIP, Atmos. Chem. Phys., 17, 12743–12778, https://doi.org/10.5194/acp-17-12743-2017, 2017.

Dessler, A. E. and Yang, P.: The Distribution of Tropical Thin Cirrus Clouds Inferred from Terra MODIS Data, J. Clim., 16(8), 1241–1247, doi:10.1175/1520-0442(2003)16<1241:TDOTTC>2.0.CO;2, 2003.

Jiang, J. H., Su, H., Zhai, C., Wu, L., Minschwaner, K., Molod, A. M., and Tompkins, A. M. (2015), An assessment of upper troposphere and lower stratosphere water vapor in MERRA, MERRA2, and ECMWF reanalyses using Aura MLS observations, *J. Geophys. Res. Atmos.*, 120, 11,468–11,485, doi:10.1002/2015JD023752.

Kärcher, B., U. Burkhardt, S. Unterstrasser, and P. Minnis (2009), Factors controlling contrail cirrus optical depth, Atmos. Chem. Phys., 9(16), 6229–6254, doi:10.5194/acp-9-6229-2009.

Minnis, P., Duda, D., Palikonda, R., Bedka, S., Boeke, R., Khlopenkov, K., Bedka, K., Chee, T., 2011. Estimating Contrail Climate Effects from Satellite Data, 3rd AIAA Atmospheric Space Environments Conference. https://doi.org/10.2514/6.2011-3375

Simone, N.W., Stettler, M.E.J., Barrett, S.R.H., 2013. Rapid estimation of global civil aviation emissions with uncertainty quantification. Transp. Res. Part D Transp. Environ. 25, 33–41. https://doi.org/10.1016/j.trd.2013.07.001

Schumann, U., Mayer, B., Graf, K. and Mannstein, H.: A Parametric Radiative Forcing Model for Contrail Cirrus, J. Appl. Meteorol. Climatol., 51(7), 1391–1406, doi:10.1175/JAMC-D-11-0242.1, 2012.

---

## Referee Report (RR1)

Review of the paper by Inés Sanz-Morère, Sebastian Eastham et al.: Effect of contrail overlap on radiative impact attributable to aviation contrails

The paper got improved in many respects.

Still it needs further changes before it can be considered for publication:

It became now a rather long paper. Parts need further explanations. Other parts may be shorted instead.

Treatment of overlaps between contrails and other clouds is an important issue but not new. But the various approaches used so far are now reviewed, and I think this is valuable. I am not sure that this paper extends our understanding of contrail-cloud overlap much, but that is not the main topic.

I agree, treatment of overlaps between contrails and other contrails is likely important, at least in regions with high air traffic density, and more understanding and proper method to account for such overlaps is welcome.

Unfortunately, I am not certain that the approach and assumptions used in the modelling of overlapping clouds, Section 3.2 and Appendix B, are fully justified. In fact, I had suggested publishing the technical part separately and letting it review by radiation transfer experts. The authors decided not to go this way. O.K. But then I now have to bring forward my concerns about the model in more detail:

**Reasons for concern:**

The authors assume (line 345) that the downward LW fluxes remains unchanged when an upper-level cloud layer is induced. I questioned this assumption in my earlier review. But the response is not fully satisfactory to me. The downward fluxes change not only when the temperatures change but they change also when the upward fluxes change and cloud layers scatter long wave radiation backward. It seems that the model cannot account for this. Please discuss. But I would not simply state that these effects are negligible without further studies.

In Chapter 3.1.1, line 235: please explain how epsilon, the contrail emissivity, is computed (see Eq.4 in Corti and Peters). It is an important and simple relationship to the optical depth, but with an adjustable parameter delta, and it assumes a simplified scattering model, which may be appropriate. But how is this parameter determined? In principle this relationship should be different for SW and LW fluxes and should depend on the ice particle properties and on absorption and scattering material in the atmosphere.

Chapter 3.1.3, Line 356 ff: The authors should make clear that Eqs. (4, 5) in the present SW model are identical to the Corti&Peters model for single layered cirrus clouds. Hence, the present model adopts the same underlying assumptions like isotropic surface scattering (see text before Eq. (12) in Corti & Peter, 2009). A more realistic non-isotropic wavelength-dependent bidirectional reflectance distribution functions (BRDF) for surface scattering could change the present model equations and its results considerably. Also any wavelength–dependent absorption by water vapor in the atmosphere between surface and cirrus layer near-infrared solar radiation absorption by ice particles in the cirrus clouds is not

included in this model. So it cannot explain, e.g., the sign change shown by Myhre and Stordal (GRL, 2001, doi: 10.1029/2001GL013193), their Fig 1, for low zenith angles and large albedo values – though these negative values may be of little relevance practically.

Please note that the Corti & Peters model results were compared to a set of libRadtran simulation results as benchmark before. See Table 2, "Cloud free" cases, in Schumann et al. (2012). Here the Corti & Peters model exhibited considerable deviations, both for LW and SW components. Hence, your model may account for cloud overlaps but may introduce other problems, e.g., because of non-isotropic surface reflection or missing water vapor absorption.

Fig. A3 shows that the ratio of RF_SW values for two different surface albedos (0.3 or 0.5) can be strongly sensitive to changes in solar zenith angles, as shown by the present model and by the FL96 model application, while the COCIP RF parameterization does not show this dependency. The mentioned Myhre and Stordal (2001) results suggest an even stronger sensitivity and sign change, though cases with high albedo may occur rarely at low zenith angles in reality. Since there is very little discussion, the results may be suggesting that the new model is "better". However, I miss a fair discussion and physical explanation of these results: How was CoCiP applied for this comparison? One of the important CoCiP input parameters is the TOA reflected shortwave radiation (RSR). How was this input determined? How frequent are cases with low zenith angles and high albedo values occurring (e.g. in your study used for Fig. 7)? Therefore, this needs either a suitable discussion (which takes more space) or, if there is no clear physical explanation and fair assessment of the results, I suggest removing lines 1400 ff and Fig A3.

I doubt about the wisdom of the decision to refer the longwave RF to the clear sky outgoing longwave radiation, $OLR_{clear}$. The clear sky OLR is unknown and cannot, strictly speaking, be measured, because for cloudy atmospheres, the clear sky situation is a fictitious situation not existing in reality. Any attempt to measure or to compute the $OLR_{clear}$ is by necessity approximate. I think one of the big steps forward with COCIP was to relate the contrail RF to the top of the atmosphere irradiances, OLR and RSR. These terms have a meaning in reality and can be measured and computed. The contrails act as disturbances of the atmosphere without contrails and the OLR and RSR for the atmosphere without contrails are well described by NWP models which compute the state of the atmosphere without contrails. Moreover, contrails impact radiation properties directly mostly above other clouds, remote from the surface. Hence, I do not agree to the wording praising the progress in using $OLR_{clear}$ instead of surface temperature, in the lines near 290-295. I ask that the alternative of TOA irradiances is at least mentioned as an alternative.

Line 295: The OLR error estimates for CERES data (1.7%) may apply to its global and annual mean values, but not to local values. Local errors can well exceed 20 W/m2. There is plenty of literature on this. Please change the text accordingly

I appreciate the comparison to the Myhre et al. (2009) test case shown in Fig 6. However, the figure is hard to read and reports the results only qualitatively. Please report and discuss mean values as in the Myhre et al. paper (their Fig. 5) or in the related Table 3 of Schumann et al. (2012) and the extended Table A1 in Schumann and Graf (2013, JGR, doi: 10.1002/jgrd.50184.).

Details:

Tables 2 and 3 refer to an asterisk "*Only linear contrails considered), but I cannot see the **asterisk** in the tables. I would prefer omitting the asterisk.

Line 120: I do not understand the sentence in the bracketed version. Why should a change from liquid to ice cloud change the sign in the net RF? Is this a general finding worth mentioning?

Table 2, line Schumann et al. (2012). Please extend the table text slightly: "Parametric RF model as a function of contrail properties, longwave and shortwave fluxes from below and above the contrail, and optical depth of clouds above the contrail."

Please explain similar to: CoCiP computes the RF as a function of contrail properties (temperature, optical depth, ice particle effective radius, ice particle habit), upward fluxes in the atmosphere from below the contrails (upward longwave radiation and reflected shortwave radiation), solar constant for given time of the year, and solar zenith angle, and optical depth of clouds above the contrail. Any clouds in the atmosphere below the contrails cause changes in the upward fluxes. CoCiP takes these upward fluxes from model output from NWP or climate model results. This way, the parameterized RF takes into account changes in the contrail-RF caused by clouds below the contrails.

Table caption near Line 115: ECMWF, replace "Forecasting" by "Forecasts".

Line 139: I miss Marquart and Mayer (2002) in the list of references.

Line 146 and corresponding reference: Replace "Radel by "Rädel".

Line 208: delete "or re-emission" (or do you think about fluorescent clouds?)

Line 275: "long-loved" ?

Line 452 and other places: Replace "Kärcher, 2009" by "Kärcher et al., 2009".

Line 597: Kärcher and Burkhardt (2013) –> Burkhardt and Kärcher (2013) or missing reference.

Eq. (9) Please omit the second term "Delta RF" – you do not need it and it is misleading because RF has the unit W/m$^2$ while the term you are discussing is a unit-less ratio.

Lines 1026 -1029: Tempus? Past tempus better than present?

Reference Boeing. https://www.boeing.com/resources/boeingdotcom/commercial/market/commercial-marketoutlook/assets/downloads/cmo-2019-report-final.pdf - Please check the address.

Appendix B1.1, Eq. (B2): there is a sign error in the second part of the equation. I have not checked how subsequent equations are affected.

Line 1315: avoid "but", e.g. by replacing ", but requires an effective radius" by "for given ice particle effective radius" (effective radius is now a standard definition). In fact, the contrail and cirrus RF depend on the particle sizes. It depends actually on the ratio of particle sizes to wavelengths. This was shown

clearly long ago, e.g., by Zhang, Y., A. Macke, and F. Albers, 1999: Effect of crystal size spectrum and crystal shape on stratiform cirrus radiative forcing. *Atmos. Res.*, **52**, 59-75, doi: 10.1016/S0169-8095(99)00026-5.

At many places in the text and in the title I suggest replacing "contrail-attributable radiative forcing " by radiative forcing by contrails".  The word "attributable" suggests wider implications (Climate changes get attributed to anthropogenic activities, e.g.). In line 1057: "global contrail RF" without "-attributable" is shorter and clear enough.

After all, I agree that it is worth to discuss the effects of contrail-contrail overlap and it is worthwhile to provide approximate methods to account for this. I think the paper could still be shortened and reduced in emphasis in respect to cloud-contrail overlap issue because that is not new. With respect to the treatment of contrail-contrail overlap, the model presented is interesting but it should be clearly stated that it is based on important simplifying assumptions and approximations which the users should be aware off.

Finally a remark: A paper that the team published in parallel (Sanz-Morère et al., in Environ Sci. Technology, 2020) cites the present submission with the sentence: "The model calculates radiative transfer using a two-stream approximation and was validated by comparison against other radiative transfer models." I would say: the present study tests the model by comparison against other radiative transfer models but it does not validate it. The term "validation" is rarely appropriate when the truth is unknown.

---

## Author Response (AR2)

**Dr. Sebastian D. Eastham**
Research Scientist
Department of Aeronautics and Astronautics
Laboratory for Aviation and the Environment

[Figure]

**Massachusetts Institute of Technology**
77 Mass. Ave. Office 33-322A, Cambridge MA 02139, USA
seastham@mit.edu | http://lae.mit.edu | +1 617–452–2550

Editorial Office
Atmospheric Chemistry and Physics

November 20th, 2020

Dear Editor,

**Re: Submission of "Impacts of multi-layer overlap on contrail radiative forcing" to *Atmospheric Chemistry and Physics***

Thank you for arranging this review of our work. We thank the reviewers also for their time and dedication. Regarding the concerns raised by anonymous reviewer #1, we have made a concerted effort to more precisely define the assumptions, weaknesses, and limitations of our approach. We have also modified the title from "Effect of contrail overlap on radiative impact attributable to aviation contrails" to "Impacts of multi-layer overlap on contrail radiative forcing", based on their recommendations.

Please find below our responses to the comments from both anonymous reviewers. We have listed the reviewer's comments in *italics* and our responses in **bold**. All line numbers refer to the revised manuscript including markup (attached).

*Anonymous Referee #1*

*Review of the paper by Inés Sanz-Morère, Sebastian Eastham et al.: Effect of contrail overlap on radiative impact attributable to aviation contrails*

*The paper got improved in many respects.*
*Still it needs further changes before it can be considered for publication:*
*It became now a rather long paper. Parts need further explanations. Other parts may be shorted instead.*

*Treatment of overlaps between contrails and other clouds is an important issue but not new. But the various approaches used so far are now reviewed, and I think this is valuable. I am not sure that this paper extends our understanding of contrail-cloud overlap much, but that is not the main topic.*

*I agree, treatment of overlaps between contrails and other contrails is likely important, at least in regions with high air traffic density, and more understanding and proper method to account for such overlaps is welcome.*

*Unfortunately, I am not certain that the approach and assumptions used in the modelling of overlapping clouds, Section 3.2 and Appendix B, are fully justified. In fact, I had suggested publishing the technical part separately and letting it review by radiation transfer experts. The authors decided not to go this way. O.K. But then I now have to bring forward my concerns about the model in more detail:*
**Thank you for your further detailed review. We have attempted to address all of the concerns below. With regards to the length of the paper, we agree that it had become rather long. With this in mind we have moved some of the more technical sections to the Appendices; most notably we have moved the description of the single-contrail radiative transfer model to Appendix A, including the detailed definition of each parameter and the assumptions made. We have also moved to the same Appendix the information on the input data required for that (single contrail RF) model. We have added summarizing statements in their place in the main text (lines 231-248, 304).**

*Reasons for concern:*

*The authors assume (line 345) that the downward LW fluxes remains unchanged when an upper-level cloud layer is induced. I questioned this assumption in my earlier review. But the response is not fully satisfactory to me. The downward fluxes change not only when the temperatures change but they change also when the upward fluxes*

*change and cloud layers scatter long wave radiation backward. It seems that the model cannot account for this. Please discuss. But I would not simply state that these effects are negligible without further studies.*

**It is true that the model does not account for longwave radiation scattering. This is due to the assumptions made by Stephens et al. (1990) which were then carried through to the work by Corti and Peter (2009). We have assumed in this work that longwave scattering will not significantly affect the results, but in light of these comments we have tried to provide a more quantitative assessment of the implications.**

**Reviewing the literature, we saw that cloud longwave scattering is frequently neglected on the basis that longwave scattering (unlike shortwave scattering) has only a small effect on the total radiative effect of clouds. Prior studies of cloud radiation interactions have estimated cloud shortwave single scattering albedos of ~1, compared to longwave scattering albedos of ~0.5 (Stephens, 1980; Stephens et al., 1990). Ritter and Geleyn (1992) suggested that the effect of longwave scattering on high clouds could be to reduce the outgoing longwave radiation (OLR), relative to no scattering, by up to 20 W/m$^2$ (compared to typical OLR values of 260 W/m$^2$), with a global mean error of 8 W/m$^2$ (Stephens et al., 2001). Costa and Shine (2006) found that neglecting longwave scattering would cause an error of approximately 10% in cloud longwave forcing (~3 W/m$^2$ error in OLR).**

**This likely implies that the warming effect of clouds is underestimated with our model, but it is not clear what this means for estimates of the effects of contrail overlap. However, comparisons show that the Corti and Peter model is typically able to reproduce the results of more complex models such as Fu-Liou (Fu and Liou, 1993) and CoCiP (Schumann et al., 2012) and we find in Appendix B that the single-contrail longwave RF estimated by each model matches to within 10%. Since the version of the Fu-Liou model which we are using for our comparisons (Fu and Liou, 1993; Fu et al., 1997; Liou et al., 1998) includes longwave scattering, this provides additional confidence in the assumption, associating that difference to the longwave radiative scattering neglection. However, parameterizations of longwave scattering for radiative transfer models exist in literature (Chou et al. (1999) and Tang et al. (2018)), and those would improve the estimate of RF$_{LW}$ compared to our current model. In summary, we chose to neglect longwave radiation scattering but acknowledge that this comes at the cost of an error in cloud RF$_{LW}$ within 10%.**

**We now include a discussion of some of the limitations of the Corti and Peter approach to calculation of longwave radiative transfer, including the lack of longwave scattering and opportunities to rectify this issue, in the manuscript. Said discussion covers the above points and can be found in the Limitations section (lines 827-838). This is also commented upon in the single contrail radiative forcing comparison case, that can now be found in Appendix B (lines 1445-1447).**

**We would also like to note that, thanks to this comment, we realized that there was a typo in our write-up of the longwave radiative forcing calculation for multiple-layer overlap (Eq. 7). This was present in the text only (not the model), and has now been corrected (line 302).**

*In Chapter 3.1.1, line 235: please explain how epsilon, the contrail emissivity, is computed (see Eq.4 in Corti and Peters). It is an important and simple relationship to the optical depth, but with an adjustable parameter delta, and it assumes a simplified scattering model, which may be appropriate. But how is this parameter determined? In principle this relationship should be different for SW and LW fluxes and should depend on the ice particle properties and on absorption and scattering material in the atmosphere.*

**Corti and Peter (2009) assume zero scattering in the longwave radiation regime, using the parameter "emissivity" to represent the interaction of the contrail with that type of radiation. Shortwave radiation is assumed not to be absorbed or reemitted by the contrail, so no emissivity parameter is required.**

**We calculate longwave contrail emissivity using the same approach as Corti and Peter. We use the same value of the parameter "delta" as is reported in their work, calculated using an approach from Stephens et al. (1990) which assumes zero longwave scattering. As mentioned in that work, the approach has been extensively used as a parameterization of cloud IR radiative transfer. We reuse this value without modification on the basis that Corti and Peter (2009) report that a 10% change in the delta parameter changes the calculated longwave radiative forcing by only about 1%. This has been added in the main paper (lines 1253-1259). The assumed value for delta is consistent with a single-scattering value of approximately 0.55, consistent with Stephens (1980). Based on the observed variation they report in longwave single scattering albedo, we would expect delta to vary by less than 3%, implying that the error due to this assumption is small. However, it is true that this parameter will realistically be dependent on additional properties of the contrail such as the size and habit of the ice crystals. We have added this caveat to the limitations section (lines 827-829).**

*Chapter 3.1.3, Line 356 ff: The authors should make clear that Eqs. (4, 5) in the present SW model are identical to the Corti&Peters model for single layered cirrus clouds. Hence, the present model adopts the same underlying assumptions like isotropic surface scattering (see text before Eq. (12) in Corti & Peter, 2009). A more realistic non-isotropic wavelength-dependent bidirectional reflectance distribution functions (BRDF) for surface scattering could change the present model equations and its results considerably.*

**This is true. While we chose that model for simplicity and efficient applicability, it does introduce some biases in the results due to the assumptions taken. Clarification of the fact that we are using the same equations from Corti and Peter (2009) for SW radiative forcing calculations has been added in line 271.**

**The isotropic wavelength-independent assumption for calculating shortwave radiative forcing model is now mentioned in the Limitations section (line 848). This assumption is included in the two-stream approximation (Coakley and Chylek, 1975) used in Corti and Peter (2009). We now explicitly state this on line 850, clarifying that this assumption has been shown to give results which are accurate to within approximately 15%, based on comparisons in Coakley and Chylek (1975) with numerical simulations, at optical depths below ~1 and solar zenith angles below 75°.**

*Also any wavelength–dependent absorption by water vapor in the atmosphere between surface and cirrus layer near-infrared solar radiation absorption by ice particles in the cirrus clouds is not included in this model. So it cannot explain, e.g., the sign change shown by Myhre and Stordal (GRL, 2001, doi: 10.1029/2001GL013193), their Fig 1, for low zenith angles and large albedo values – though these negative values may be of little relevance practically.[...]*

**Following this comment, we decided to investigate the disagreement between our model and the results from Myhre and Stordal (2001). Reproducing their Figure 1 (below,) we find that we are able to qualitatively reproduce the sign reversal they describe at low solar zenith angles and high values of albedo. However, we agree that wavelength-dependent calculations would improve the accuracy of our estimates, and this is now stated on line 848. Indeed, Myhre and Stordal (2001) use a multi-stream model with four bands in the solar regime, which is likely a contributor to some of the differences we observe between our model and theirs at very low values of albedo and solar zenith angle. These differences are also found in our comparison case (see Appendix B), however, they are consistently less than 30%. Although, as the reviewer points out, these situations are relatively uncommon (specifically occasions with both very low albedo and solar zenith angles lower than 30 degrees), this means that our model will overestimate contrail cooling under such circumstances. Most consistent results between the three models tested in Appendix B are found at the global average albedo value (= 0.3), with errors lower than 10%. The differences shown at high solar zenith angle are also commented upon in Appendix B.**

[Figure]

**Figure 1:** Shortwave radiative forcing due to a single contrail as a function of solar zenith angle and land surface albedo. Left: results from our model; right: reference figure from Myhre and Stordal (2001). Contrail properties have been adapted to obtain similar magnitude of results: optical depth of 0.002, asymmetry parameter of 0.77.

*Please note that the Corti & Peters model results were compared to a set of libRadtran simulation results as benchmark before. See Table 2,"Cloud free" cases, in Schumann et al. (2012). Here the Corti & Peters model exhibited considerable deviations, both for LW and SW components. Hence, your model may account for cloud overlaps but may introduce other problems, e.g., because of non-isotropic surface reflection or missing water vapor absorption.*

**This is true. Although we do include a comparison in Appendix B of our model to models such as Fu-Liou (Fu and Liou, 1993; Fu, 1996) and CoCiP (Schumann et al., 2012), we agree that there are systematic biases due (in part) to certain physical processes being neglected or simplified. These assumptions are now listed as limitations in section 5, and we have introduced a comment in line 816 on the potential magnitude of the errors based on the comparison developed for Table 2 in Schumann et al. (2012).**

*Fig. A3 shows that the ratio of RF_SW values for two different surface albedos (0.3 or 0.5) can be strongly sensitive to changes in solar zenith angles, as shown by the present model and by the FL96 model application, while the COCIP RF parameterization does not show this dependency. The mentioned Myhre and Stordal (2001) results suggest an even stronger sensitivity and sign change, though cases with high albedo may occur rarely at low zenith angles in reality. Since there is very little discussion, the results may be suggesting that the new model is "better". However, I miss a fair discussion and physical explanation of these results: How was CoCiP applied for this comparison? One of the important CoCiP input parameters is the TOA reflected shortwave radiation (RSR). How was this input determined? How frequent are cases with low zenith angles and high albedo values occurring (e.g. in your study used for Fig. 7)? Therefore, this needs either a suitable discussion (which takes more space) or, if there is no clear physical explanation and fair assessment of the results, I suggest removing lines 1400 ff and Fig A3.*

**Upon reviewing this part of the appendix, we agree that this figure and discussion do not fairly represent CoCiP. Since the paper is already rather long, we decided to remove the figure and corresponding lines, as suggested. To answer the question on data used for the CoCiP simulation in Appendix B (Appendix A in the last version), we use the same TOA shortwave radiation as was used in the simulation with FL (line 1433).**

*I doubt about the wisdom of the decision to refer the longwave RF to the clear sky outgoing longwave radiation, OLRclear. The clear sky OLR is unknown and cannot, strictly speaking, be measured, because for cloudy atmospheres, the clear sky situation is a fictitious situation not existing in reality. Any attempt to measure or to compute the OLRclear is by necessity approximate. I think one of the big steps forward with COCIP was to relate the contrail RF to the top of the atmosphere irradiances, OLR and RSR. These terms have a meaning in reality and can be measured and computed. The contrails act as disturbances of the atmosphere without contrails and the OLR and RSR for the atmosphere without contrails are well described by NWP models which compute the state of the atmosphere without contrails. Moreover, contrails impact radiation properties directly mostly above other clouds, remote from the surface. Hence, I do not agree to the wording praising the progress in using OLRclear instead of surface temperature, in the lines near 290-295. I ask that the alternative of TOA irradiances is at least mentioned as an alternative.*

**We agree. Our intention is to identify metrics which can be easily and accurately extracted from models (for future simulations) or observations (for past scenarios), and it is true that the all-sky OLR has significant advantages in the latter situation in particular. The use of all-sky OLR is now discussed as an alternative option on lines 1323-1326 of Appendix A (the new location of the discussion of the single contrail radiative transfer model).**

*Line 295: The OLR error estimates for CERES data (1.7%) may apply to its global and annual mean values, but not to local values. Local errors can well exceed 20 W/m2. There is plenty of literature on this. Please change the text accordingly.*

**Thank you for catching this error. The text has been changed accordingly, based on literature estimates of potential local errors in CERES data (Loeb et al., 2018) (line 1318).**

*I appreciate the comparison to the Myhre et al. (2009) test case shown in Fig 6. However, the figure is hard to read and reports the results only qualitatively. Please report and discuss mean values as in the Myhre et al. paper (their Fig. 5) or in the related Table 3 of Schumann et al. (2012) and the extended Table A1 in Schumann and Graf (2013, JGR, doi: 10.1002/jgrd.50184.).*

**The figure has been increased in size to improve clarity, and we have added an extended discussion of the mean values in the SI (lines 125-138). This includes a new Table (S1) which compares mean shortwave, longwave, and net RF sensitivity values to those in Myhre et al. (2009), Schumann et al. (2012) and Schumann and Graf (2013). This discussion is now referenced in the main paper on line 649-653. Clear-sky results are consistent with existing values in literature, accounting for the global sensitivity variation due to contrail microphysics that is already commented on in Sanz-Morère et al. (2020). All-sky results are consistent in net effect, but longwave and shortwave terms are lower. This is likely due to the maximum cloud-contrail overlap assumption that is used.**

*Details:*

*Tables 2 and 3 refer to an asterisk "*Only linear contrails considered), but I cannot see the asterisk in the tables. I would prefer omitting the asterisk.*

**This was a typo and has been removed.**

*Line 120: I do not understand the sentence in the bracketed version. Why should a change from liquid to ice cloud change the sign in the net RF? Is this a general finding worth mentioning?*

**We realized that the phrasing of this statement was confusing, as it gave the impression that only the optical properties of the individual particles had changed. The comparison by Spangenberg et al. (2013) used observations of natural liquid and ice clouds, and as such included all of the differences which are typical between the two. This includes not only the different asymmetry parameters but also the greater optical depth, higher temperature, and lower altitude of liquid clouds. Motivated in part by this, we investigate how these factors influence the effect of cloud-contrail RF (Section 4.1.1, lines 450-463). With regards to this comment, we have modified the description of Spangenberg et al (2013)'s findings to try and make it clearer (lines 118-121).**

*Table 2, line Schumann et al. (2012). Please extend the table text slightly: "Parametric RF model as a function of contrail properties, longwave and shortwave fluxes from below and above the contrail, and optical depth of clouds above the contrail." Please explain similar to: CoCiP computes the RF as a function of contrail properties (temperature, optical depth, ice particle effective radius, ice particle habit), upward fluxes in the atmosphere from below the contrails (upward longwave radiation and reflected shortwave radiation), solar constant for given time of the year, and solar zenith angle, and optical depth of clouds above the contrail. Any clouds in the atmosphere below the contrails cause changes in the upward fluxes. CoCiP takes these upward fluxes from model output from NWP or climate model results. This way, the parameterized RF takes into account changes in the contrail-RF caused by clouds below the contrails.*

**We have made the suggested changes. Both Table 2 and Section 2.1 (lines 163-167) now include this information when discussing CoCiP.**

*Table caption near Line 115: ECMWF, replace "Forecasting" by "Forecasts".*
**This has been corrected on line 114.**

*Line 139: I miss Marquart and Mayer (2002) in the list of references.*
**The reference has been added on line 1126.**

*Line 146 and corresponding reference: Replace "Radel by "Rädel".*
**This has been corrected throughout the paper.**

*Line 208: delete "or re-emission" (or do you think about fluorescent clouds?)*
**This has been deleted.**

*Line 275: "long-loved"?*
**This typo has been corrected.**

*Line 452 and other places: Replace "Kärcher, 2009" by "Kärcher et al., 2009".*
**This has been corrected throughout the paper.**

*Line 597: Kärcher and Burkhardt (2013) –> Burkhardt and Kärcher (2013) or missing reference.*
**This was a missing reference: "Effects of optical depth variability on contrail radiative forcing", by *Kärcher and Burkhardt (2013)*. The reference has now been added.**

*Eq. (9) Please omit the second term "Delta RF" – you do not need it and it is misleading because RF has the unit W/m2 while the term you are discussing is a unit-less ratio.*
**Agreed - the term has been deleted (line 511).**

*Lines 1026 -1029: Tempus? Past tempus better than present?*
**We have chosen to present all previous literature as "past" and all work in this manuscript as "present". We have performed another sweep of the manuscript to try and ensure that this treatment is consistent throughout.**

*Reference                                                                                                                     Boeing.*
*https://www.boeing.com/resources/boeingdotcom/commercial/market/commercialmarketoutlook/ assets/downloads/cmo-2019-report-final.pdf - Please check the address.*
https://www.boeing.com/resources/boeingdotcom/market/assets/downloads/2020_CMO_PDF_Download.pdf
**This reference has been corrected and updated to 2020 (line 66).**

*Appendix B1.1, Eq. (B2): there is a sign error in the second part of the equation. I have not checked how subsequent equations are affected.*
**We apologize for this typo in the formulation. This has been corrected in every occasion in Section C1 (lines 1519, 1578). This typo was present only in the manuscript and did not affect the model code.**

*Line 1315: avoid "but", e.g. by replacing ", but requires an effective radius" by "for given ice particle effective radius" (effective radius is now a standard definition). In fact, the contrail and cirrus RF depend on the particle sizes. It depends actually on the ratio of particle sizes to wavelengths. This was shown clearly long ago, e.g., by Zhang, Y., A. Macke, and F. Albers, 1999: Effect of crystal size spectrum and crystal shape on stratiform cirrus radiative forcing. Atmos. Res., 52, 59-75, doi: 10.1016/S0169-8095(99)00026-5.*
**This has been changed as suggested (line 1424).**

*At many places in the text and in the title I suggest replacing "contrail-attributable radiative forcing " by radiative forcing by contrails". The word "attributable" suggests wider implications (Climate changes get attributed to anthropogenic activities, e.g.). In line 1057: "global contrail RF" without "-attributable" is shorter and clear enough.*
**We agree that this wording is more precise and have made the relevant change throughout the text, including in the title, changed from "Effect of contrail overlap on radiative impact attributable to aviation contrails" to "Impacts of multi-layer overlap on contrail radiative forcing".**

*After all, I agree that it is worth to discuss the effects of contrail-contrail overlap and it is worthwhile to provide approximate methods to account for this. I think the paper could still be shortened and reduced in emphasis in respect to cloud-contrail overlap issue because that is not new. With respect to the treatment of contrail-contrail overlap, the model presented is interesting but it should be clearly stated that it is based on important simplifying assumptions and approximations which the users should be aware off.*
**We have moved some of the technical description to the Appendices or Supplementary Information in an effort to shorten the paper, although all such discussion is still referred to in the main text. We have also made an effort to clarify simplifying assumption and approximations, bringing these together in our extended Limitations section.**

*Finally a remark: A paper that the team published in parallel (Sanz-Morère et al., in Environ Sci. Technology, 2020) cites the present submission with the sentence: "The model calculates radiative transfer using a two-stream approximation and was validated by comparison against other radiative transfer models." I would say: the present study tests the model by comparison against other radiative transfer models but it does not validate it. The term "validation" is rarely appropriate when the truth is unknown.*
**We agree. The word "validate" has been removed and changed to "compare" or "evaluate" (as appropriate) throughout the paper.**

*Anonymous Reviewer #2*

*The paper was significantly improved by a clearer definition of the goals of the paper, sharpening the arguments and a major rewrite of the parts summarizing the background literature. The paper in its current form is very interesting and contains useful information that adds to our understanding of contrail RF. Some parts of the paper still contain parts/sentences from the prior paper version that the authors agreed in their answers to remove or wording that needs to be corrected. A few small corrections should be made to the paper before it can be published.*
**Thank you for this review, and we were pleased that you found the revisions to be an improvement. We also apologize for the oversights which resulted in agreed-upon changes not being fully realized in the previous revisions. In addition to responding to the specific comments below, we performed an additional sweep to ensure that there were no additional oversights of this nature from the previous review.**

*Comments:*
*1. You say that 'Contrail contrail overlap can likely be neglected'. But you actually only look at whether it can be neglected when estimating the radiative response. Contrails forming in older contrails, which seems to be part of your contrail-contrail overlap, may still have a significant impact microphysically. It should be specified that contrail contrail overlap can likely be neglected radiatively which leaves it open if it might have an impact microphsically.*
**T**his has been corrected to clarify that only radiative impacts of overlaps are negligible (line 801).**

*2. You may want to cite the new Lee et al. (2020) assessment instead of Lee et al. (2009) as this has been published recently in Atmospheric Environment.*
**Thank you for this indication, the reference has been updated (line 61).**

*3. Page 9: Burkhardt and Kärcher (2011) and Bock and Burkhardt (2016) do not use the approach of Ponater et al.*

*(2002) who simulates line-shaped contrails. Instead they use a very different, process-based contrail cirrus parameterization based on Burkhardt and Kärcher (2009). Please correct this.*
**This has been corrected in line 185.**

*4. Page 19 line 422-424: contrail-contrail overlap is mostly assumed to occur with maximum-random (and not random) overlap in climate models (see your table 3). What does 'true maximum overlap' mean? How do you know what is the truth? I thought we agreed that maximum overlap is an extreme assumption.*
**The word "true" has been removed, as we agree that it doesn't provide any additional information (line 327).**

*5. Page 21 Line 470-471: How would it be possible to capture the effect of changes in natural cloudiness due to the presence of contrails in observational data while not including the contrails themselves? Either you have a data set of natural clouds plus contrails which will include the natural cloud adjustments to contrails or you may choose only situations in which you believe no contrails can be present and then the data won't include cloud adjustments or contrails.*
**This is true. Our previous statement was imprecise and has been replaced. We now instead state (lines 358-362):**

**"Finally, contrail cirrus may also modify "natural" cloud coverage by changing the availability of atmospheric water. Any such effects would be inherently included in observations, including those retrieved by CERES for year 2015. Our approach does not allow us to separate out the effect of this interaction, but its impact has previously been estimated to reduce global contrail radiative forcing by approximately a fifth (Burkhardt and Kärcher, 2011) and by 15% (Schumann et al., 2015)."**

*6. Page 21 Line 483-484: How can a nonlinearity be representative of a bias? Additionally, according to your answers to my last review we agree that your study does not give an estimate of the bias of the radiative forcing estimates in the literature. You should remove this sentence together with the sentence on page 24 line 535-536.*
**This concept has been removed throughout the paper.**

*7. In section 4.2 and 4.3.1 it should be mentioned at the beginning of those sections that you assume maximum overlap.*
**This has been added for improving clarity (lines 620-621, 753-757).**

*8. You still use often the word 'interaction' instead of 'overlap'. You should go through the whole paper and make sure that you use the correct word.*
**Agreed. The incorrect use of "interaction" when we instead mean "overlap" has been fixed throughout the paper.**

*9. Page 44, line 1012: the overlap of contrails and clouds could only be calculated (instead of making overlap assumptions) if the model resolution gets close to a few km and fractional cloud cover schemes are not needed anymore. As soon as you need a fractional cloud cover scheme you do not know which part of a model grid box is cloudy and which part is cloud free. This means that you cannot calculate overlap without making overlap assumptions. Furthermore, it is necessary to prescribe real flight movements connected with a particular synoptic situation as flight tracks depend on and clouds vary with meteorology.*
**We agree. We now clarify this in the paper on lines 917-919.**

*10. The section of 'Priorities for future work' should include improvements in the temporal resolution of cloud data. If the temporal resolution is not improved then the correlation of contrail and natural clouds and therefore their overlap cannot be captured.*
**This is now included on line 922.**

*11. Page 45, line 1028-1029: I agree that the results in this paper improve our understanding of the factors which contribute to global contrail RF, but I do not agree that the work helps us improve estimates of contrail RF – I thought we had agreed on that. I also cannot find any further mention of a resulting improvement of the current estimates in the conclusions. If you are talking about CERM not having considered overlap with clouds before then please say that the work improves estimates made by CERM.*
**We apologize for this; the sentence should have been removed previously. We have removed the statement in question (lines 934-935).**

[revised manuscript text omitted]

---

## Author Response (AR3)

Editorial Office Atmospheric Chemistry and Physics

December 27th, 2020

Dear Editor,

**Re: Submission of "Impacts of multi-layer overlap on contrail radiative forcing" to Atmospheric Chemistry and Physics**

Thank you again for arranging the reviews of our work. We thank the reviewers also for their time and dedication. We have addressed each of the comments made by the two reviewers in the final submitted manuscript.

Please find below our responses to the comments from both anonymous reviewers. We have listed the reviewer's comments in *italics* and our responses in **bold**. All line numbers refer to the revised, clean manuscript.

Anonymous Referee #1

Nice paper! Thank you!

Line 21: misses an "of" after "sign" This has now been corrected.

Reference Myhre and Stordal (202) is missing The appropriate reference (Myhre and Stordal, 2001) has now been added to the bibliography.

**Anonymous Reviewer #2**

As I commented already in my last review the paper is much improved. Therefore, I had only a quick look through the paper and want to comment on a couple of small mistakes and suggest slight changes. I want to congratulate the authors on this interesting paper.

We are grateful for the reviewer's comments, which we agree has resulted in a much improved paper.

Comments:

1. Table 1:

*a)* The Burkhardt and Kärcher estimates were calculated for air traffic of the year 2002 for both linear contrails and contrail cirrus. Please correct the year. **Done.**

b) The Chen and Gettelman estimate was revisited and corrected in the Lee et al 2020 paper  $\mathbb{Z}$  From Lee et al: 'The estimate of Chen and Gettelman (2013) was corrected by redoing the CAM simulation using a lower ice crystal radius of 7 µm and a larger contrail cross-sectional area of 0.09 km2 for the initialization of contrails at an age of about 15–20 min, in agreement with observations (Schumann et al., 2017). The resulting change in cirrus cloudiness including the adjustment in cloudiness due to the presence of contrail cirrus leads to a radiative forcing of 57 mW m– 2.' It would be good to note this in your table.

A comment has been added in the caption which clarifies this adjustment and directs the reader to Lee et al 2020.

2. Line 65: When you talk about the factors that are important for estimating the increase in contrail cirrus radiative forcing in the future you omitted 'climate change' as a factor. Chen and Gettelman (2016) as well as Bock and Burkhardt (2019) discuss (amongst other factors) the impact of climate change on the contrail cirrus radiative forcing for future air traffic.

We have added "changes in background conditions due to climate change" as a factor and cite both papers (lines 65-66).

3. Line 67: The impact of a reduction in initial ice crystal numbers (as resulting from a use of biofuels) on contrail properties is analyzed in detail in Burkhardt et al. (2018). They show a shortening of contrail life times and a decrease in contrail optical depth

We now cite Burkhardt et al (2018) when discussing the effect of use of biofuels (line 65).

4. Lee et al (2020) includes a long discussion of uncertainties connected with contrail cirrus radiative forcing (see their Appendix E) discussing uncertainties in the upper tropospheric water budget, contrail cloud overlap, contrail life times and other factors

We have added Lee et al (2020) to the sources cited in the discussion (line 71).

5. Line 173: 'These approaches'.... is a bit unclear. I suggest 'The approaches estimating the impact of cloud overlap on contrail radiative forcing' ...

We have changed the wording as recommended (line 175).

6. Line 323: 'In several previous radiative forcing calculations in literature, clouds and contrails have been assumed to maximally overlap,'. You have deleted the entries in table 1 which specified the overlap schemes connected with the different models, but as far as I can see all entries would have been 'maximum random overlap' except for the Schumann and the Spangenberg estimate. Either you state that most models use maximum random overlap or you cite exactly which models use maximum overlap.

We have modified this discussion (lines 325-329) to more accurately describe the approaches used in the past. We now specify the two examples cited by the reviewer.

7. Line 326: According to your table 3, two models use random overlap while four models use maximum random overlap. Therefore, most climate models assume maximum random overlap. Please correct this. **The relevant discussion on lines 331-333 has been corrected appropriately.**

8. Line 362: Please note that Bickel et al. (2020) estimate a much stronger reaction of natural clouds to the presence of contrail cirrus.

Bickel et al (2020)'s estimate is now included in our comparison on lines 364-365.

9. Line 739: It may be of interest to compare the contrail cloud overlap in this study with the results of Bock and Burkhardt (2016) who find a large variability in the fraction of contrail coverage that leads to an increase in overall cloud coverage. On average this fraction amounts to 43% while over the Northern Atlantic and Northern Pacific this fraction is about 20%.

We agree that this is an interesting and relevant comparison, as it may explain some of our observed variations. We have modified the discussion on lines 791-794 to call out the possible influence of this factor.

Thank you again for your time and effort in arranging the reviews of this paper. We hope you now find it satisfactory for publication in *Atmospheric Chemistry and Physics*.

Regards,

Sebastian D. Eastham